# Towards Understanding Parametric Generalized Category Discovery on Graphs

**Bowen Deng** [* 1 2]  **Lele Fu** [* 2]  **Jialong Chen** [1]  **Sheng Huang** [2]  **Tianchi Liao** [3]  **Tao Zhang** [2]  **Chuan Chen** [1]

## Abstract

Generalized Category Discovery (GCD) aims to identify both known and novel categories in unlabeled data by leveraging knowledge from old classes. However, existing methods are limited to non-graph data; lack theoretical foundations to answer *When and how known classes can help GCD*. We introduce the Graph GCD task; provide the first rigorous theoretical analysis of *parametric GCD*. By quantifying the relationship between old and new classes in the embedding space using the Wasserstein distance $W$, we derive the first provable GCD loss bound based on $W$. This analysis highlights two necessary conditions for effective GCD. However, we uncover, through a Pairwise Markov Random Field perspective, that popular graph contrastive learning (GCL) methods inherently violate these conditions. To address this limitation, we propose SWIRL, a novel GCL method for GCD. Experimental results validate our (theoretical) findings and demonstrate SWIRL's effectiveness.

## 1. Introduction

Graph machine learning (GML) (Wu et al., 2021b; Liu et al., 2022a) has made notable strides in many fields like recommendation systems (Pal et al., 2020) and financial risk management (Motie & Raahemi, 2024). Traditional GML assumes all categories are known during training, limiting its ability to handle unknown new classes during testing in real-world open-world scenarios. Prior solutions, such as Graph Open-set Recognition (Wu et al., 2021a; Zhang et al., 2023a; 2024), detect and reject nodes from unseen classes, while Graph Novel Category Discovery (GNCD) (Jin et al.,

2024; Hou et al., 2024) aims to identify and label novel classes. However, GNCD assumes only new-class samples during testing, which is often unrealistic. We introduce Graph Generalized Category Discovery (GGCD), which addresses both old and new-class nodes during testing.

Compared to GGCD, Visual GCD (VGCD) has made significant progress. VGCD methods consist of two main parts: contrastive learning (CL) and classification. Based on the classifier type, VGCD can be divided into: **1)** non-parametric methods, such as Vanilla GCD (Vaze et al., 2022), which use classifiers like Semi-Supervised K-means (SS-KM); **2)** parametric GCD methods, such as SimGCD (Wen et al., 2023), that use parametric classifiers like MLPs. Regardless of the method or data type, a common consensus is that leveraging knowledge from old classes aids in distinguishing new classes. However, the theoretical mechanisms behind this consensus are unclear, motivating our research:

***Question 1: When and how do old classes help (parametric) generalized category discovery on graphs?***

Despite empirical progress, the theoretical foundations of GCD remain limited. Chiaroni et al. (2023) proposed maximizing the mutual information between representations and classifier predictions. Rastegar et al. (2024) derived and minimized an upper bound on the Kullback-Leibler (KL) divergence between the predicted distribution $\hat{p}(\hat{y}_i = \hat{y}_j)$ and the true one $p(y_i = y_j)$ over all $(i, j)$ pairs. Both theories fail to elucidate the intrinsic role of old-class knowledge in new-class discovery and lack related provable GCD loss bounds, due to the lack of formal analysis of the relationship between old and new classess and its impact on GCD.

Intuitively, when the embeddings of old classes $l$ and $l'$ are well-separated, distinguishing between new classes $u$ and $u'$ can be achieved by aligning their embeddings with those of $l$ and $l'$, respectively. Specifically, The closer the predictions for $u$ are to those for $l$, and for $u'$ to $l'$, the more effectively the knowledge for differentiating $l$ and $l'$ transfers to $u$ and $u'$. Formally, we quantify this relationship, considering both embeddings and labels, using the Wasserstein distance $W$ between the joint (embedding, label) distributions of old and new classes. Following the common practice of replacing discrete metrics with differentiable surrogate losses, we define a surrogate loss Eq. 5 that faithfully reflects GCD performance. We then derive

---

[*]Equal contribution  [1]School of Computer Science and Engineering, Sun Yat-Sen University, Guangzhou, China  [2]School of Systems Science and Engineering, Sun Yat-Sen University, Guangzhou, China  [3]School of Software Engineering, Sun Yat-Sen University, Zhuhai, China. Correspondence to: Chuan Chen <chenchuan@mail.sysu.edu.cn>.

*Proceedings of the $42^{nd}$ International Conference on Machine Learning*, Vancouver, Canada. PMLR 267, 2025. Copyright 2025 by the author(s).

an upper bound for this loss based on $W$. Analyzing the influence of $W$ on this upper bound reveals that effective GCD requires constraining the global structure of categories in the embedding space. However, mainstream graph contrastive learning (GCL) methods mainly optimize pairwise local structures, failing to impose these global constraints. Our theoretical analysis using Pairwise Markov Random Field (PMRF) confirms this limitation. Additionally, GCN (Kipf & Welling, 2017) encoders' smoothing effect mixes categories, also disrupting the global structure. To address these, we propose SWIRL, a new GCL method that provides improved control over the relationships among categories in the embedding space. **Our main contributions are:**

1. We introduce the GGCD task for graph data and adapt multiple VGCD methods for GGCD.

2. We define a GCD loss, quantify the relationship between old and new classes using Wasserstein distance, and then develop the first provable GCD loss upper bound theory (Theorem 3.5), answering **Question 1**.

3. We identify the negative impact of the GCN encoder on GCD, and provide the first PMRF-based analysis of GCL, revealing the undesired randomness of category relations in the embedding space (Theorem 4.3).

4. We propose SWIRL, a new GCL method for GGCD.

5. We validate all (theoretical) analyses on synthetic datasets and demonstrate SWIRL's effectiveness on real-world graphs.

## 2. Graph GCD Preliminaries

We formulate the GGCD problem, design GGCD baselines, and define a loss that faithfully reflects GCD performance.

### 2.1. GGCD Problem Formulation

Given an $n$-node attributed graph $\mathcal{G} = (\mathbf{A}, \mathbf{X})$, where $\mathbf{A} \in \mathcal{A} := \{0,1\}^{n \times n}$ is the binary adjacency matrix and $\mathbf{X} \in \mathbb{R}^{n \times d}$ denotes the node features, $\mathbf{A}_{ij} = 1$ means nodes $i$ and $j$ are linked and the feature vector of node $i$ is given by $\mathbf{x}_i := [\mathbf{X}]_{i:} \in \mathcal{X}$. The node-level dataset $\mathcal{D} = (\mathbf{A}, \{(\mathbf{x}_i, y_i)\}_{i=1}^n)$ is partitioned into two parts: **1)** the labeled subset $\mathcal{D}_L = (\mathbf{A}, \{(\mathbf{x}_i, y_i)\}_{i=1}^{n_{\mathcal{L}}})$, consisting of nodes from $C_o$ known old classes $\mathcal{L} = \{l_1, l_2, \ldots, l_{C_o}\}$; and **2)** the unlabeled subset $\mathcal{D}_U$ of nodes without labels. In GCD, the unlabeled nodes belong to $C$ distinct categories $\mathcal{C} = \mathcal{L} \cup \mathcal{U}$, where $\mathcal{U} = \{u_1, u_2, \ldots, u_{C_n}\}$ denotes the new classes. We adapt the transductive setting (Vaze et al., 2022) to graphs such that all edges and node features are accessible during training but only the labels of nodes from $\mathcal{D}_L$ are known. In testing, the nodes from old classes are expected to be accurately

classified, while those from new classes should be grouped into clusters that represent distinct new classes.

### 2.2. Adapting VGCD Baselines to Graphs

For node-level graph tasks, embedding graph information into the embeddings $\mathbf{z}$ allows subsequent processing to rely only on $\mathbf{z}$. Thus, we can adapt VGCD methods to graphs by replacing the CL module with GCL, leaving the classifier design unchanged. The adaptation details is in Appendix B.

### 2.3. The Surrogate Loss for GCD Performance

The performance of GCD is primarily evaluated by: 1) semi-supervised classification accuracy on old-class data, and 2) clustering accuracy on new-class data. GNN encoders can inject graph structure information into embeddings and thus deduce the embedding dataset $\mathcal{D}^z = \{(\mathbf{z}_i, \mathbf{y}_i)\}_{i=1}^n$. The conditionals of the data from the old class $l$ and the new class $u$ are separately denoted by $p_l = p(\mathbf{z}|y = l) = p(\mathbf{z}|\mathbf{y}^l)$ and $p_u = p(\mathbf{z}|y = u) = p(\mathbf{z}|\mathbf{y}^u)$, where $\mathbf{y}^l, \mathbf{y}^u \in \mathcal{Y}_C = \left\{ \mathbf{y} \in \mathbb{R}_{\geq 0}^C \mid \sum_c^C y_c = 1 \right\}$ are respectively the unique label vectors of class $l \in \mathcal{L}$ and $u \in \mathcal{U}$. The label distribution within old/new/all classes is denoted as $p_{\mathcal{L}}(c)/p_{\mathcal{U}}(c)/p(c)$.

**Old Capability**. For the whole old class data, the population classification loss of (parametric) classifier $h$ reads

$$L_{\mathcal{L}}(h) = \sum_{l \in \mathcal{L}} p_{\mathcal{L}}(l) \mathbb{E}_{\mathbf{z} \sim p(\mathbf{z}|\mathbf{y}^l)} L(h(\mathbf{z}), \mathbf{y}^l), \qquad (1)$$

where $L : \mathcal{Y}_C \times \mathcal{Y}_C \to \mathbb{R}$ measures the discrepancy between two label distributions. During training, only the empirical version $\hat{p}_l$ of $p_l$ is available, leading to the empirical loss $\hat{L}_{\mathcal{L}}(h)$. $L$ can also quantify the similarity between two hypotheses $h$ and $h^*$, given the data from the class set $\mathcal{C}$

$$E_{\mathcal{C}} = \sum_{c \in \mathcal{C}} p(c) \mathbb{E}_{\mathbf{z} \sim p_c} L(h(\mathbf{z}), h^*(\mathbf{z})), \qquad (2)$$

where $p(c)$ is the label distribution of all data from $\mathcal{C}$.

**New Capability**. For the new-class nodes $(\mathbf{z}_i, y_i)$ from class $u$, the predicted labels $\hat{y} = \arg\max_i \mathbf{y}_i$ may not match the true one $u$. Thus, we use a clustering loss instead of a classification loss to distinguish new classes.

$$L_{\mathcal{U}}(h) = \sum_{u \in \mathcal{U}} p_{\mathcal{U}}(u) \mathbb{E}_{p(\mathbf{z}|u)} \mathbb{E}_{p(\mathbf{z}^+|u)} L(h(\mathbf{z}), h(\mathbf{z}^+)) + \alpha F \qquad (3)$$

$$F = \sum_{u \neq u'} p_{\mathcal{U}}(u) p_{\mathcal{U}}(u') \mathbb{E}_{p(\mathbf{z}|u)} \mathbb{E}_{p(\mathbf{z}^-|u')} \left[ -L(h(\mathbf{z}), h(\mathbf{z}^-)) \right],$$

where the first term attracts the intra-class samples, $F$ repels the inter-class ones, and $\alpha > 0$ scales the repulsion force.

Considering both old and new capability leads to the loss

$$L_{te}^G(h) = \beta L_{\mathcal{L}}(h) + L_{\mathcal{U}}(h), \beta > 0. \qquad (4)$$

Optimizing Eq. (4) may lead to a trivial GCD solution: Suppose $C_n = C_o$ and all old-class samples are predicted perfectly, i.e., $L_{\mathcal{L}}(h) = 0$. Without loss of generality, assume that all nodes from the new class $u_1 = C_o + 1$ are all predicted as $l_1 = 1$, those from $u_2 = C_o + 2$ are predicted as $l_2 = 2$, and so on. In this case, $L_{\mathcal{U}}(h) = 0$, leading to $L_{te}^G(h) = 0$. But we cannot distinguish new-class data from old-class data (**Reject Capability**). To resolve this issue, the expected prediction $\bar{h}_{\mathcal{U}} = \mathbb{E}_{p_{\mathcal{U}}(\mathbf{z},y)} h(\mathbf{z})$ across all new-class data should be supported only on $\mathcal{U}$. Thus, we have to minimize the KL-divergence between $\bar{h}_{\mathcal{U}}$ and $p_{\mathcal{U}}(c)$, yielding *the final GCD loss* of classifier $h$.

$$L_{te}(h) = \beta L_{\mathcal{L}}(h) + L_{\mathcal{U}}(h) + \gamma D_{KL}(p_{\mathcal{U}}(c)\|\bar{h}_{\mathcal{U}}), \gamma > 0 \quad (5)$$

# 3. Understand Parametric GCD Classifiers

Using the classifier $h$ and the embedding dataset induced by GNN encoders, we have defined the GCD loss Eq. (5). Addressing Question 1 involves investigating how the the old-and-new relationship influences **Old/New/Reject** capabilities. We reflect this relationship with the old-and-new Wasserstein distance, and analyze its impact on GCD.

## 3.1. Quantify the Old-and-new Relationship

To investigate the relationship between old and new classes using both embedding and label information, we introduce the joint Wasserstein distance in Definition 3.1.

**Definition 3.1.** The Wasserstein distance between two joint probability distributions $p_1 = p_{\mathcal{U}}(\mathbf{z}_1, u)$ and $p_2 = p_{\mathcal{L}}(\mathbf{z}_2, l)$ over the metric space $(\mathcal{Z} \times \mathcal{C}, D^\lambda)$ is given by

$$W^\lambda(p_1, p_2) = \inf_{\pi \in \Pi} \int D^\lambda \left[(\mathbf{z}_1, u), (\mathbf{z}_2, l)\right] d\pi \left[(\mathbf{z}_1, u); (\mathbf{z}_2, l)\right],$$

where $D^\lambda \left[(\mathbf{z}_1, u), (\mathbf{z}_2, l)\right] = \lambda D(\mathbf{z}_1, \mathbf{z}_2) + L(\mathbf{y}^u, \mathbf{y}^l)$ is the cost of transporting $(\mathbf{z}_1, u)$ to $(\mathbf{z}_2, l)$, $\lambda > 0$ weights the embedding distance, and $\Pi(p_1, p_2)$ is the set of couplings over $(\mathcal{Z} \times \mathcal{C})^2$ such that $\int_{\mathcal{Z}} \int_{\mathcal{L}} \pi \left[(\mathbf{z}_1, u); (\mathbf{z}_2, l)\right] d\mathbf{z}_2 d_l = p_1$ and $\int_{\mathcal{Z}} \int_{\mathcal{U}} \pi \left[(\mathbf{z}_1, u); (\mathbf{z}_2, l)\right] d\mathbf{z}_1 d_u = p_2$ .

## 3.2. The Assumptions of Our GCD Theory

We present and discuss the assumptions used in this work.

**Assumption 3.2.** The embedding space is bounded $\|\mathbf{z}\| \leq B_z$ and $h$ is Lipschitz such that $L(h(\mathbf{z}_1), h(\mathbf{z}_2)) \leq S$.

Assumption 3.2 requires that embeddings are bounded and the classifier is Lipschitz. The boundedness can be met by bounding the spectral norms of encoder weights (Tang & Liu, 2023) and normalizing the input node features (Verma & Zhang, 2019). The Lipschitz continuity, which depends on the spectral norms of $h$'s weights (Virmaux & Scaman, 2018), is a common assumption in neural network analysis (Redko et al., 2022; Khromov & Singh, 2023).

**Assumption 3.3.** The loss function $L(\cdot, \cdot) : \mathcal{Y}_C \times \mathcal{Y}_C \to \mathbb{R}_+$ satisfies the triangle inequality and is $r$-Lipschitz ($r > 0$) w.r.t. the first argument. That is

$$|L(\mathbf{y}_1, \mathbf{y}_3) - L(\mathbf{y}_2, \mathbf{y}_3)| \leq rL(\mathbf{y}_1, \mathbf{y}_2). \quad (6)$$

Losses such as MSE, MAE and Jensen–Shannon distance (Englesson & Azizpour, 2021) satisfies Assumption 3.3 but Cross Entropy (CE) $D_{CE}$ does not. Nevertheless, the equivalence of MSE and CE has been demonstrated empirically (Hui & Belkin, 2020) and theoretically (Zhou et al., 2022). Thus, our theory, based on Assumption 3.3, is expected to also hold when $L = D_{CE}$, as further evidenced by our illustrative experiments involving CE (Sec.s 3.4 and 4.3).

**Assumption 3.4.** Given distributions $p_1(\mathbf{z}_1)$ and $p_2(\mathbf{z}_2)$, $h : \mathcal{Z} \to \mathcal{Y}_C$ is said $\phi(\lambda)$-Lipschitz transferable if

$$\Pr_{\pi(p_1, p_2)} \{L(h(\mathbf{z}_1), h(\mathbf{z}_2)) > \lambda D(\mathbf{z}_1, \mathbf{z}_2)\} \leq \phi(\lambda), \lambda > 0 \quad (7)$$

where $\phi : \mathbb{R} \to [0, 1]$, $L$ and $D$ denote the respective metrics in $\mathcal{Y}_C$ and $\mathcal{Z}$, and $\pi(p_1, p_2)$ is the joint distribution of $(\mathbf{z}_1, \mathbf{z}_2)$ coupled by $p_1(\mathbf{z}_1)$ and $p_2(\mathbf{z}_2)$.

Assumption 3.4 ensures that, under different distributions, the classifier's output typically does not change by more than $\lambda$-times the input variation. If this property does not hold, the predictions for new-class nodes will be almost unaffected by the old-class knowledge (embedded in the classifier weights). This contradicts the GCD's motivation of discovering new classes by leveraging old knowledge. Therefore, Assumption 3.4 is essential for the tractability of GCD problems when employing parametric GCD methods.

## 3.3. The GCD Loss Upper Bound Theorem

Under these assumptions, we analyze the influence of $W$ on the GCD loss Eq. (5), leading to Theorem 3.5.

**Theorem 3.5** (The GCD Loss Upper Bound Theory). *Let Assumptions 3.2, 3.3 and 3.4 hold and $h^*$ be the optimal hypothesis that minimizes the GCD loss Eq. (5). Let $n_{\mathcal{L}} = \sum_l n_l$ and $n_{\mathcal{U}} = \sum_u n_u$ be the total numbers of old and new class samples, respectively. And suppose that the dataset is class-balanced. Then there exists, $c'$ and $n_1$, such that for $n_{\mathcal{U}} > n_1$, $n_{\mathcal{L}} > n_1$, and all $\lambda > 0$, $L_{te}(h)$ is bounded as, with probability at least $1 - \delta - \omega$ ($\delta, \omega > 0$),*

$$L_{te}(h) \leq \alpha F + (2 + \beta)\hat{L}_{\mathcal{L}}(h^*) + 2W^{r\lambda}(\hat{p}_{\mathcal{U}}(\mathbf{z}, y), \hat{p}_{\mathcal{L}}(\mathbf{z}, y))$$
$$+ \frac{\gamma}{b} D_{KL}(P_{\mathcal{C}}\|\bar{h}) + 2E_{\mathcal{U}} + E_{\mathcal{L}} + T_G, \quad (8a)$$

$$T_G = (2 + \beta)\sqrt{\frac{S^2}{2n'_{\mathcal{L}}} \ln \frac{2}{\omega}}$$
$$+ 2\left[rS\phi(\lambda) + \sqrt{\frac{2}{c'} \log \frac{2}{\delta}} \left(\frac{1}{\sqrt{n_{\mathcal{U}}}} + \frac{1}{\sqrt{n_{\mathcal{L}}}}\right)\right],$$

where $b = \Pr(y \in \mathcal{U})$, $P_\mathcal{C}$ is the uniform distribution over $\mathcal{C}$, $\bar{h} = \mathbb{E}_{p(\mathbf{z},y)}h(\mathbf{z})$ is the expected prediction over both the old and new data, and $\hat{p_\mathcal{L}}(\mathbf{z}, y)$ and $\hat{p_\mathcal{U}}(\mathbf{z}, y)$ are respectively the empirical old and new joint distributions. $W^{r\lambda}$ is hereafter abbreviated as $W$. $n'_\mathcal{L}$ is the number of old-class training samples while $n_\mathcal{L}/n_\mathcal{U}$ is the total number of old-/new-class samples in the whole embedding dataset $\mathcal{D}^z$.

In the bound Eq. (8a), $\hat{L}_\mathcal{L}(h^*)$ indicates the (old-class) training classification loss of the optimal GCD solution $h^*$. $E_\mathcal{L}/E_\mathcal{U}$ represents the deviation between any $h$ and $h^*$ on the old-/new-class distribution. $\hat{L}_\mathcal{L}(h^*)$ with $E_\mathcal{L}$ corresponds to **Old Capability**. $F$ reflects $h$'s ability to distinguish new classes. $F$ plus $E_\mathcal{U}$ corresponds to **New Capability** under the constraint of old-class data. $D_{KL}(P_\mathcal{C}\|\bar{h})$ corresponds to **Reject Capability**. The old-and-new distribution discrepancy $W^{r\lambda}(\hat{p_\mathcal{U}}(\mathbf{z}, y), \hat{p_\mathcal{L}}(\mathbf{z}, y))$ (abbreviated as $W$ later) can influence **all three Capabilities**, as elaborated later. $T_G$ summarizes all the sample complexity terms.

To provide an intuitive interpretation of Theorem 3.5, we present in Appendix E.2 an example with a total of four classes: two old classes (cats and trucks) and two new classes (lions and sports cars). Fig. 5 illustrates the relationships between these classes and their impact on the Wasserstein distance between old-class and new-class data.

**Implications of Theorem 3.5**

*Remark* 3.6. $W$ in Theorem 3.5 describes the relationship between old and new classes, and should be small to bound the GCD loss. A small $W$ implies that the categories with closer semantic relations (i.e., smaller label distances in $\mathcal{Y}_C$) should also be more proximate in $\mathcal{Z}$, according to the joint-distribution-style Definition 3.1. That is pushing new classes towards those old classes semantically closer to them.

*Remark* 3.7. When considering $W$ in isolation, Remark 3.6 encourages a small $W$. However, a too small value implies that the new-class data entirely collapse into old classes, which degrades Reject, and then New and Old Capabilities. This aligns with Theorem 3.5: when new classes are mixed into old ones, $D_{KL}(P_\mathcal{C}\|\bar{h})$ in Eq. (8a) increases substantially, causing the bound to rise even $W$ decreases.

*Remark* 3.8. Theorem 3.5 justifies Entropy Regularization (ER) (Assran et al., 2022; Wen et al., 2023), which pushes the expected prediction $\bar{h}$ uniformly distributed across $\mathcal{C}$, appears as $D_{KL}(P_\mathcal{C}\|\bar{h})$ in the upper bound Eq. (8a). This term, according to Proposition 3.9, upper bounds the reject term $D_{KL}(p_\mathcal{U}(c)\|\bar{h}_\mathcal{U})$ in the GCD loss Eq. (5).

**Proposition 3.9.** *If $P_\mathcal{U}, P_\mathcal{C} \in \mathcal{Y}_C$ are uniform distributions over $\mathcal{U}$ and $\mathcal{C}$ respectively, and the data distribution is class-balanced, then it follows that*

$$D_{KL}(P_\mathcal{U}\|\bar{h}_\mathcal{U})) \leq \frac{1}{b}D_{KL}(P_\mathcal{C}\|\bar{h}), \tag{9}$$

where $b = \Pr(y \in \mathcal{U}) = \sum_{y \in \mathcal{U}} \int_\mathcal{Z} p(\mathbf{z}, y)d\mathbf{z}$, $\bar{h} = \mathbb{E}_{p(\mathbf{z},y)}h(\mathbf{z})$, and $D_{KL}(P_\mathcal{C}\|\bar{h})$ is minimized when the entropy $H(\bar{h}) = -\sum_{c \in \mathcal{C}} \bar{h}(c) \log \bar{h}(c)$ is maximized.

## 3.4. Illustrative Experiments

To validate the above remarks, particularly the impact of $W$, we conducted a series of experiments. Specifically, we compared the performance of an MLP classifier across multiple embedding datasets with varying $W$ values to investigate how the old-and-new relationship affects the Old/New/Reject Capabilities, thereby addressing Question 1. Since Theorem 3.5 focuses on the classification phase after obtaining embeddings, we use only the CE and ER losses to train the final MLP classifier on the synthetic embedding datasets. This setup allows us to isolate the effects of representation learning and elucidates the embedding properties required for a parametric classifier to achieve strong GCD performance.

**Synthetic Embedding Datasets.** We generate three 2D embedding datasets $\mathcal{D}_1^z$, $\mathcal{D}_2^z$, and $\mathcal{D}_3^z$, as shown in Fig. 1a-1c. Each dataset contains eight classes/clusters, with four inner clusters representing new classes (centroids labeled as ✖, ✖, ✖, ✖) and four outer clusters representing old classes (centroids labeled as ★, ★, ★, ★). The dataset generation models and hyperparameters are provided in Appendix G.3. From $\mathcal{D}_1^z$ to $\mathcal{D}_3^z$, the relative distances between old and new class centroids progressively decrease (i.e., $W_1 > W_2 > W_3$), reflecting different old-and-new relationships.

**Verify Remark 3.6: Large $W$ Undermines GCD.** In $\mathcal{D}_1^z$ (Fig. 1a), new classes are mixed and are closer to each other than to old classes, resulting in a large $W_1$. By increasing the distances between new classes while reducing the relative distance between new and old classes, we obtain $\mathcal{D}_2^z$ (Fig. 1b), where $W_2 < W_1$. As stated in Remark 3.6, a decrease in $W$ implies that the distance between new classes and their adjacent old classes becomes smaller, which allows more of the distinguishing features between old classes to transfer to the new classes. From $\mathcal{D}_1^z$ to $\mathcal{D}_2^z$, ✖ moves closer to ★, ✖ to ★, ✖ to ★ , and ✖ to ★. As a result, the New Capability in $\mathcal{D}_2^z$ outperforms that in $\mathcal{D}_1^z$: as shown in Fig. 1e, Reject ACC[1] remains nearly unchanged, while New RACC improves by about 7 points.

**Verify Remark 3.7: Too Small $W$ Undermines GCD.** By further reducing the relative distance between new and old classes from $\mathcal{D}_2^z$, new-class data move even closer to the old classes, resulting in $\mathcal{D}_3^z$ (Fig. 1c) with a smaller Wasserstein distance $W_3 < W_2$. Although the new and old classes are now closer, the overlap between the data of new and old

---

[1]All evaluation metrics are formally defined and discussed in Appendix G.1, where we refine the conventional GCD metrics towards more faithful GCD evaluation.

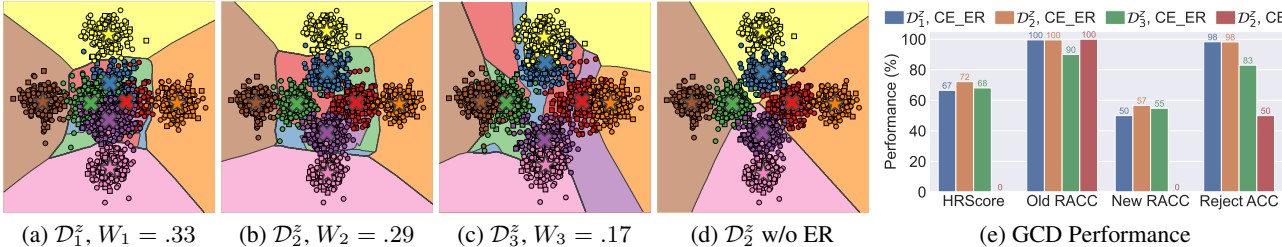

(a) $\mathcal{D}_1^z$, $W_1 = .33$    (b) $\mathcal{D}_2^z$, $W_2 = .29$    (c) $\mathcal{D}_3^z$, $W_3 = .17$    (d) $\mathcal{D}_2^z$ w/o ER    (e) GCD Performance

Figure 1: (a-c) The datasets and the decision boundaries (marked by background colors) of MLPs trained with the CE+ER criterion. (d) The decision boundary of MLP trained with CE and without ER on $\mathcal{D}_2^z$ . (e) The GCD performance.

classes counteracts the benefits of the reduced $W$, and the performance does not improve as it did in the transition from $\mathcal{D}_1^z$ to $\mathcal{D}_2^z$. This overlap leads to more misclassifications, where new-class samples are predicted as old classes and vice versa (Remark 3.7), which is reflected in a significant drop in Reject ACC. As shown in Fig. 1e, Reject ACC in $\mathcal{D}_3^z$ drops significantly compared to $\mathcal{D}_2^z$, and both New RACC and Old RACC also decline due to the confusion between new and old classes.

**Verify Remark 3.8: Entropy Regularization Matters.** In $\mathcal{D}_2^z$ , when ER is removed, as noted in Remark 3.8, all new-class data are predicted as old classes as shown by the decision boundary in Fig. 1d, resulting in a complete loss of New Capability in Fig. 1e.

### 3.5. Main Takeaway: Two Lessons

Theorem 3.5 provides an upper bound for the GCD loss. Analyzing this bound implies that, for effective GCD performance, the embeddings induced by the GNN encoder should satisfy **Lesson 1**: *a sufficiently small old-and-new Wasserstein distance $W$ discussed in Remark 3.6*; **Lesson 2**: *limited overlap between new and old classes (i.e., a not too small $W$) discussed in Remarks 3.7 and 3.8*.

## 4. GCL Loses Category Semantic Relations

The small $W$ in **Lesson 1** requires that for a pair of old and new classes $(u, l)$ that are closer than another pair $(u, l')$ in the category semantic space $\mathcal{Y}_C$, the embeddings of $u$ and $l$ should be much closer than those of $(u, l')$. That is, the embedding distances between classes, i.e., the Category Embedding Relations, should align with the Category Semantic Relations in $\mathcal{Y}_C$. **Lesson 2** implies that there should be a certain distance kept between the embeddings of old and new classes. Both lessons impose constraints on the category embedding relations. However, we find that the GCL methods in the parametric GCD baselines (e.g., SimGCD) inherently fail to satisfy these constraints, as revealed by the PMRF interpretation of GCL (Sec. 4.1) and the smoothing effect of GCN encoder (Sec. 4.2).

### 4.1. GCL Framework Violates Lesson 1

**PMRF View of GCL Framework** Inspired by Tan et al. (2023), we encode the similarity relations among all raw and possibly augmented nodes using $\mathbf{B} \in [0, 1]^{N \times N}$, where $B_{ij}$ is the chance of nodes $i$ and $j$ being sampled as a positive pair under data augmentation. As a form of human prior in GCL, the augmentation should align $\mathbf{B}$ with human cognition of node relations and guide the model to approximate $\mathbf{B}$. In the raw space $\mathcal{X}$, approximating $\mathbf{B}$ through simple similarity computations is challenging. However, a good encoder $f$ can produce an embedding space $\mathcal{Z}$, where $\mathbf{B}$ can be effectively approximated by *the node embedding relation matrix* $\mathbf{K} \in \mathbb{R}_{\geq 0}^{N \times N}$, where $K_{ij} = k(\mathbf{z}_i, \mathbf{z}_j)$ is induced by a simple kernel function $k(\cdot, \cdot)$ in the embedding space $\mathcal{Z}$.

We follow the coupling framework from Assel et al. (2022), and treat the embeddings as observations of a Pairwise Markov Random Field (PMRF) defined on a graph $\mathbf{W} \in \{0, 1\}^{N \times N}$, with each category corresponding to a graph connected component (CC). According to spectral graph theory (Chung, 1997), the Laplacian $\mathbf{L}$ of this graph has a null-space $\ker \mathbf{L}$ of rank $C$, and the orthogonal complement $(\ker \mathbf{L})^\perp$. A signal $\mathbf{Z} \in \mathbb{R}^{N \times d}$ can be decomposed as $\mathbf{Z} = \mathbf{Z}_0 + \mathbf{Z}_1$, where $\mathbf{Z}_0$ and $\mathbf{Z}_1$ are the projections onto the subspaces $\mathcal{S}_0 = (\ker \mathbf{L}) \otimes \mathbb{R}^d$ and $\mathcal{S}_1 = (\ker \mathbf{L})^\perp \otimes \mathbb{R}^d$, respectively. For a node $i$ in class $c$, $\mathbf{Z}_{0,i}$ is the mean embedding of all nodes in class $c$, enabling $\mathbf{Z}_0$ to model the global category positions, while $\mathbf{Z}_1$ captures the relative positions of nodes within their respective CCs (aka clusters).

The conditional measures on the orthogonal subspaces $\mathcal{S}_0$ and $\mathcal{S}_1$, denoted $p^\varepsilon(\mathbf{Z}_0|\mathbf{W})$ (parameterized by $\varepsilon > 0$) and $p_k(\mathbf{Z}_1|\mathbf{W})$ (based on kernel $k$), are orthogonal. Their product defines the joint measure $p(\mathbf{Z}|\mathbf{W})$ on $S = \mathcal{S}_0 \oplus \mathcal{S}_1$. Introducing a graph prior $p(\mathbf{W}; \pi)$ parameterized by $\pi$, the posterior of subgraph conditional on *node embeddings*, reads $p(\mathbf{W}|\mathbf{Z}) \propto p(\mathbf{W}; \pi)p(\mathbf{Z}|\mathbf{W})$. Minimizing InfoNCE loss amounts to minimizing the cross-entropy $D_{CE}(p(\mathbf{W}_X; \mathbf{B}) \| p(\mathbf{W}|\mathbf{Z}))$. Here, $p(\mathbf{W}_X; \mathbf{B}) \propto \Omega_D(\mathbf{W}_X) \prod_{(i,j) \in [N]^2} B_{ij}^{W_{X,ij}}$ is the distribution of subgraphs sampled from $\mathbf{B}$, where each subgraph $\mathbf{W}_X$ is

sampled with one random augmentation step. $\Omega_D(\mathbf{W}) \triangleq \prod_i \mathbb{I}(W_{i+} = 1)$ filters out the subgraphs where any node $i$ has more than one outgoing edges, aligning with the single-positive-sample manner in GCL methods such as GRACE.

**Theorem 4.1.** *Minimizing InfoNCE loss is equivalent to minimizing the cross entropy $D_{CE}(p(\mathbf{W}_X; \mathbf{B}) \| p(\mathbf{W}|\mathbf{Z}))$ between the subgraph distribution $p(\mathbf{W}_X; \mathbf{B})$, which depends on data augmentation, and the subgraph posterior $p(\mathbf{W}|\mathbf{Z})$ in the PMRF.*

Apart from InfoNCE loss, SupCon loss is also common. It utilizes augmentations of known same-class samples as extra positive samples. If each node $i$ has $m_i$ known same-class nodes; then, one augmentation step can yield a subgraph satisfying $\Omega_F(\mathbf{W}) \triangleq \prod_i \mathbb{I}(W_{i+} \leq m_i)$. Since augmentations of different nodes are independent, this process is equivalent to independently sampling $m_i$ times subgraphs satisfying $\Omega_D(\mathbf{W})$. From this, we reach Corollary 4.2.

**Corollary 4.2.** *Minimizing SupCon loss is equivalent to minimizing the cross-entropy between subgraph distributions stated in Theorem 4.1 execept the subgraphs here are contrained by $\Omega_F(\mathbf{W})$ instead of $\Omega_D(\mathbf{W})$.*

**The GCL Framework Loses Category Semantic Relation** The above presents a more refined treatment of the embedding space than Tan et al. (2023), exposing a critical limitation not achievable using existing contrastive learning theorems (Huang et al., 2022; Hu et al., 2022; Tan et al., 2023; Parulekar et al., 2023; Waida et al., 2023; Ge et al., 2024). Specifically, *the embedding relations (i.e., distances) between categories are random and thus not aligned with the (either known or unknown) category semantic relations in $\mathcal{Y}_C$, violating **Lesson 1**. The point lies in the degeneracy of the conditional distribution on the PMRF constructed in the embedding space: the variance of the distribution controlling the global category positions $\mathbf{Z}_0$ becomes infinite under the equivalences stated in Theorem 4.1 and Corollary 4.2. Theorem 4.3 summaries this uncontrollability of Category Embedding Relations in GCL.

**Theorem 4.3.** *The InfoNCE or SupCon loss optimization is equivalent to minimizing $D_{CE}(p(\mathbf{W}_X; \mathbf{B}) \| p(\mathbf{W}|\mathbf{Z}))$ when the conditional distribution of category centers $p^\varepsilon(\mathbf{Z}_0|\mathbf{W})$ diffuse uninformatively.*

### 4.2. GCL Encoder Violates Two Lessons

Besides the GCL framework, GCNs (Kipf & Welling, 2017), widely used as GCL encoders, have the smoothing effect (Rusch et al., 2023) that can lead to undesired category embedding relations. The **first** issue occurs when a new class $u$ is connected to other neighboring new classes $u'$, and local smoothness pulls them closer together, pushing them relatively further from old classes. This results in an increase in the old-and-new Wasserstein distance $W$,

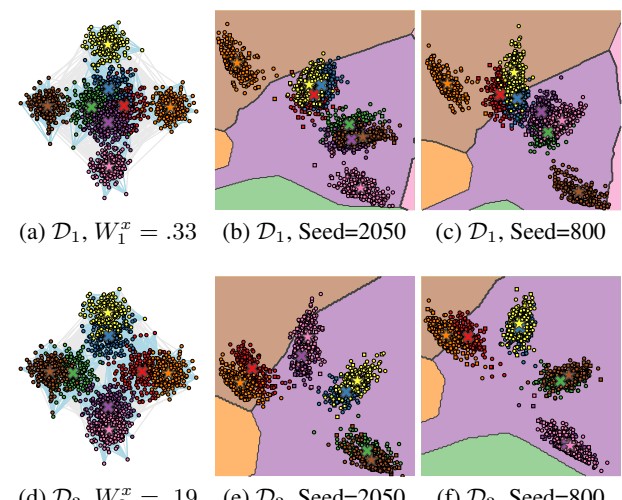

(a) $\mathcal{D}_1$, $W_1^x = .33$    (b) $\mathcal{D}_1$, Seed=2050    (c) $\mathcal{D}_1$, Seed=800

(d) $\mathcal{D}_3$, $W_3^x = .19$    (e) $\mathcal{D}_3$, Seed=2050    (f) $\mathcal{D}_3$, Seed=800

Figure 2: (a,d) The CSBM graphs; (b,c,e,f) the embedding space learned by GCL with the InfoNCE+SupCon criterion.

violating **Lesson 1**. The **second** issue arises when a new class $u$ is considerably linked to an adjacent old class $l$, causing the node embeddings of $u$ to collapse into those of $l$ due to local smoothness. This violates **Lesson 2**.

### 4.3. Illustrative Experiments

**Synthetic CSBM Graphs** (Deshpande et al., 2018) are widely used to evaluate graph learning methods. We generate two CSBM graphs, $\mathcal{D}_1$ and $\mathcal{D}_3$, by first generating node features and labels like $\mathcal{D}_1^z$ and $\mathcal{D}_3^z$ (in Sec. 3.4), and then constructing graph structures with intra-class and inter-class connection probabilities. See Appendix G.4 for details. The main challenges in $\mathcal{D}_1$ and $\mathcal{D}_3$ are, respectively, distinguishing: 1) new classes $u$ and $u'$; 2) old class $l$ and new class $u$.

**Verify Theorem 4.3**. Figs. 2a and 2d illustrate $\mathcal{D}_1$ and $\mathcal{D}_3$. Figs. 2b and 2e show the embedding spaces learned with GRACE, using InfoNCE and SupCon losses, while excluding other components of SimGCD. To showcase the randomness of category embedding relations/distances stated in Theorem 4.3, we run the experiments with different random seeds, resulting in Figs. 2c and 2f. Comparing Figs. 2b and 2c for $\mathcal{D}_1$, ★ and ★ have roughly swapped global positions, while the relative positions of ★, ✖, and ✖ have also changed markedly. Specifically, in Fig. 2c, ★ is more distant from ✖ and ✖ than in Fig. 2b. For $\mathcal{D}_3$, from Fig. 2e to 2f, the original clockwise arrangement of (★, ✖) → (★, ✖) → (★, ✖) → (★, ✖) is disrupted. The distances $D(★, ✖)$ and $D(★, ✖)$, get very small. Furthermore, we find that triangles formed by the embedding centers of any three classes exhibit substantial deformation in both $\mathcal{D}_1$ and $\mathcal{D}_3$ when the seeds are changed.

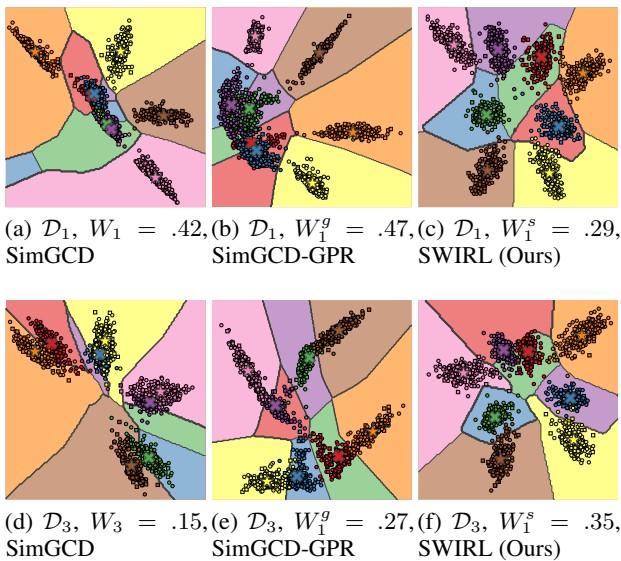

(a) $\mathcal{D}_1$, $W_1 = .42$, SimGCD
(b) $\mathcal{D}_1$, $W_1^g = .47$, SimGCD-GPR
(c) $\mathcal{D}_1$, $W_1^s = .29$, SWIRL (Ours)

(d) $\mathcal{D}_3$, $W_3 = .15$, SimGCD
(e) $\mathcal{D}_3$, $W_1^g = .27$, SimGCD-GPR
(f) $\mathcal{D}_3$, $W_1^s = .35$, SWIRL (Ours)

Figure 3: The embedding space learned by different GCD methods. Backgournd colors mark the decision boundaries.

**Verify the impact of GCN encoder on GCD.** Sec. 4.2 discusses the impact of the GCN encoder's smoothing effect on GCD. To reveal this effect, we replace GCN with GPR (Chien et al., 2021) that can escape from local smoothness, denoted as SimGCD-GPR. In $\mathcal{D}_1$, new classes are highly mixed, with an even greater degree of mixing observed in the SimGCD representation space (Fig. 3a). After replacing the encoder with GPR, the reduction in local smoothness results in increased embedding distances between the new classes (Fig. 3b). In $\mathcal{D}_3$, the mixing of old and new classes prevents SimGCD from distinguishing ⭐ and ✖, ★ and ✖, ☆ and ✖, and ★ and ✖ (Fig. 3d). This issue is mitigated by SimGCD-GPR that uses non-smooth GPR (Fig. 3e).

## 5. Proposed SWIRL for Parametric GCD

Sec. 3 presents two lessons outlining the conditions for good GCD performance that embeddings should satisfy. Sec. 4 reveals that the GCL module in the adapted SimGCD struggles to meet these conditions about *category embedding relations*. In this section, we propose Semantic-aWare dIlation contRastive Learning (SWIRL), a novel GCL method for parametric GGCD, following **Lessons 1** and **2**.

**The Category Relation Speculation via Prototypes.** To adhere to **Lessons 1** and **2** regarding category relations, estimating these relations without relying on the labels of new-class nodes is essential. To achieve this, we utilize SS-KM on the node embeddings averaged over two augmentation views to obtain $K$ prototypes $\mathscr{P} = \{\mathbf{s}^1, \ldots, \mathbf{s}^K\}$ and the node-to-prototype assignments. Among these, the first $C_o$ prototypes $\mathcal{O} = \{\mathbf{s}^1, \ldots, \mathbf{s}^{C_o}\}$ correspond to the

clusters of $C_o$ old classes, while the remaining ones represent unknown clusters. Typically, we set $K > C_o$ to model the embedding space in a fine-grained manner. For a node $i$, it is assigned to prototype $\mathbf{s}(i)$, and the prototype index is $\mathrm{id}[\mathbf{s}(i)] \in [1 : K]$. The prototype relations are then used as a surrogate for category relations.

**Representation Learning with Category-Semantic-Aware Dilation.** SWIRL employs the SWIRL loss, an instance-to-prototype contrastive loss that leverages prototype relations. It follows **Lessons 1** and **2** by controlling the repulsion force (and consequently the distances) between classes. Unlike InfoNCE or SupCon, which uniformly repel all negative samples, SWIRL applies differentiated repulsion. Specifically, we set six levels of repulsion force, denoted as $t_6 < t_5 < t_4 < t_3 < t_2 < t_1$. **On Lesson 2**: To prevent mixing between adjacent categories, for a node $i$ assigned to $\mathbf{s}(i) \in \mathcal{O}$, the supervision signals from old classes make these assignments relatively reliable. Therefore, we apply **(i)** the smallest repulsion force $t_6$ between $i$ and $\mathbf{s}(i)$, and **(ii)** the largest force $t_1$ between $i$ and all other old prototypes. For a node $j$ assigned to $\mathbf{s}(j) \notin \mathcal{O}$, since these assignments are less reliable than those for old prototypes, we apply **(iii)** a moderately low repulsion force $t_5$ between $j$ and $\mathbf{s}(j)$, and **(iv)** a moderately high repulsion force $t_2$ between $j$ and all other new prototypes. **On Lesson 1**: To prevent large Wasserstein distances between old and new classes, the separation between these classes should remain moderate. Therefore, we apply **(v)** a low repulsion force $t_4$ to each pair of a new-class node and an old prototype. Similarly, **(vi)** we apply a moderately strong repulsion force $t_3$ between each old-class node and a new prototype. Finally, the loss for an (either raw or augmented) node $i$ is

$$L_{SW}(i) = -\log \frac{r(i, \mathrm{id}[\mathbf{s}(i)])e^{-D(\mathbf{z}_i, \mathbf{s}(i))}}{\sum_k^K r(i, k)e^{-D(\mathbf{z}_i, \mathbf{s}^k)}}, \quad (10)$$

where $r(i, k) : [1 : 2n] \times [1 : K] \to [t_1 : t_6]$ chooses the force level according to the above designs.

**The overall loss of SWIRL.** The core of SWIRL lies in aligning the category relations in the learned embedding space, as much as possible, with the Category Embedding Relations advocated by **Lessons 1** and **2**. After obtaining the clusters and the node-to-prototype assignments (i.e., pseudo-labels) through SS-KM, we push away different clusters with varying repulsion forces based on these pseudo-labels and their confidence levels that depend on how much old class prototypes $\mathcal{O} = \{\mathbf{s}^1, \ldots, \mathbf{s}^{C_o}\}$ are involved in. The varying repulsion forces are implemented by, as explained in Sec. C.2, applying a weight $r(i, k)$ to the scores of negative (sample, prototype) pairs. Specifically, we define six levels of repulsion forces $0 < t_6 < t_5 < t_4 < t_3 < t_2 < t_1$ for

the SWIRL loss.

$$L_{SW} = -\sum_{i=1}^{2n} \log \frac{r(i, \mathrm{id}[\mathbf{s}(i)])e^{-D(\mathbf{z}_i, \mathbf{s}(i))}}{\sum_k^K r(i,k)e^{-D(\mathbf{z}_i, \mathbf{s}^k)}} \quad (11)$$

$$r(i,k) = \begin{cases} \exp(t_6 sD(\mathbf{z}_i, \mathbf{s}^k)) & , \mathbf{s}(i) = \mathbf{s}^k \in \mathscr{O} \\ \exp(t_1 sD(\mathbf{z}_i, \mathbf{s}^k)) & , \mathbf{s}(i) \in \mathscr{O} \neq \mathbf{s}^k \in \mathscr{O} \\ \exp(t_3 sD(\mathbf{z}_i, \mathbf{s}^k)) & , \mathbf{s}(i) \in \mathscr{O} \neq \mathbf{s}^k \notin \mathscr{O} \\ \exp(t_4 sD(\mathbf{z}_i, \mathbf{s}^k)) & , \mathbf{s}(i) \notin \mathscr{O} \neq \mathbf{s}^k \in \mathscr{O} \\ \exp(t_5 sD(\mathbf{z}_i, \mathbf{s}^k)) & , \mathbf{s}(i) \notin \mathscr{O} = \mathbf{s}^k \notin \mathscr{O} \\ \exp(t_2 sD(\mathbf{z}_i, \mathbf{s}^k)) & , \mathbf{s}(i) \notin \mathscr{O} \neq \mathbf{s}^k \notin \mathscr{O} \end{cases}$$

where $s > 0$ controls the scale of repulsion force, $\mathrm{id}[\mathbf{s}(i)]$ is the prototype index of node $i$'s prototype $\mathbf{s}(i)$. Since $L_{SW}$ primarily emphasizes the global structure in the embedding space and may neglect the preservation of local structures, we adopt a weighted combination of the InfoNCE loss $L_{NCE}$ and the SWIRL loss as the total representation learning objective. Furthermore, following SimGCD, we incorporate the Entropy Regularization (ER) loss on all nodes and apply the Cross Entropy (CE) loss on those labeled (old-class) nodes. The overall training loss for SWIRL is

$$\mathcal{L}_{SW} = (1-\alpha_2)(L_{NCE} + \beta_1 L_{SW} + \alpha_1 L_{ER}) + \alpha_2 L_{CE},$$

where $\alpha_1, \alpha_2 > 0$ are the weights shared with SimGCD, and $\beta_1 > 0$ is an additional hyperparameter for SWIRL. The impact of varying $\beta_1$ is analyzed in Appendix G.7.

**Complexity analysis**. The complete training procedure of SWIRL is summarized in Alg. 1. The SS-KM algorithm, which is executed every $t$ epochs, has a space complexity of $O(nd + Kd)$ and a time complexity of $O(nKId)$, where $I$ is the number of SS-KM iterations and $n$, $K$, and $d$ represent the number of samples, prototypes, and embedding dimension, respectively. When amortized over all epochs, its per-epoch cost reduces to $O(nKId/t)$. Meanwhile, $L_{SW}$ contributes a time complexity of $O(nKd)$ per epoch, dominated by the computation of similarities between samples and prototypes, while requiring $O(nK)$ space for storing similarity matrices. Let $T$ be the training epochs. Overall, SWIRL results in a total time complexity of $O(TnKd)$, assuming a small constant $I/t$, and an overall space complexity of $O(nK + nd)$, which is significantly more efficient than the $O(n^2 d)$ complexity of $L_{NCE}$, making the extra overhead from $L_{SW}$ negligible.

### 5.1. Illustrative Experiments: SWIRL excels SimCGD

To evaluate whether SWIRL can establish desired category embedding relations, we conducted experiments on the CSBM graphs $\mathcal{D}_1$ and $\mathcal{D}_3$ introduced in Sec. 4.3. As shown in Figs 3c and 3f, SWIRL learns more discriminative embeddings. On $\mathcal{D}_1$, the old-and-new Wasserstein distance $W_1^s$ of SWIRL's representation space is substantially smaller than

Table 1: The performance on CSBM graphs $\mathcal{D}_1$ and $\mathcal{D}_3$. SWIRL denotes SWIRL-GCN.

| $\mathcal{D}_1$ | **HRScore** | Old RACC | New RACC | Reject ACC |
|---|---|---|---|---|
| SimGCD | 68.22 | 99.58 | 51.88 | **99.79** |
| SimGCD-GPR | 77.18 | 99.79 | 62.92 | 99.48 |
| SWIRL-GPR | 87.72 | **100.0** | 78.13 | 98.54 |
| SWIRL | **96.24** | 97.29 | **95.21** | 98.54 |
| $\mathcal{D}_3$ | **HRScore** | Old RACC | New RACC | Reject ACC |
| SimGCD | 64.76 | **98.96** | 48.12 | 73.75 |
| SimGCD-GPR | 88.77 | 98.12 | 81.04 | 90.52 |
| SWIRL-GPR | 82.39 | 96.88 | 71.67 | 89.58 |
| SWIRL | **94.03** | 95.83 | **92.29** | **97.81** |

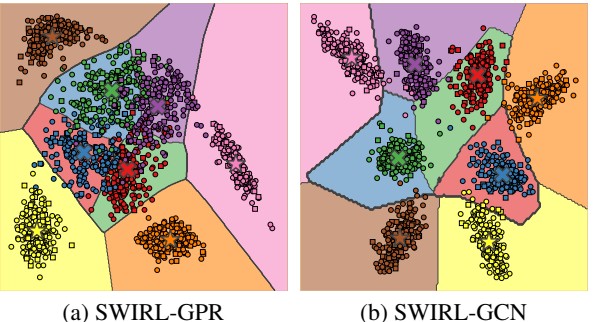

(a) SWIRL-GPR  (b) SWIRL-GCN

Figure 4: The representation spaces learned with different GNN encoders on $\mathcal{D}_3$. (a) Low intra-cluster cohesion blurs the inter-cluster boundaries. (b) High intra-cluster cohesion facilitates inter-cluster separability.

that in SimGCD, satisfying **Lesson 1**. On $\mathcal{D}_3$, SWIRL effectively separates all adjacent old-and-new category pairs, resulting in a larger old-and-new Wasserstein distance $W_3^s$ than SimGCD, satisfying **Lesson 2**. And as shown in Table 1, SWIRL-GCN outperforms the others on both datasets.

**SWIRL with GPR encoders**. Unlike SWIRL-GCN, which consistently benefits CGN across datasets, SWIRL-GPR outperforms SimGCD-GPR on $\mathcal{D}_1$ but underperforms on $\mathcal{D}_3$. To investigate this discrepancy, we visualize the representation space of SWIRL-GPR on $\mathcal{D}_3$ in Fig. 4. It shows significantly weaker intra-cluster cohesion compared to SWIRL-GCN. Low intra-cluster cohesion hampers the formation of clear inter-cluster separations and decision boundaries, leading to GCD failure. We attribute this issue to both the GPR encoder and SWIRL's emphasis on information from distant nodes/spaces beyond local neighborhoods, which neglects local structural cues and weakens cluster compactness. Moreover, prior work (Chen et al., 2023) has noted that GPRs are inherently difficult to train with current GCL methods. Therefore, we recommend using GCN instead of GPR with SWIRL.

Table 2: Mean and standard deviation of performance (%) on real-world graph datasets. The best metric is in bold.

| Datasets | Metrics | GCN | SS-KM | UNO+ | VanillaGCD | SimGCD | SWIRL (Ours) |
|---|---|---|---|---|---|---|---|
| Cora | **HRScore** | 6.88±10.76 | 3.65±0.0 | 30.47±17.04 | 62.40±6.59 | 53.14±20.3 | **64.53±4.63** |
| | Old RACC | **92.22±0.69** | 54.79±0.0 | 91.78±0.87 | 84.55±4.85 | 77.10±7.48 | 85.70±2.47 |
| | New RACC | 3.87±6.18 | 1.89±0.0 | 19.34±12.0 | 49.80±7.63 | 44.41±19.82 | **52.03±6.02** |
| | Reject ACC | 40.56±6.93 | 37.44±0.0 | 56.76±12.77 | 84.10±1.26 | 77.42±18.33 | **88.16±1.97** |
| Citeseer | **HRScore** | 0.00±0.00 | 39.90±0.0 | 41.48±10.98 | **65.35±6.10** | 46.36±3.72 | 55.62±3.03 |
| | Old RACC | **69.91±1.72** | 48.41±0.0 | 51.50±7.61 | 65.23±1.74 | 59.91±2.83 | 55.95±3.88 |
| | New RACC | 0.00±0.00 | 33.93±0.0 | 37.61±13.76 | **66.36±11.20** | 38.04±4.97 | 55.32±2.35 |
| | Reject ACC | 44.00±0.00 | 63.20±0.0 | 74.32±15.4 | 83.16±4.94 | 78.78±2.39 | **84.62±0.61** |
| Wiki | **HRScore** | 15.39±3.99 | 25.11±0.0 | 50.97±6.38 | 51.32±1.27 | 46.51±5.10 | **52.93±2.72** |
| | Old RACC | 74.47±0.71 | 44.35±0.0 | 64.88±5.04 | 56.32±1.87 | **75.05±0.42** | 57.52±2.91 |
| | New RACC | 8.64±2.43 | 17.51±0.0 | 43.11±9.96 | 47.20±2.13 | 33.91±5.55 | **49.03±2.87** |
| | Reject ACC | 65.31±2.85 | 58.00±0.0 | **77.51±3.41** | 72.79±2.61 | 74.98±2.34 | 75.19±1.51 |
| A-Photo | **HRScore** | 55.83±1.32 | 50.37±0.0 | 76.72±5.12 | 66.16±3.15 | 67.13±8.60 | **77.62±4.27** |
| | Old RACC | **91.15±1.11** | 65.97±0.0 | 85.54±2.57 | 88.36±3.37 | 89.55±1.99 | 90.08±3.16 |
| | New RACC | 40.24±1.22 | 40.74±0.0 | **69.98±8.57** | 52.93±3.31 | 54.58±11.48 | 68.33±5.66 |
| | Reject ACC | 82.78±1.00 | 71.29±0.0 | 91.72±2.77 | **94.98±1.56** | 91.51±6.55 | 93.80±0.33 |
| A-Computers | **HRScore** | 32.94±15.56 | 27.43±0.0 | 49.68±7.66 | 56.88±3.79 | 32.38±2.81 | **61.46±4.54** |
| | Old RACC | 77.61±0.99 | 44.14±0.0 | 73.87±4.16 | 70.82±3.18 | 29.71±6.78 | **85.34±8.79** |
| | New RACC | 21.99±11.03 | 19.92±0.0 | 38.11±9.41 | 47.56±3.97 | 38.71±9.04 | **48.32±4.48** |
| | Reject ACC | 78.47±5.21 | 60.89±0.0 | 81.46±3.44 | 78.83±3.30 | 52.56±4.02 | **88.38±2.24** |

# 6. Experiments on Real-world Graph Datasets

We now evaluate all GGCD methods on real-world graphs.

## 6.1. Experimental Setup

**Datasets and Split.** We created node-level GGCD datasets based on five existing datasets: Cora, Citeseer, Wiki, A-Computers, and A-Photo. For Cora and Citeseer, we used the public splits, while for the other three datasets, The entire node set is stratified into train, validation, and test subsets in a 2:2:6 ratio. In each dataset, the first $C//2$ classes are designated as old classes, while the remaining classes are considered new classes. The old-class nodes in the training set form $\mathcal{D}_L$, while all other nodes constitute $\mathcal{D}_U$. During testing, predictions are made for all nodes in $\mathcal{D}_U$, with performance evaluated specifically on nodes that also belong to the test set. See Appendix G for the evaluation protocol and implementation details.

## 6.2. The GGCD Performance

Our primary goal is to develop a theoretical understanding of Parametric GCD. Beyond the theoretical contributions, our simple yet effective method, SWIRL, shows strong empirical performance. Our comparisons include several GGCD methods adapted from VGCD, such as Vanilla GCD, SimGCD, UNO+, SS-KM, and GCN. Among these, SS-KM directly predicts node features, while GCN incorporates an additional ER loss alongside the cross-entropy loss. As shown in Table 2, SWIRL achieves a significantly higher overall performance, measured by HRScore, compared to its competitors. Notably, on the largest dataset, A-computers, SWIRL outperforms all baselines across all metrics. On Citeseer, while SWIRL still lags behind Vanilla GCD, it substantially outperforms other parametric GCD methods, i.e., SimGCD and UNO+. These results highlight SWIRL's effectiveness and the importance of understanding parametric GCD performance for developing better methods.

# 7. Conclusions

We introduce the Generalized Graph Category Discovery (GGCD) task and develop several baselines. We provide the first theoretical answer to the core GCD question: "*When and how do old classes help (parametric) generalized category discovery on graphs?*" Using the Wasserstein distance $W$ between the joint (embedding, label) distributions of old- and new-class nodes, we quantify their relationship and its impact on GCD. Theorem 3.5 formalizes this, offering a provable GCD loss upper bound dependent on $W$ and identifying the necessary category embedding relation conditions for low GCD loss. We proceed to analyze GCL methods that employ InfoNCE-style losses, which are commonly used in parametric GCD baselines, through the lens of PMRF and the smoothing effect of GCNs. Our analysis reveals that such GCL methods often fail to meet the conditions for low GCD loss. To address this, we propose SWIRL, a new GCL method that controls the category embedding relations. Experiments on synthetic and real graph datasets validate our theoretical analysis and confirm SWIRL's effectiveness.

## Acknowledgments

The research is supported by the National Key Research and Development Program of China (2023YFB2703700) and the National Natural Science Foundation of China (62176269).

## Impact Statement

This paper presents work whose goal is to advance the field of Machine Learning. There are many potential societal consequences of our work, none which we feel must be specifically highlighted here

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

# Appendix

## Table of Contents

## A. Related Work

### A.1. Visual Generalized Category Discovery (VGCD)

The Generalized Category Discovery (GCD) task was first introduced by Vaze et al. (2022) for visual image data (Krizhevsky, 2009; Deng et al., 2009; Tan et al., 2019; Krause et al., 2013; Wah et al., 2011). Concurrently, Cao et al. (2021) proposed the same setting under the name open-world semi-supervised learning. The GCD problem is a composite task that involves (i) semi-supervised classification of known-class samples and (ii) clustering of novel-class samples. Thus, it is closely related

to both semi-supervised learning (Yang et al., 2023) and (deep) clustering (Lloyd, 1982; Xie et al., 2016; Lu et al., 2022; Ren et al., 2025).

Vaze et al. (2022) developed a simple yet robust Visual GCD (VGCD) method, Vanilla GCD, which first learns image representations through semi-supervised contrastive learning using SimCLR (Chen et al., 2020) and SupCon (Khosla et al., 2020), followed by clustering these representations using a Semi-Supervised K-Means (SS-KM) classifier. SS-KM enforces old-class samples with known labels to be assigned to their corresponding old-class clusters, enabling both classification for old classes and cluster-based discovery of novel classes. Subsequent research has evolved along two main directions. The first direction aims to enhance the representation learning component while maintaining a non-parametric classifier. The second direction investigates replacing the non-parametric SS-KM, which is inefficient for mini-batch inference, with a parametric classifier, such as a prototype-based classifier.

**Non-parametric VGCD Methods.** In GCD, contrastive learning is employed for representation learning, with the key challenges being the construction of positive and negative sample pairs and the selection of an appropriate pretext task. Subsequent methods primarily focus on improving these aspects.

**DCCL** (Pu et al., 2023) constructs a graph adjacency matrix based on sample embeddings and applies InfoMap (Rosvall & Bergstrom, 2008) for clustering on the graph to obtain conceptual labels. It then computes cluster centroids for each concept and performs contrastive learning at both the instance-to-concept and instance-to-instance levels. After representation learning, SS-KM is used to obtain the final results. **PromptCAL** (Zhang et al., 2023b) builds K-Nearest-Neighbors (KNN) graphs in the embedding space and determines edge weights by counting the number of shared KNN neighbors between two nodes. It then applies graph diffusion to capture long-range sample relationships. Using predefined edge weight thresholds and labeled data, it constructs a binary affinity graph $G$, where endpoints of edges in $G$ are treated as positive pairs, while all other sample pairs serve as negative pairs for additional contrastive loss. SS-KM is used to generate the final results after representation learning. To address the GCD problem without prior knowledge of the number of categories, Zhao et al. (2023) proposed a parametric semi-supervised clustering variant of Gaussian Mixture Model (GMM), **GPC**, which alternates between contrastive representation learning and category number estimation. Upon completion of training, both the encoder and cluster centers are obtained. During inference, classification is performed by identifying the nearest cluster center. Inspired by the human categorization system, Rastegar et al. (2023) propose **InfoSieve**, a representation learning method that seeks optimal hierarchical sample encoding in the embedding space. SS-KM is used for final clustering. For fine-grained semantic distinction tasks, Rastegar et al. (2024) further introduce **SelEx**, a representation learning method designed to mitigate the disruption of fine-grained semantics caused by data augmentation while providing flexible hierarchical category semantic structures.

**Parametric VGCD Methods.** Without additional constraints, a parametric MLP classifier tends to assign new-class test samples to the most similar old class, resulting in a severe bias towards old classes. Therefore, in parametric GCD, beyond improving representation learning, appropriately handling the classifier's class preference is equally crucial.

Wen et al. (2023) investigated why parametric classifiers underperform compared to the non-parametric SS-KM and identified bias towards old classes as the key issue. To mitigate this bias, they introduced Entropy Regularization (**ER**), which encourages the model's overall prediction distribution to align with a uniform class distribution. Their method, SimGCD, was the first parametric classifier to surpass Vanilla GCD. (Chiaroni et al., 2023) proposed **InfoMax**, which maximizes the mutual information between embeddings and a prototype classifier's prediction distribution. They incorporated known labeled samples into a parametric mutual information maximization objective with a weighting factor $\lambda$. To address class imbalance in long-tail distributions, they further designed a strategy for selecting $\lambda$. Cao et al. (2024) observed that in later training stages, SimGCD may misclassify old-class samples as new classes, leading to a bias towards new classes. To counteract this, they introduced **LegoGCD**, which identifies potential known samples in unlabeled data and applies entropy regularization to stabilize their predictions.

The methods mentioned above construct positive and negative sample pairs and apply contrastive loss following SimCLR and SupCon, without leveraging the rich semantic information embedded in old-class sample labels and representations. As a result, they struggle to generate high-quality pseudo-labels for training the classification head and encoder. Several methods aim to address this.

**GCA** (Otholt et al., 2024) employs unsupervised clustering to form many small local clusters and then aggregates these clusters into target classes using provided labels and neighborhood relations. The aggregated results facilitate the construction

of positive and negative pairs, which are then used to train the encoder $f$ and MLP classifier $h$ with binary cross-entropy loss on every pairs. Unlike non-parametric methods, GCA requires clustering on the entire dataset only during training to generate pseudo-labels, while at inference, it efficiently supports mini-batch predictions using only the classifier $h$. Vaze et al. (2023) tested SimGCD on the more challenging Clever4 dataset and found that it performed poorly on new classes, which they attributed to the low quality of pseudo-labels assigned to new-class samples. Inspired by Mean-Teacher semi-supervised learning (Tarvainen & Valpola, 2017) , they introduced $\mu$**GCD**, a parametric GCD method designed to be more robust to noisy labels (of new-class samples). **SPTNet** (Wang et al., 2023) introduced trainable visual prompts combined with fine-tuning model parameters to fully exploit the capabilities of pre-trained visual models and better adapt to the GCD task. All three methods—GCA, $\mu$GCD, and SPTNet—incorporate uniform label distribution as a prior to mitigate bias towards old classes.

**Current Limitations.**    While these methods have achieved success, their design and evaluation have primarily focused on image datasets, overlooking the widely prevalent graph data. To address this gap, we introduce the node-level Graph GCD task and establish multiple baseline methods. More importantly, existing GCD methods have not thoroughly investigated the mechanism by which knowledge from old categories facilitates novel category discovery. In a related task, Novel Category Discovery (NCD), Sun et al. (2023) proposed NSCL, a graph-theoretically inspired representation learning algorithm, and analyzed the intrinsic mechanisms of novel category discovery from the perspective of spectral contrastive learning theory (HaoChen et al., 2021). However, GCD is much more complex than NCD, as it involves not only distinguishing novel categories but also ensuring accurate classification of old-class sample. Furthermore, existing GCD methods differ from NSCL, making its theoretical framework inadequate to answer the fundamental question: "***When and how do old classes help generalized category discovery?***" In this work, we systematically address this question in the context of *parametric GCD*. Since different methods employ diverse representation learning modules, we aim for generalizable conclusions by conducting a theoretical analysis of classifier performance within the embedding spaces induced by encoders. After establishing the necessary embedding conditions for good GCD performance, we further examine whether the learned embeddings from graph representation learning modules satisfy these conditions. Taken together, these efforts represent our first systematic attempt to answer: "***When and how do old classes help (parametric) generalized category discovery (on graphs)?***"

### A.2. Open-world Graph Learning

In real-world applications, deployed models frequently encounter novel categories that were unseen during training. This challenge, inherent to open-world machine learning, has been extensively studied in non-graph data (Parmar et al., 2023; Zhu et al., 2024). Existing non-graph approaches for handling novel categories can be broadly classified into three paradigms. **1)** Open-Set Recognition (OSR): Accurately classifies known-class samples while assigning all unknown-class samples to a single "unknown" category (Bendale & Boult, 2016; Zhang & Patel, 2017; Perera & Patel, 2019; Zhang et al., 2020; Vaze et al., 2021; Zhou et al., 2021; Yang et al., 2022; Huang et al., 2023a). **2)** Novel Category Discovery (NCD): Clusters unknown-class samples into distinct novel categories (Han et al., 2019; Zhong et al., 2021; Han et al., 2022; Li et al., 2023; Troisemaine et al., 2023). **3)** Generalized Category Discovery (GCD): Simultaneously classifies known-class samples and clusters unknown-class samples, reviewed in Sec. A.1.

However, progress in open-world graph learning remains limited. **1)** Graph OSR: Wu et al. (2021a) introduced OpenWGL, the first transductive OSR model for static graphs, while Zhang et al. (2022) proposed OSSC, an OSR model for discrete-time dynamic graphs. The $\mathcal{G}^2Pxy$ model (Zhang et al., 2023a) addresses inductive OSR on static graphs. More recently, Zhang et al. (2024) handled inductive graph OSR tasks in the presence of in-distribution samples with incorrect labels. **2)** Graph NCD: Jin et al. (2024) proposed ORAL, the first transductive Graph NCD method for static graphs. Hou et al. (2024) introduced NC-NCD, a two-stage training framework where only labeled old-class nodes are available in the first stage, and only unlabeled new-class nodes are available in the second stage. Although NC-NCD considers all classes during testing, its staged training paradigm inherently leaks side information about the distinction between old and new categories, making it a less challenging task than GCD, where the ability to distinguish old and new categories is crucial. **3)** Other Open-World Graph Learning Tasks: Xu et al. (2024) explored Open-World Graph Active Learning, where the model selects the most valuable nodes for labeling upon the appearance of novel-category nodes and then retrains accordingly. Galke et al. (2021); Liu et al. (2021); Feng et al. (2023) studied Continual Learning on Evolving Graphs, where OSR, NCD, and GCD information is incrementally revealed to the model, making catastrophic forgetting the key challenge.

We follow the transductive setting (Vaze et al., 2022), focus on the GCD tasks for static graphs, and extend multiple VGCD

baselines to graph data.

## B. Details of GGCD Baselines Adapted from VGCD

VGCD methods can generally be broken down into two parts: 1) contrsative learning to train an encoder $f$, and 2) classification with a semi-supervised classifier $h$ that can be either non-parametric (e.g., SS-KM) or parametric (e.g., MLP). For graph adaptation, we focus on the representation learning part, because effective GNN encoders can encode graph structural information into node embeddings, making the explicit use of structures not mandatory in the classification part.

There are numerous graph contrastive learning methods available (Liu et al., 2022a; Zhu et al., 2020; Xia et al., 2022; Liu et al., 2022c; Huang et al., 2023b; Liu et al., 2022b; Deng et al., 2025). However, to minimize modifications to the VGCD method while deriving the GGCD baselines, we select GRACE, which is most similar to SimCLR, as the foundation for the graph representation part.

### B.1. Adapting SimCLR and SupCon for Graphs

**SimCLR → GRACE**: The representation learning part of VGCD methods is typically build upon SimCLR (Chen et al., 2020). The popular graph contrastive learning (GCL) method **GRACE** (Zhu et al., 2020) directly extends SimCLR to graph data by proposing the graph data augmentations and using GNN encoders. Two graph views $\mathcal{G}_1$ and $\mathcal{G}_2$ are generated from the input graph $\mathcal{G} = (\mathbf{A}, \mathbf{X})$ by applying augmentations like edge dropping, node feature masking, or subgraph sampling. A shared GNN encoder $f : (\mathcal{A}, \mathcal{X}) \to \mathcal{Z}$ maps nodes in each augmented view to embeddings. For a node $u$, its representation in $\mathcal{G}_1$ and $\mathcal{G}_2$ is separately given by:

$$\mathbf{z}_u^{(1)} = \left[\mathbf{Z}^{(1)}\right]_u = [f(\mathbf{A}_1, \mathbf{X}_1)]_u, \qquad \mathbf{z}_u^{(2)} = \left[\mathbf{Z}^{(2)}\right]_u = [f(\mathbf{A}_2, \mathbf{X}_2)]_u. \tag{12}$$

Without loss of generality, we concatenate these two views into $\mathbf{Z} = \begin{bmatrix} \mathbf{Z}^{(1)} \\ \mathbf{Z}^{(2)} \end{bmatrix}$ such that *the positive sample of node $i$ is*

$$j = \begin{cases} i + n & \textit{if i comes from view 1} \\ i - n & \textit{if i comes from view 2} \end{cases}.$$

That is $\mathbf{z}_j = \mathbf{z}_{i+n} = \mathbf{z}_i^{(2)}$ if $i \leq n$ and $\mathbf{z}_j = \mathbf{z}_{i-n} = \mathbf{z}_{i-n}^{(1)}$ if $n < i \leq 2n$. Then the **InfoNCE loss** $L_{NCE}$ (Gutmann & Hyvärinen, 2010) is used to train $f$. In terms of cosine similarity $\cos(\cdot, \cdot)$, this loss encourages the alignment of the same node's embeddings across different views while separating those of different nodes.

$$L_{NCE}(i) = L_{NCE}(\mathbf{z}_i, \mathbf{z}_j) = -\log \frac{\exp\{\cos(\mathbf{z}_i, \mathbf{z}_j)/\tau\}}{\sum_{l=1}^{2n} \exp\{\cos(\mathbf{z}_i, \mathbf{z}_l)/\tau\}}, \tau > 0 \tag{13}$$

And the loss over all nodes is

$$L_{NCE} = \sum_{i=1}^{2n} L_{NCE}(i) = -\sum_{i=1}^{2n} \log \frac{\exp\{\cos(\mathbf{z}_i, \mathbf{z}_j)/\tau\}}{\sum_{l=1}^{2n} \exp\{\cos(\mathbf{z}_i, \mathbf{z}_l)/\tau\}}, \tau > 0. \tag{14}$$

**SupCon → GRACE-SC**: Since **SupCon** (Khosla et al., 2020) differs from SimCLR solely in its use of labels to construct positive and negative pairs, it can be easily adapted to graphs by building on GRACE. We replace the InfoNCE loss in GRACE with the **SupCon loss**

$$L_{SC}(i) = L_{SC}(\mathbf{z}_i, \mathbf{z}_j) = \frac{-1}{m_i} \sum_{j \in \mathcal{S}(i) \setminus i} \log \frac{\exp\{\cos(\mathbf{z}_i, \mathbf{z}_j)/\tau\}}{\sum_{l=1}^{2n} \exp\{\cos(\mathbf{z}_i, \mathbf{z}_l)/\tau\}}, \tag{15}$$

$$L_{SC} = \sum_{i=1}^{2n} L_{SC}(i) = -\sum_{i=1}^{2n} \frac{1}{m_i} \sum_{j \in \mathcal{S}(i) \setminus i} \log \frac{\exp\{\cos(\mathbf{z}_i, \mathbf{z}_j)/\tau\}}{\sum_{l=1}^{2n} \exp\{\cos(\mathbf{z}_i, \mathbf{z}_l)/\tau\}} \tag{16}$$

where $\mathcal{S}(i)$ includes all augmented nodes with the same label as node $i$, and $m_i$ is the cardinality of the set $\mathcal{S}(i) \setminus i$. We refer to this SupCon adaptation as **GRACE-SC**.

## B.2. Adapting VGCD Methods for Graphs

**Vanilla GCD** (Vaze et al., 2022)    uses SimCLR and SupCon for representation learning, and then Semi-Supervised K-means (SS-KM) for label assignment. SS-KM utilizes the known old-class nodes to derive old-class centroids and enforces the assignment of these nodes to their respective old-class centroids. Unlabeled nodes are subsequently clustered following the standard K-means algorithm. *Adaptation*: We replace SimCLR and SupCon with GRACE and GRACE-SC introduced in Sec. B.1. All other components remain unchanged. The total training loss is

$$\mathcal{L}_{VanillaGCD} = (1 - \alpha_2)L_{NCE} + \alpha_2 L_{SC}, \qquad \alpha_2 > 0.$$

**UNO+** (Fini et al., 2021; Vaze et al., 2022)    extends the self-supervised CL method SwAV (Caron et al., 2020) to the semi-supervised GCD task. It employs a parametric classifier to predict old and new class labels for each augmented view. A cost matrix is then constructed based on the predictions to solve the Optimal Transport (OT) assignment, where the assignment of one view serves as pseudo-labels for the other view. *Adaptation*: Data augmentation follows GRACE, with $f(\cdot)$ replaced by a GCN. All other components remain unchanged.

**SimGCD** (Wen et al., 2023)    adopts the representation learning methods in Vanilla GCD while replacing SS-KM with a parametric prototype classifier. The classifier provides predictions for two views, and then the prediction for every view is distilled as the pseudo-label to the other view. This approach is Self-Distillation (**SD**) (Assran et al., 2022) with the **SD loss**

$$L_{SD}(i) = L_{CE}(\mathbf{h}_j, \mathbf{h}_i)$$
$$L_{SD} = \sum_{i=1}^{2n} L_{SD}(i) = \sum_{i=1}^{2n} D_{CE}(\mathbf{h}_j, \mathbf{h}_i). \tag{17}$$

For the labeled samples, SimGCD employs the Cross Entropy (CE) loss

$$L_{CE} = \sum_{i \in \mathcal{D}_L} D_{CE}(\mathbf{y}_i, \mathbf{h}_i). \tag{18}$$

Additionally, SimCGD utilizes the Entropy Regularization (**ER**) loss $L_{ER}$ (Assran et al., 2022) on the entire $\mathcal{D}$ to mitigate the bias towards old classes , which becomes popular later in VGCD (Vaze et al., 2023; Wang et al., 2023).

$$L_{ER} = -H(\bar{\mathbf{h}}) = \sum_{c=1}^{C} \left[\bar{\mathbf{h}}\right]_c \log \left[\bar{\mathbf{h}}\right]_c \tag{19}$$
$$\bar{\mathbf{h}} = \frac{1}{2n} \sum_{i \in 2n} \mathbf{h}_i \tag{20}$$

Finally, the training objective for SimGCD is

$$\mathcal{L}_{SimGCD} = (1 - \alpha_2)(L_{NCE} + L_{SD} + \alpha_1 L_{ER}) + \alpha_2(L_{SC} + L_{CE}), \qquad \alpha_1, \alpha_2 > 0. \tag{21}$$

*Adaptation*: We replace SimCLR and SupCon with GRACE and GRACE-SC introduced in Sec. B.1. All other components remain unchanged.

After adapting these VGCD methods for graph data, the GNN encoder (e.g., GCN) maps the graph data into the embedding space, i.e., $f : (\mathcal{A}, \mathcal{X}) \to \mathcal{Z}$. And the final prediction is made by the parametric or non-parametric classifier $h : \mathcal{Z} \to \mathcal{Y}_C$, where $\mathcal{Y}_C$ is the set of all probability distributions over $C$ categories, i.e., $\mathcal{Y}_C = \left\{ \mathbf{y} \in \mathbb{R}_{\geq 0}^C \mid \sum_c^C y_c = 1 \right\}$. Unless otherwise specified, in the following content, we will refer to the adapted versions of VGCD baselines for graph datasets directly by their original names. For instance, the adapted version of SimGCD will still be referred to as SimGCD.

## C. More Details of the Proposed GCL Method, SWIRL

### C.1. The Total Loss and Full Procedure of SWIRL

---

**Algorithm 1** The full procedure of SWIRL

---

**Input Data:** The graph $\mathcal{G} = (\mathbf{A}, \mathbf{X})$ with $E$ edges and $d_0$ node feature dimensions; The full node set $\mathcal{D} = (\mathbf{A}, \{(\mathbf{x}_i, y_i)\}_{i=1}^n)$; The labeled subset $\mathcal{D}_L = (\mathbf{A}, \{(\mathbf{x}_i, y_i)\}_{i=1}^{n_\mathcal{L}})$ of $C_0$ old classes; The unlabeled subset $\mathcal{D}_U$ of nodes from all $C > C_0$ classes.

**Model:** The GCN encoder $f : (\mathcal{A}, \mathcal{X}) \to \mathcal{Z}$ and the MLP classifier $h : \mathcal{Z} \to \mathcal{Y}_\mathcal{C}$

**Common Hyperparameters:** The edge drop rate $pe$ and the feature drop rate $px$ in data augmentation module; The training epochs $T$; The loss weights $\alpha_1 = 2.$ and $\alpha_2 = 0.35$.

**SWIRL Hyperparameters:** The prototype number $K \geq C$; The repulsion force scale $s$; The weight $\beta_1$ of the SWIRL loss.

**Output**: The predicted labels for all unlabeled nodes.

---

1: **while** $t < T$ **do**
2:     Perform SS-KM on $\bar{\mathbf{Z}}$ to get $K$ prototypes $\{\mathbf{s}^1, \ldots, \mathbf{s}^K\}$, which include $C_o$ old prototypes $\mathcal{O} = \{\mathbf{s}^1, \ldots, \mathbf{s}^{C_o}\}$.
3:     Based on $\mathbf{Z}$, compute $L_{SWIRL}$ with Eq. (11), $L_{NCE}$ with Eq. (14), $L_{ER}$ with Eq. (19), and $L_{CE}$ with Eq. (18).
4:     Compute the total training loss $\mathcal{L}_{SWIRL} = (1 - \alpha_2)(L_{NCE} + \beta_1 L_{SWIRL} + \alpha_1 L_{ER}) + \alpha_2 L_{CE}$.
5:     Backward to update $f$ and $h$.
6:     $t \leftarrow t + 1$.
7: **end while**
8: Set the model ($f$ and $h$) to 'eval' mode and make predictions.

---

## C.2. The Gradients of InfoNCE-style Loss with Pair Weights

Our SWIRL loss can be viewed as a weighted InfoNCE-style loss. Here, we analyze the impact of the weight $r(i, k)$ on the gradient of the loss function w.r.t. the pairwise distance between samples, i.e., the repulsion force.

$$
\begin{aligned}
L_w &= -\log \frac{\exp\{-D(\mathbf{z}_i, \mathbf{z}_j)/\tau\}}{\sum_{k=1}^{2n} r(i, k) \exp\{-D(\mathbf{z}_i, \mathbf{z}_k)/\tau\}} \\
&= \frac{D(\mathbf{z}_i, \mathbf{z}_j)}{\tau} + \log\left(\sum_{k=1}^{2n} r(i, k) \exp\{-D(\mathbf{z}_i, \mathbf{z}_k)/\tau\}\right), \qquad \tau > 0
\end{aligned}
$$

Differentiating $L_w$ with respect to the positive-pair distance $D(\mathbf{z}_i, \mathbf{z}_j)$ leads to

$$
\frac{\partial L_w}{\partial D(\mathbf{z}_i, \mathbf{z}_j)} = \frac{1}{\tau} - \frac{r(i, j) \exp\{-D(\mathbf{z}_i, \mathbf{z}_j)/\tau\}}{\tau \sum_{k=1}^{2n} r(i, k) \exp\{-D(\mathbf{z}_i, \mathbf{z}_k)/\tau\}}. \tag{22}
$$

Similarly, we get the gradient w.r.t. the negative-pair distance $D(\mathbf{z}_i, \mathbf{z}_k)$

$$
\frac{\partial L_w}{\partial D(\mathbf{z}_i, \mathbf{z}_k)} = -\frac{r(i, k) \exp\{-D(\mathbf{z}_i, \mathbf{z}_k)/\tau\}}{\tau \sum_{k'=1}^{2n} r(i, k') \exp\{-D(\mathbf{z}_i, \mathbf{z}_{k'})/\tau\}}. \tag{23}
$$

Denote by $p(k|i)$ the softmaxed weighted similarity

$$
p(k|i) = \frac{r(i, k) \exp\{-D(\mathbf{z}_i, \mathbf{z}_k)/\tau\}}{\sum_{k=1}^{2n} r(i, k') \exp\{-D(\mathbf{z}_i, \mathbf{z}_{k'})/\tau\}},
$$

then we get

$$
\frac{\partial L_w}{\partial D(\mathbf{z}_i, \mathbf{z}_j)} = \frac{1 - p(j|i)}{\tau} \tag{24}
$$

$$
\frac{\partial L_w}{\partial D(\mathbf{z}_i, \mathbf{z}_k)} = -\frac{p(k|i)}{\tau}. \tag{25}
$$

**For positive-pair** $(i, j)$

- When increasing $r(i, j)$ and thus $p(j|i)$, the positive-pair distance gets a smaller gradient magnitude as Eq. (24) and the attraction is slowed down.

- Conversely, decreasing $r(i, j)$ causes a stronger attraction force that brings $i$ and $j$ together.

**For negative-pair** $(i, k)$

- When increasing $r(i, k)$ and thereby $p(k|i)$, the negative-pair distance gets a larger gradient magnitude as Eq. (25) and the repulsion is accelerated.

- Conversely, decreasing $r(i, k)$ leads to a weaker repulsion force.

# D. Some Proof Auxiliaries

## D.1. Auxiliaries for Theorems on Parametric GCD Classifiers in Sec. 3

**Theorem D.1** (Bolley et al. (2007), Theorem 2.1). *Let $\mu$ be a probability measure in the metric space $(\mathcal{Z}, D(\cdot, \cdot))$ that admits a square-exponential moment. That is for any $\mathbf{z}_2 \in \mathcal{Z}$*

$$\int e^{\alpha D^2(\mathbf{z}_1, \mathbf{z}_2)} d\mu(\mathbf{z}_1) < +\infty,$$

*or equivalently $\mu$ satisfies the Talagrand inequality $T_1(c) : W_1(\eta, \mu) \leq \sqrt{\frac{2}{c} H(\eta|\mu)}, c > 0$ (Djellout et al., 2004), where $W_1(\eta, \mu)$ is the Wassersteom distance. Denote by $\hat{\mu} := \frac{1}{n} \sum_{i=1}^{N} \delta_{\mathbf{z}_i}$ the corresponding empirical measure defineded on a sample of independent variables $\{\mathbf{z}_i\}_{i=1}^{N}$ drawn from $\mu$. Then, for any $d' > \dim(\mathcal{Z})$ and $c' < c$, there exists a constant $N_0$ determined only by $c'$, $d'$, and some square-expoential moment of $\mu$, such that for any $\varepsilon > 0$ and $N \geq N_0 \max \left( \varepsilon^{-(d'+2)}, 1 \right)$*

$$\Pr[W_1(\mu, \hat{\mu}) > \varepsilon] \leq \exp \left( -\frac{c'}{2} N\varepsilon^2 \right). \tag{26}$$

*And by setting $\exp \left( -\frac{c'}{2} N\varepsilon^2 \right) = \delta$, it follows that with probability at least $1 - \delta$,*

$$W_1(\mu, \hat{\mu}) \leq \sqrt{\frac{2}{Nc'} \ln \frac{1}{\delta}}. \tag{27}$$

**Theorem D.2.** *Given a hypothesis $h : \mathcal{Z} \to \mathcal{S}_\mathcal{C}$ that outputs a prediction in a $|\mathcal{C}|$-dimensional simplex (i.e., non-negative entries summing to 1), we denote the loss on one sample $(\mathbf{z}_i, \mathbf{y}_i)$ that adheres to Assumption 3.2 by $L_i = L(\mathbf{z}_i, \mathbf{y}_i)$. Let the empirical mean loss on $n$ samples $\{\mathbf{z}_i, \mathbf{y}_i\}_{i=1}^{n}$ and the generalization loss respectively be*

$$\hat{R}(h) = \frac{1}{n} \sum_{i=1}^{n} L_i = \frac{1}{n} \sum_{i=1}^{n} L(\mathbf{z}_i, \mathbf{y}_i)$$

$$R(h) = \mathbb{E}_{(\mathbf{z}, \mathbf{y})} L(\mathbf{z}, \mathbf{y}).$$

*Then it holds that for $\varepsilon > 0$, $\Pr \left\{ |\hat{R}(h) - R(h)| \geq \varepsilon \right\} \leq 2 \exp(-2n\varepsilon^2/S^2)$. And with probability at least $1 - \delta$,*

$$R(h) \leq \hat{R}(h) + \sqrt{\frac{S^2}{2n} \ln \frac{2}{\delta}}. \tag{28}$$

*Proof.* The loss $L_i$ is bounded in $[0, S]$ according to Assumption 3.2. Directly applying Hoeffding's Inequality finishes the proof, similar to Corollary 2.10 in (Mohri et al., 2018). Ineq. (28) is obtained by setting $2 \exp(-2n\varepsilon^2/S^2) = \delta$. $\qquad \square$

**Lemma D.3.** *If $L(\cdot, \cdot)$ is a convex function w.r.t. one input parameter given the other fixed, then it follows that*

$$\mathbb{E}_{\mathbf{z} \sim p_u} \mathbb{E}_{\mathbf{z}^- \sim p_{u'}} L(h(\mathbf{z}), h(\mathbf{z}^-)) \geq L(\mathbb{E}_{\mathbf{z} \sim p_u} h(\mathbf{z}), \mathbb{E}_{\mathbf{z}^- \sim p_{u'}} h(\mathbf{z}^-)).$$

*Proof.* Since $L$ is a convex function, we have, according to Jensen's inequality, that

$$\mathbb{E}_{\mathbf{z}\sim p_u}\mathbb{E}_{\mathbf{z}^-\sim p_{u'}}L(h(\mathbf{z}),h(\mathbf{z}^-)) = \int_{\mathbf{z}} p_u(\mathbf{z})\int_{\mathbf{z}^-} p_u(\mathbf{z}^-)L(h(\mathbf{z}),h(\mathbf{z}^-))d\mathbf{z}d\mathbf{z}^- \tag{29a}$$

$$= \int_{\mathbf{z}} p_u(\mathbf{z})d\mathbf{z}\int_{\mathbf{z}^-} p_u(\mathbf{z}^-)L(h(\mathbf{z}),h(\mathbf{z}^-))d\mathbf{z}^- \tag{29b}$$

$$\geq \int_{\mathbf{z}} p_u(\mathbf{z})L(h(\mathbf{z}),\mathbb{E}(h(\mathbf{z}^-)))d\mathbf{z} \tag{29c}$$

$$\geq L(\mathbb{E}(h(\mathbf{z})),\mathbb{E}(h(\mathbf{z}^-))). \tag{29d}$$

$\square$

**Lemma D.4.** *Given a $\phi(\lambda)$-Lipschitz transferable hypothesis $h$ (Assumption 3.4), the union of regions where the transferable Lipschitzness does not hold is denoted by*

$$\Omega := \{(\mathbf{z}_1,\mathbf{z}_2) : L(h(\mathbf{z}_1),h(\mathbf{z}_2)) \leq \lambda D(\mathbf{z}_1,\mathbf{z}_2)\},$$

*where $\lambda > 0$, $D(\cdot,\cdot)$ is a metric in $\mathcal{Z}$ and $L(\cdot,\cdot)$ satisfies Assumptions 3.3 and 3.2. Then it holds that*

$$\int_{(\mathcal{Z}\times\mathcal{C})^2} kL(h(\mathbf{z}_1),h(\mathbf{z}_2))d\pi((\mathbf{z}_1,u),(\mathbf{z}_2,l)) \leq kS\phi(\lambda) + \int_{(\mathcal{Z}\times\mathcal{C})^2} k\lambda D(\mathbf{z}_1,\mathbf{z}_2)d\pi((\mathbf{z}_1,u),(\mathbf{z}_2,l)). \tag{30}$$

*Proof.* In the following proof, Ineq. (31a) comes from that $L(h(\mathbf{z}_1),h(\mathbf{z}_2))$ is bounded by $S$ in the region $(\mathcal{Z}\times\mathcal{Z})\setminus\Omega$ (see Assumption 3.2). Ineq. (31b) is due to the definition of the region $\Omega$, which admits the transferable Lipschitzness defined in Assumption (3.4). Ineq. (31c) is the result of non-negativity of a norm $D$.

$$\int_{(\mathcal{Z}\times\mathcal{C})^2} kL(h(\mathbf{z}_1),h(\mathbf{z}_2))d\pi((\mathbf{z}_1,u),(\mathbf{z}_2,l))$$

$$= \int_{\mathcal{Z}}\int_{\mathcal{Z}} kL(h(\mathbf{z}_1),h(\mathbf{z}_2))\int_{\mathcal{C}}\int_{\mathcal{C}}\pi((\mathbf{z}_1,u),(\mathbf{z}_2,l))d\mathbf{z}_1 du d\mathbf{z}_2 dl$$

$$= \int_{\mathcal{Z}}\int_{\mathcal{Z}} kL(h(\mathbf{z}_1),h(\mathbf{z}_2))d\mathbf{z}_1 d\mathbf{z}_2\int_{\mathcal{C}}\int_{\mathcal{C}}\pi(\mathbf{z}_1,\mathbf{z}_2)\pi(u,l|\mathbf{z}_1,\mathbf{z}_2)du dl$$

$$= \int_{\mathcal{Z}\times\mathcal{Z}} kL(h(\mathbf{z}_1),h(\mathbf{z}_2)\pi(\mathbf{z}_1,\mathbf{z}_2)d\mathbf{z}_1 d\mathbf{z}_2$$

$$= \int_{(\mathcal{Z}\times\mathcal{Z})\setminus\Omega} kL(h(\mathbf{z}_1),h(\mathbf{z}_2)\pi(\mathbf{z}_1,\mathbf{z}_2)d\mathbf{z}_1 d\mathbf{z}_2 + \int_{\Omega} kL(h(\mathbf{z}_1),h(\mathbf{z}_2)\pi(\mathbf{z}_1,\mathbf{z}_2)d\mathbf{z}_1 d\mathbf{z}_2$$

$$\leq k\int_{(\mathcal{Z}\times\mathcal{Z})\setminus\Omega} S\pi(\mathbf{z}_1,\mathbf{z}_2)d\mathbf{z}_1 d\mathbf{z}_2 + \int_{\Omega} kL(h(\mathbf{z}_1),h(\mathbf{z}_2)\pi(\mathbf{z}_1,\mathbf{z}_2)d\mathbf{z}_1 d\mathbf{z}_2 \tag{31a}$$

$$\leq kS\int_{(\mathcal{Z}\times\mathcal{Z})\setminus\Omega} \pi(\mathbf{z}_1,\mathbf{z}_2)d\mathbf{z}_1 d\mathbf{z}_2 + \int_{\Omega} k\lambda D(\mathbf{z}_1,\mathbf{z}_2)\pi(\mathbf{z}_1,\mathbf{z}_2)d\mathbf{z}_1 d\mathbf{z}_2 \tag{31b}$$

$$= kS\phi(\lambda) + \int_{\Omega} k\lambda D(\mathbf{z}_1,\mathbf{z}_2)\pi(\mathbf{z}_1,\mathbf{z}_2)d\mathbf{z}_1 d\mathbf{z}_2$$

$$\leq kS\phi(\lambda) + \int_{\mathcal{Z}\times\mathcal{Z}} k\lambda D(\mathbf{z}_1,\mathbf{z}_2)\pi(\mathbf{z}_1,\mathbf{z}_2)d\mathbf{z}_1 d\mathbf{z}_2 \tag{31c}$$

$$= kS\phi(\lambda) + \int_{(\mathcal{Z}\times\mathcal{C})^2} k\lambda D(\mathbf{z}_1,\mathbf{z}_2)d\pi((\mathbf{z}_1,u),(\mathbf{z}_2,l)) \tag{31d}$$

$\square$

**Lemma D.5.** *Given a function $h^* : \mathcal{Z} \to \mathcal{Y}_{\mathcal{C}}$ and two joint distributions $p(\mathbf{z}_1,u)$ and $p(\mathbf{z}_2,l)$ over the space $\mathcal{Z}\times\mathcal{C}$, the*

*following expectation about $L : \mathcal{Y}_C \times \mathcal{Y}_C \to \mathbb{R}_{\geq 0}$ over the coupling distribution $\pi((\mathbf{z}_1, u), (\mathbf{z}_2, l))$ hold.*

$$\int_{(\mathcal{Z} \times \mathcal{C})^2} \left[ L(h^*(\mathbf{z}_1), \mathbf{y}^u) - L(h^*(\mathbf{z}_2), \mathbf{y}^l) \right] d\pi((\mathbf{z}_1, u), (\mathbf{z}_2, l))$$

$$= \int_{\mathcal{Z} \times \mathcal{C}} p(\mathbf{z}_1, u) L(h^*(\mathbf{z}_1), \mathbf{y}^u) d(\mathbf{z}_1, u) - \int_{\mathcal{Z} \times \mathcal{C}} p(\mathbf{z}_2, l) L(h^*(\mathbf{z}_2), \mathbf{y}^l) d(\mathbf{z}_2, l)$$

*Proof.* The proof is substantially based on the definition of coupling, which reads that $p(\mathbf{z}_1, u)$ and $p(\mathbf{z}_2, l)$ are two marginal distributions of it.

$$\int_{(\mathcal{Z} \times \mathcal{C})^2} \left[ L(h^*(\mathbf{z}_1), \mathbf{y}^u) - L(h^*(\mathbf{z}_2), \mathbf{y}^l) \right] d\pi((\mathbf{z}_1, u), (\mathbf{z}_2, l)) \tag{32a}$$

$$= \int_{(\mathcal{Z} \times \mathcal{C})^2} L(h^*(\mathbf{z}_1), \mathbf{y}^u) d\pi((\mathbf{z}_1, u), (\mathbf{z}_2, l)) - \int_{(\mathcal{Z} \times \mathcal{C})^2} L(h^*(\mathbf{z}_2), \mathbf{y}^l) d\pi((\mathbf{z}_1, u), (\mathbf{z}_2, l)) \tag{32b}$$

$$= \int_{(\mathcal{Z} \times \mathcal{C})^2} L(h^*(\mathbf{z}_1), \mathbf{y}^u) \pi((\mathbf{z}_1, u), (\mathbf{z}_2, l)) d(\mathbf{z}_1, u) d(\mathbf{z}_2, l)$$
$$\qquad - \int_{(\mathcal{Z} \times \mathcal{C})^2} L(h^*(\mathbf{z}_2), \mathbf{y}^l) \pi((\mathbf{z}_1, u), (\mathbf{z}_2, l)) d(\mathbf{z}_1, u) d(\mathbf{z}_2, l) \tag{32c}$$

$$= \int_{\mathcal{Z} \times \mathcal{C}} \pi((\mathbf{z}_1, u), (\mathbf{z}_2, l)) d(\mathbf{z}_2, l) \int_{\mathcal{Z} \times \mathcal{C}} L(h^*(\mathbf{z}_1), \mathbf{y}^u) d(\mathbf{z}_1, u)$$
$$\qquad - \int_{\mathcal{Z} \times \mathcal{C}} \pi((\mathbf{z}_1, u), (\mathbf{z}_2, l)) d(\mathbf{z}_1, u) \int_{\mathcal{Z} \times \mathcal{C}} L(h^*(\mathbf{z}_2), \mathbf{y}^l) d(\mathbf{z}_2, l) \tag{32d}$$

$$= \int_{\mathcal{Z} \times \mathcal{C}} p(\mathbf{z}_1, u) L(h^*(\mathbf{z}_1), \mathbf{y}^u) d(\mathbf{z}_1, u) - \int_{\mathcal{Z} \times \mathcal{C}} p(\mathbf{z}_2, l) L(h^*(\mathbf{z}_2), \mathbf{y}^l) d(\mathbf{z}_2, l). \tag{32e}$$

$\square$

## D.2. Auxiliaries for Theorems on (Graph) Contrastive Learning Presented in Sec. 4

**Theorem D.6** (Assel et al. (2022); Theorem 1)**.** *The Pairwise Markov Random Field (PMRF) on the graph $\mathbf{W}$ is with the unnormalized density function*

$$f_k(\mathbf{Z}, \mathbf{W}) \to \prod_{(i,j) \in [N]^2} k(\mathbf{z}_i - \mathbf{z}_j)^{W_{ij}}.$$

*Let each category correspond to a graph connected component, then $\mathbf{Z} \in \mathbb{R}^{N \times d}$ can be orthogonally decomposed into $\mathcal{S}_0 = (\ker \mathbf{L}) \otimes \mathbb{R}^d$ and $\mathcal{S}_1 = (\ker \mathbf{L})^\perp \otimes \mathbb{R}^d$, where $\mathbf{L}$ is the graph Laplacian matrix of $\mathbf{W}$. If $k$ is $\lambda_{\mathbb{R}^d}$-integrable and bounded above $\lambda_{\mathbb{R}^d}$ almost everywhere, then $f_k(\mathbf{Z}, \mathbf{W})$ is $\lambda_{\mathcal{S}_1}$-integrable.*

**Lemma D.7** (Tan et al. (2023); Lemma 2.4)**.** *Suppose the space $\mathcal{X}$ is constructed by $M$ spaces $\mathcal{X} = \mathcal{X}_1 \times \mathcal{X}_2 \times \cdots \times \mathcal{X}_M$. On the $m$-th subspace there are probability distributions $P_m$ and $Q_m$ over $\mathcal{X}_m$. Then the cross entropy between the joint distributions $P = P_1 \otimes P_2 \otimes \cdots \otimes P_M$ and $Q = Q_1 \otimes Q_2 \otimes \cdots \otimes Q_M$ can be decomposed into all subspaces*

$$D_{CE}(P \| Q) \triangleq -\mathbb{E}_{x \sim P} \log Q(x) = \sum_{m=1}^{M} D_{CE}(P_m \| Q_m).$$

**Lemma D.8.** *If $k$ is shift invariant kernel, then $f_k(\mathbf{Z}, \mathbf{W})$ is not integrable on $\mathcal{S}_0$.*

*Proof.* With the shift property of $k$, we readily get

$$
\begin{aligned}
f_k(\mathbf{Z}_1, \mathbf{W}) &= f_k(\mathbf{Z} - \mathbf{Z}_0, \mathbf{W}) \\
&= \prod_{(i,j)\in[N]^2} k(\mathbf{z}_i - \mathbf{z}_{0,i}, \mathbf{z}_j - \mathbf{z}_{0,j})^{W_{ij}} \\
&= \prod_{(i,j)\in[N]^2} k(\mathbf{z}_i - \mathbf{z}_{0,i}, \mathbf{z}_j - \mathbf{z}_{0,i})^{W_{ij}} \\
&= \prod_{(i,j)\in[N]^2} k(\mathbf{z}_i, \mathbf{z}_j)^{W_{ij}} = f_k(\mathbf{Z}, \mathbf{W}),
\end{aligned}
\tag{33}
$$

where Eq. (33) is because that $k(\mathbf{z}_i - \mathbf{z}_{0,i}, \mathbf{z}_j - \mathbf{z}_{0,i})^{W_{ij}} \neq 1$ only when $j$ is from the same category as $i$ (i.e., $\mathbf{z}_{0,i} = \mathbf{z}_{0,j}$). In contrast, if they are from different classes (and hence clusters), $k(\mathbf{z}_i - \mathbf{z}_{0,i}, \mathbf{z}_j - \mathbf{z}_{0,j})^{W_{ij}} = 1$ contributes nothing to the density function. Such invariance and non-negativity of $f_k$ makes it not integrable on $\mathcal{S}_0$. $\square$

## E. Proofs of Theorems on Parametric GCD Classifiers in Sec. 3

### E.1. Proof of GCD Upper Bound in Theorem 3.5

**Theorem E.1** (Restate Theorem 3.5). *Let Assumptions 3.2, 3.3 and 3.4 hold and $h^*$be the optimal hypothesis that minimizes the GCD loss Eq. (5). Let $n_{\mathcal{L}} = \sum_l n_l$ and $n_{\mathcal{U}} = \sum_u n_u$ be the total numbers of old and new class samples, respectively. Then there exists, $c'$ and $n_1$, such that for $n_{\mathcal{U}} > n_1$, $n_{\mathcal{L}} > n_1$, and all $\lambda > 0$, $L_{te}(h)$ is bounded as, with probability at least $1 - \delta - \omega$ ($\delta, \omega > 0$),*

$$
\begin{aligned}
L_{te}(h) \leq & \alpha F + (2+\beta)\hat{L}_{\mathcal{L}}(h^*) + 2W^{r\lambda}(\hat{p_{\mathcal{U}}}(\mathbf{z}, y), \hat{p_{\mathcal{L}}}(\mathbf{z}, y)) \\
& + \frac{\gamma}{b}D_{KL}(P_{\mathcal{C}} \| \bar{h}) + 2E_{\mathcal{U}} + E_{\mathcal{L}} + T_G, \\
T_G = & (2+\beta)\sqrt{\frac{S^2}{2n'_{\mathcal{L}}}\ln\frac{2}{\omega}} \\
& + 2\left[ rS\phi(\lambda) + \sqrt{\frac{2}{c'}\log\frac{2}{\delta}}\left(\frac{1}{\sqrt{n_{\mathcal{U}}}} + \frac{1}{\sqrt{n_{\mathcal{L}}}}\right) \right],
\end{aligned}
$$

*where $b = \Pr(y \in \mathcal{U})$, $P_{\mathcal{C}}$ is the uniform distribution over $\mathcal{C}$, $\bar{h} = \mathbb{E}_{p(\mathbf{z},y)}h(\mathbf{z})$ is the expected prediction over both the old and new data, and $\hat{p_{\mathcal{L}}}(\mathbf{z}, y)$ and $\hat{p_{\mathcal{U}}}(\mathbf{z}, y)$ are respectively the empirical old and new joint distributions. $W^{r\lambda}$ is hereafter abbreviated as $W$. $n'_{\mathcal{L}}$ is the number of old-class training samples while $n_{\mathcal{L}}/n_{\mathcal{U}}$ is the total number of old-/new-class samples in the whole embedding dataset $\mathcal{D}^z$.*

*Proof.* We first expand the GCD loss

$$
L_{te}(h) = \underbrace{\beta L_{\mathcal{L}}(h) + L_{\mathcal{U}}(h)}_{L_{te}^G(h} + \gamma D_{KL}(p_{\mathcal{U}}(c) \| \bar{h}_{\mathcal{U}}), \beta, \gamma > 0,
\tag{35}
$$

and then bound the consequent components. By finding the upper bounds for these components, we successfully connect the old-class and new-class data with the Wasserstein distance between joint (embedding, label) distributions.

**Step 1** We first tackle $L_{te}^G(h)$, leaving the entropy regularization to later.

$$L_{te}^G(h) = \beta L_{\mathcal{L}}(h) + L_{\mathcal{U}}(h) \tag{36a}$$

$$= \beta \sum_{l \in \mathcal{L}} p_{\mathcal{L}}(l) \mathbb{E}_{\mathbf{z} \sim p_l} L(h(\mathbf{z}), \mathbf{y}^l) + \sum_{u \in \mathcal{U}} p_{\mathcal{U}}(u) \mathbb{E}_{\mathbf{z} \sim p_u} \mathbb{E}_{\mathbf{z}^+ \sim p_u} L(h(\mathbf{z}), h(\mathbf{z}^+)) \tag{36b}$$

$$- \alpha \sum_{u \in \mathcal{U}} p_{\mathcal{U}}(u) \sum_{u \neq u'} p_{\mathcal{U}}(u') \mathbb{E}_{\mathbf{z} \sim p_u} \mathbb{E}_{\mathbf{z}^- \sim p_{u'}} L(h(\mathbf{z}), h(\mathbf{z}^-)) \tag{36c}$$

$$\leq \sum_{u \in \mathcal{U}} p_{\mathcal{U}}(u) \left[ \mathbb{E}_{\mathbf{z} \sim p_u} \mathbb{E}_{\mathbf{z}^+ \sim p_u} L(h(\mathbf{z}), h^*(\mathbf{z})) + L(h^*(\mathbf{z}), h^*(\mathbf{z}^+)) + L(h^*(\mathbf{z}^+), h(\mathbf{z}^+)) \right] \tag{36d}$$

$$+ \alpha \underbrace{\sum_{u \in \mathcal{U}} p_{\mathcal{U}}(u) \sum_{u \neq u'} p_{\mathcal{U}}(u') \left[ -\mathbb{E}_{\mathbf{z} \sim p_u} \mathbb{E}_{\mathbf{z}^- \sim p_{u'}} L(h(\mathbf{z}), h(\mathbf{z}^-)) \right]}_{F_1 : \text{The repulsion force between new classes}} + \beta \sum_{l \in \mathcal{L}} p_{\mathcal{L}}(l) \mathbb{E}_{\mathbf{z} \sim p_l} L(h(\mathbf{z}), \mathbf{y}^l) \tag{36e}$$

$$= 2 \sum_{u \in \mathcal{U}} p_{\mathcal{U}}(u) \mathbb{E}_{\mathbf{z} \sim p_u} L(h(\mathbf{z}), h^*(\mathbf{z})) + \sum_{u \in \mathcal{U}} p_{\mathcal{U}}(u) \mathbb{E}_{\mathbf{z} \sim p_u} \mathbb{E}_{\mathbf{z}^+ \sim p_u} L(h^*(\mathbf{z}), h^*(\mathbf{z}^+)) \tag{36f}$$

$$+ \alpha F_1 + \beta \sum_{l \in \mathcal{L}} p_{\mathcal{L}}(l) \mathbb{E}_{\mathbf{z} \sim p_l} L(h(\mathbf{z}), \mathbf{y}^l) \tag{36g}$$

$$\leq 2 \underbrace{\sum_{u \in \mathcal{U}} p_{\mathcal{U}}(u) \mathbb{E}_{\mathbf{z} \sim p_u} L(h(\mathbf{z}), h^*(\mathbf{z}))}_{E_{\mathcal{U}} : \text{The estimation error on new class data}} + \beta \underbrace{\sum_{l \in \mathcal{L}} p_{\mathcal{L}}(l) \mathbb{E}_{\mathbf{z} \sim p_l} L(h(\mathbf{z}), h^*(\mathbf{z}))}_{E_{\mathcal{L}} : \text{The estimation error on old class data}} + \alpha F_1 \tag{36h}$$

$$+ (2 + \beta) \underbrace{\sum_{l \in \mathcal{L}} p_{\mathcal{L}}(l) \mathbb{E}_{\mathbf{z} \sim p_l} L(h^*(\mathbf{z}), \mathbf{y}^l)}_{R_{\mathcal{L}}(h^*) : \text{The old class error of } h^*} \tag{36i}$$

$$+ \underbrace{\sum_{u \in \mathcal{U}} p_{\mathcal{U}}(u) \mathbb{E}_{\mathbf{z} \sim p_u} \mathbb{E}_{\mathbf{z}^+ \sim p_u} L(h^*(\mathbf{z}), h^*(\mathbf{z}^+))}_{\text{The attraction within new classes}} - 2 \sum_{l \in \mathcal{L}} p_{\mathcal{L}}(l) \mathbb{E}_{\mathbf{z} \sim p_l} L(h^*(\mathbf{z}), \mathbf{y}^l) \tag{36j}$$

$$\underbrace{\qquad\qquad\qquad\qquad\qquad\qquad\qquad\qquad\qquad\qquad\qquad\qquad\qquad\qquad}_{T_1 : \text{The interplay between old and new classes}}$$

$$= 2 E_{\mathcal{U}} + E_{\mathcal{L}} + \alpha F_1 + (2 + \beta) R_{\mathcal{L}}(h^*) + T_1 \tag{36k}$$

**Step 2** Now we further cope with $R_{\mathcal{L}}(h^*)$.

Like classical generalization bound on single hypothesis (Mohri et al., 2018), we relate our bound with the empirical loss $\hat{R}_{\mathcal{L}}(h^*) = \sum_{l \in \mathcal{L}} \hat{p_{\mathcal{L}}}(l) \mathbb{E}_{\mathbf{z} \sim \hat{p}_l} L(h^*(\mathbf{z}), \mathbf{y}^l)$ by applying Theorem D.2 to $R_{\mathcal{L}}(h^*)$. Then with probability at least $1 - \omega$, we have

$$R_{\mathcal{L}}(h^*) \leq \hat{R}_{\mathcal{L}}(h^*) + \sqrt{\frac{S^2}{2n_{\mathcal{L}}} \ln \frac{2}{\omega}}. \tag{37}$$

**Step 3** We investigate the interplay between old and new classes $T_1$, which is the core of this work. To simplify the symbols, we abbreviate $p_{\mathcal{L}}(l)$ to $p(l)$, $p_{\mathcal{U}}(u)$ to $p(u)$, $p_{\mathcal{U}}(\mathbf{z}, u)$ to $p(\mathbf{z}, u)$, $p_{\mathcal{L}}(\mathbf{z}, l)$ to $p(\mathbf{z}, l)$.

Ineq. (38c) comes from the triangle inequality of norm.

Eq. (38h) is a consequence of that the coupling $\pi((\mathbf{z}_1, u), (\mathbf{z}_2, l))$ between $p(\mathbf{z}_1, u)$ and $p(\mathbf{z}_2, l)$ (i.e., joint distribution of $(\mathbf{z}_1, u)$ and $(\mathbf{z}_2, l)$) is defined to have the marginals $p(\mathbf{z}_1, u)$ and $p(\mathbf{z}_2, l)$. Thus we can recover Eq. (38g), given any or the optimal coupling $\pi$ minimizing the final bound, as stated in Lemma (D.5).

To simplify the process and enhance understanding, we slightly abuse notation by replacing the summation symbol over $u$ and $l$ with the integral symbol. Note that this substitution does not affect the overall correctness of the reasoning logic as the

expectation operator is linear.

$$T_1 = \sum_{u \in \mathcal{U}} p(u) \mathbb{E}_{\mathbf{z} \sim p_u} \mathbb{E}_{\mathbf{z}^+ \sim p_u} L(h^*(\mathbf{z}), h^*(\mathbf{z}^+)) - 2 \sum_{l \in \mathcal{L}} p(l) \mathbb{E}_{\mathbf{z} \sim p_l} L(h^*(\mathbf{z}), \mathbf{y}^l) \tag{38a}$$

$$= \sum_{u \in \mathcal{U}} p(u) \int L(h^*(\mathbf{z}), h^*(\mathbf{z}^+)) p_u(\mathbf{z}) p_u(\mathbf{z}^+) d\mathbf{z} d\mathbf{z}^+ - 2 \sum_{l \in \mathcal{L}} p(l) \int L(h^*(\mathbf{z}), \mathbf{y}^l) p_l(\mathbf{z}) d\mathbf{z} \tag{38b}$$

$$\leq \sum_{u \in \mathcal{U}} p(u) \int \int \left[ L(h^*(\mathbf{z}), \mathbf{y}^u) + L(\mathbf{y}^u, h^*(\mathbf{z}^+)) \right] p_u(\mathbf{z}) p_u(\mathbf{z}^+) d\mathbf{z} d\mathbf{z}^+ - 2 \sum_{l \in \mathcal{L}} p(l) \int L(h^*(\mathbf{z}), \mathbf{y}^l) p_l(\mathbf{z}) d\mathbf{z} \tag{38c}$$

$$= \sum_{u \in \mathcal{U}} p(u) \left[ \int L(h^*(\mathbf{z}), \mathbf{y}^u) p_u(\mathbf{z}) d\mathbf{z} + \int L(\mathbf{y}^u, h^*(\mathbf{z}^+) p_u(\mathbf{z}^+) d\mathbf{z}^+ \right] - 2 \sum_{l \in \mathcal{L}} p(l) \int L(h^*(\mathbf{z}), \mathbf{y}^l) p_l(\mathbf{z}) d\mathbf{z} \tag{38d}$$

$$= 2 \left[ \sum_{u \in \mathcal{U}} p(u) \int L(h^*(\mathbf{z}), \mathbf{y}^u) p_u(\mathbf{z}) d\mathbf{z} - \sum_{l \in \mathcal{L}} p(l) \int L(h^*(\mathbf{z}), \mathbf{y}^l) p_l(\mathbf{z}) d\mathbf{z} \right] \tag{38e}$$

$$= 2 \left[ \sum_{u \in \mathcal{U}} p(u) \int L(h^*(\mathbf{z}), \mathbf{y}^u) p(\mathbf{z}|u) d\mathbf{z} - \sum_{l \in \mathcal{L}} p(l) \int L(h^*(\mathbf{z}), \mathbf{y}^l) p(\mathbf{z}|l) d\mathbf{z} \right] \tag{38f}$$

$$= 2 \left[ \int_{u \in \mathcal{U}} \int_{\mathcal{Z}} L(h^*(\mathbf{z}_1), \mathbf{y}^u) p(\mathbf{z}_1, u) d\mathbf{z}_1 du - \int_{l \in \mathcal{L}} \int_{\mathcal{Z}} L(h^*(\mathbf{z}_2), \mathbf{y}^l) p(\mathbf{z}_2, l) d\mathbf{z}_2 dl \right] \tag{38g}$$

$$= 2 \int_{(\mathcal{Z} \times \mathcal{C})^2} \left[ L(h^*(\mathbf{z}_1), \mathbf{y}^u) - L(h^*(\mathbf{z}_2), \mathbf{y}^l) \right] d\pi((\mathbf{z}_1, u), (\mathbf{z}_2, l)) \tag{38h}$$

$$= 2 \int_{(\mathcal{Z} \times \mathcal{C})^2} \left[ L(h^*(\mathbf{z}_1), \mathbf{y}^u) - L(h^*(\mathbf{z}_2), \mathbf{y}^u) + L(h^*(\mathbf{z}_2), \mathbf{y}^u) - L(h^*(\mathbf{z}_2), \mathbf{y}^l) \right] d\pi((\mathbf{z}_1, u), (\mathbf{z}_2, l)) \tag{38i}$$

$$\leq 2 \int_{(\mathcal{Z} \times \mathcal{C})^2} \left[ |L(h^*(\mathbf{z}_1), \mathbf{y}^u) - L(h^*(\mathbf{z}_2), \mathbf{y}^u)| + |L(h^*(\mathbf{z}_2), \mathbf{y}^u) - L(h^*(\mathbf{z}_2), \mathbf{y}^l)| \right] d\pi((\mathbf{z}_1, u), (\mathbf{z}_2, l)) \tag{38j}$$

$$\leq 2 \int_{(\mathcal{Z} \times \mathcal{C})^2} \left[ rL(h^*(\mathbf{z}_1), h^*(\mathbf{z}_2)) + |L(h^*(\mathbf{z}_2), \mathbf{y}^u) - L(h^*(\mathbf{z}_2), \mathbf{y}^l)| \right] d\pi((\mathbf{z}_1, u), (\mathbf{z}_2, l)) \tag{38k}$$

$$\leq 2 \left[ rS\phi(\lambda) + \int_{(\mathcal{Z} \times \mathcal{C})^2} \left[ r\lambda D(\mathbf{z}_1, \mathbf{z}_2) + |L(h^*(\mathbf{z}_2), \mathbf{y}^u) - L(h^*(\mathbf{z}_2), \mathbf{y}^l)| \right] d\pi((\mathbf{z}_1, u), (\mathbf{z}_2, l)) \right] \tag{38l}$$

$$\leq 2 \left[ rS\phi(\lambda) + \int_{(\mathcal{Z} \times \mathcal{C})^2} \left[ r\lambda D(\mathbf{z}_1, \mathbf{z}_2) + L(\mathbf{y}^u, \mathbf{y}^l) \right] d\pi((\mathbf{z}_1, u), (\mathbf{z}_2, l)) \right] \tag{38m}$$

$$= 2 \left[ rS\phi(\lambda) + W_1^{r\lambda}(p(\mathbf{z}, u), p(\mathbf{z}, l)) \right] \tag{38n}$$

Ineq. (38k) is because of the $r$-Lipschitzness of the discrepancy function $L$ in its first argument (Assumption 3.3).

Ineq. (38l) holds owing to Assumption 3.4 and Assumption 3.2, which lead to the term $rS\phi(\lambda)$ that covers the sample space regions violating the transferable Lipschitzness claimed in Assumption 3.4. A detailed explanation about this can be found in Lemma (D.4). $W_1^{r\lambda}(\hat{p_{\mathcal{U}}}(\mathbf{z}, y), \hat{p_{\mathcal{L}}}(\mathbf{z}, y))$

Ineq. (38m) follows from the triangle inequality of $L$. $W_1^{r\lambda}(p(\mathbf{z}, u), p(\mathbf{z}, l))$ is the discrepancy between the underlying distributions, which can be only estimated via the empirical distributions $\hat{p}(\mathbf{z}, u)$ and $\hat{p}(\mathbf{z}, l)$.

Eq. (38n) is given by Definition 3.1. According to Theorem D.1, it holds that with probability at least $1 - \delta$

$$\begin{aligned} T_1 &\leq 2 \left[ rS\phi(\lambda) + W_1^{r\lambda}(\hat{p}(\mathbf{z}, u), \hat{p}(\mathbf{z}, l)) + W_1^{r\lambda}(p(\mathbf{z}, u), \hat{p}(\mathbf{z}, u)) + W_1^{r\lambda}(p(\mathbf{z}, l), \hat{p}(\mathbf{z}, l)) \right] \\ &= 2 \underbrace{\left[ rS\phi(\lambda) + W_1^{r\lambda}(\hat{p}(\mathbf{z}, u), \hat{p}(\mathbf{z}, l)) + \sqrt{\frac{2}{c'} \log \frac{2}{\delta}} \left( \frac{1}{\sqrt{n_{\mathcal{U}}}} + \frac{1}{\sqrt{n_{\mathcal{L}}}} \right) \right]}_{\text{The estimated interplay between old and new classes}}. \end{aligned} \tag{39a}$$

Here the first inequality is because Wasserstein distance is a valid metric and hence satisfies the triangle inequality. And the second equality is obtained by setting the probability thresholds in Theorem D.1 to $\delta/2$ for $W_1^{r\lambda}(p(\mathbf{z}, u), \hat{p}(\mathbf{z}, u))$ and $W_1^{r\lambda}(p(\mathbf{z}, l), \hat{p}(\mathbf{z}, l))$.

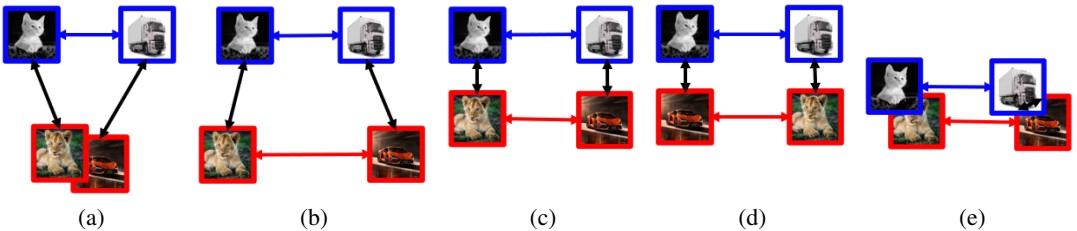

(a)  (b)  (c)  (d)  (e)

Figure 5: An illustrative example of Theorem 3.5 with 2 new classes and 2 new classes. The GCD loss upper bound for all other cases is higher than that of case (c). **(a)** Poor knowledge transfer: The lion is far from cat, and sports cars are far from truck. The blue segment (cat-truck distance) cannot constrain lion-sports car separation. **(b)** Unconstrained new-class separation: Though new classes (lion, sports cars) may naturally separate, this distance is not unconstrained by old-class knowledge. **(c)** Ideal case: Semantically related old-new classes (e.g., cat-lion, truck–sports car) cluster appropriately. **(d)** Misaligned semantics: Global representation distances resemble (c), but semantically incoherent (e.g., cat near sports cars). Thus, the old-new Wasserstein distance (Def. 3.1) is significantly higher. **(e)** Loss of reject capability: New classes (lion/sports cars) overly overlap with old classes (cat/truck). While Wasserstein distance is small, the model fails to reject ambiguous samples (old vs. new class), leading to a large $D_{KL}(P_{\mathcal{C}}\|\bar{h})$ in Eq. (8a).

**Step 4** The probabilistic inequalities (39a) and (37) are then combined. According to the Boole–Fréchet inequalities (Hailperin, 1986), we have, with probability at least $1 - \omega - \delta$,

$$T_1 + R_{\mathcal{L}}(h^*) \leq \text{The sum of two upper bounds.}$$

Substituting the upper bounds of $T_1$ and $R_{\mathcal{L}}(h^*)$ into Eq. (36k) reaches the upper bound on $L_{te}^G(h)$.

**Step 5** In sequel, the upper bound on $\gamma D_{KL}(p_{\mathcal{U}}(c)\|\bar{h}_{\mathcal{U}})$ given in 3.9 is substituted into Eq. 35, which finishes the proof. $\square$

### E.2. An Illustrative Example of Theorem 3.5

Suppose we have a total of four categories: two old classes (cats and trucks) and two new classes (lions and sports cars). After training on feature-label paired data from the two old classes, the classifier has already achieved strong discriminative performance in distinguishing between the old classes (cats and trucks). In the representation space learned using InfoNCE and SupCon contrastive losses, the representations of cats and trucks are also well-separated.

At this stage, in the representation space derived by the encoder, there may exist five typical relationship patterns between old and new classes, as illustrated in Fig. 5. Among these, the third scenario (Fig. 5c) best aligns with human intuition, and it corresponds to the smallest upper bound of the GCD loss in Theorem 3.5 . The other scenarios represent undesirable cases that we aim to avoid. Notably, Fig. 5d may be easily confused with Fig. 5c; however, in Fig. 5d , semantically similar old and new classes are actually not adjacent, which contradicts our intuition and results in a larger GCD loss upper bound.

### E.3. The Proof of Entropy Regularization Trick in Proposition 3.9

**Proposition E.2** (Restate Proposition 3.9). *If $P_{\mathcal{U}}, P_{\mathcal{C}} \in \mathcal{Y}_C$ are uniform distributions over $\mathcal{U}$ and $\mathcal{C}$ respectively, and the data distribution is class-balanced, then it follows that*

$$D_{KL}(P_{\mathcal{U}}\|\bar{h}_{\mathcal{U}})) \leq \frac{1}{b}D_{KL}(P_{\mathcal{C}}\|\bar{h}), \tag{40}$$

*where $b = \Pr(y \in \mathcal{U})$, $\bar{h} = \mathbb{E}_{p(\mathbf{z},y)}h(\mathbf{z})$, and $D_{KL}(P_{\mathcal{C}}\|\bar{h})$ is minimized when the entropy $H(\bar{h}) = -\sum_{c \in \mathcal{C}} \bar{h}(c) \log \bar{h}(c)$ is maximized.*

*Proof.* For the entire underlying data dsitribution $p(\mathbf{z}, y)$, we introduce a binary random variable $t$ to indicate whether data originates from an old or new category. By definition, $p(t = 0) = a$ and $p(t = 1) = 1 - a$ represent the proportions of new and old category data, respectively. We can then decompose the distribution as

$$
\begin{aligned}
p(\mathbf{z}, y) &= p(\mathbf{z}, y|t = 0)p(t = 0) + p(\mathbf{z}, y|t = 1)p(t = 1) \\
&= ap_{\mathcal{U}}(\mathbf{z}, y) + (1 - a)p_{\mathcal{L}}(\mathbf{z}, y),
\end{aligned}
$$

leading to the expansion of $\bar{h}$ below.

$$
\begin{align}
\bar{h} &= \mathbb{E}_{p(\mathbf{z},y)}h(\mathbf{z}) \tag{41a} \\
&= \mathbb{E}_{p(\mathbf{z},y,t)}h(\mathbf{z}) \tag{41b} \\
&= a\sum_{u\in\mathcal{U}}\int_{\mathcal{Z}}p_{\mathcal{U}}(\mathbf{z},y)h(\mathbf{z})d\mathbf{z} + (1-a)\sum_{l\in\mathcal{L}}\int_{\mathcal{Z}}p_{\mathcal{L}}(\mathbf{z},y)h(\mathbf{z})d\mathbf{z} \tag{41c} \\
&= a\mathbb{E}_{p_{\mathcal{U}}(\mathbf{z},y)}h(\mathbf{z}) + (1-a)\mathbb{E}_{p_{\mathcal{L}}(\mathbf{z},l)}h(\mathbf{z}) \tag{41d} \\
&= a\bar{h}_u + (1-a)\bar{h}_l \tag{41e}
\end{align}
$$

Since the data distribution is class-balanced, $a = C_n/C$ and $1 - a = C_o/C$. And current parameteric GCD methods (Wen et al., 2023; Vaze et al., 2023) rarely misclassify old class samples as belonging to new classes, so for any $c \in \mathcal{U}$, $\bar{h}_l(c) = 0$. With this at hand, we readily get that

$$
\begin{align}
D_{KL}(P_{\mathcal{C}}\|\bar{h}) &= \sum_{y\in\mathcal{U}}P_{\mathcal{C}}(y)\log\frac{P_{\mathcal{C}}(y)}{a\bar{h}_u(y) + (1-a)\bar{h}_l(y)} + \sum_{y\in\mathcal{L}}P_{\mathcal{C}}(y)\log\frac{P_{\mathcal{C}}(y)}{\bar{h}(y)} \tag{42a} \\
&= \sum_{y\in\mathcal{U}}\frac{1}{C}\log\frac{\frac{1}{C}}{a\bar{h}_u(y)} + D_{KL}(P_{\mathcal{C}}\|\bar{h}|\mathcal{L}) \tag{42b} \\
&= \sum_{y\in\mathcal{U}}\frac{1}{C}\log\frac{1}{C_n\bar{h}_u(y)} + D_{KL}(P_{\mathcal{C}}\|\bar{h}\mid\mathcal{L}) \tag{42c} \\
&= \sum_{y\in\mathcal{U}}\frac{a}{C_n}\log\frac{1}{C_n\bar{h}_u(y)} + D_{KL}(P_{\mathcal{C}}\|\bar{h}\mid\mathcal{L}) \tag{42d} \\
&= aD_{KL}(P_{\mathcal{U}}\|\bar{h}_u) + D_{KL}(P_{\mathcal{C}}\|\bar{h}\mid\mathcal{L}). \tag{42e}
\end{align}
$$

As $D_{KL}(P_{\mathcal{C}}\|\bar{h}\mid\mathcal{L}) \geq 0$, it follows that

$$
D_{KL}(P_{\mathcal{U}}\|\bar{h}_u) \leq \frac{1}{a}D_{KL}(P_{\mathcal{C}}\|\bar{h}).
$$

Because $P_{\mathcal{C}}$ is a uniform distribution, the minimizer of $D_{KL}(P_{\mathcal{C}}\|\bar{h})$ is a uniform distribution, which, is the maximizer of the entropy $H(\bar{h})$ (Thomas & Joy, 2006). $\qquad\square$

## F. Proofs of Theorems on (Graph) Contrastive Learning Presented in Sec. 4

### F.1. Proof of Theorem 4.1

**Theorem F.1** (Restate Theorem 4.1). *Minimizing InfoNCE loss is equivalent to minimizing the cross-entropy between the subgraph distribution $p(\mathbf{W}_X; \mathbf{B})$, which depends on data augmentation, and the subgraph posterior distribution $p(\mathbf{W}|\mathbf{Z})$.*

*Proof.* The InfoNCE loss is usually formulated as

$$
L_{NCE}(\mathbf{z}_i, \mathbf{z}_j) = -\log\frac{\exp\{\cos(\mathbf{z}_i, \mathbf{z}_j)/\tau\}}{\sum_k \exp\{\cos(\mathbf{z}_i, \mathbf{z}_k)/\tau\}}, \tag{43}
$$

where $\mathbf{z}_i$ and $\mathbf{z}_j$ are the $i$-th and $j$-th rows of the embedding feature matrix $\mathbf{Z}$, $\tau > 0$ is the temperature, and $\cos(\cdot,\cdot)$ is the cosine similarity. To compare $D_{CE}(p(\mathbf{W}_X; \mathbf{B})\|p(\mathbf{W}|\mathbf{Z}))$ with it, we need to get a more detailed form of this cross entropy.

**The unornalized density function of PMRF in $\mathcal{Z}$.**

For probabilistic modeling, we introduce one unnornmlaized density function based on a kernel $k$ for the PMRF in $\mathcal{Z}$.

$$
f_k(\mathbf{Z}, \mathbf{W}) \to \prod_{(i,j)\in[N]^2}k(\mathbf{z}_i - \mathbf{z}_j)^{W_{ij}}. \tag{44}
$$

In a PMRF, the dependencies among random variables are restricted to variable pairs (i.e., $(i, j)$). As a result, the unnormalized density can naturally be modeled as a product form over all variable pairs. Here, a shift-invariant kernel $k(\mathbf{z}_i, \mathbf{z}_j) = k(\mathbf{z}_i - \mathbf{z}_j)$ is used, aligning with the shift-invariant Gaussian kernel in InfoNCE, indicating that the relationships between variables depend solely on their relative positions. Intuitively, larger edge weights $W_{ij}$ and higher node similarity $k(\mathbf{z}_i - \mathbf{z}_j)$ correspond to higher subgraph scores, making such subgraphs more likely to appear.

**The density function of the projection on $\mathcal{S}_1 = (\ker \mathbf{L})^\perp \otimes \mathbb{R}^d$.**

According to Theorem D.6, $f_k(\mathbf{Z}, \mathbf{W})$ is integrable on $\mathcal{S}_1 = (\ker \mathbf{L})^\perp \otimes \mathbb{R}^d$. Thus a valid distribution $p(\mathbf{Z}_1 | \mathbf{W})$ on the the measure space $(\mathcal{S}_1, \mathcal{B}(\mathcal{S}_1), \lambda_{\mathcal{S}_1})$ is accessible, where $\mathcal{B}(\mathcal{S}_1)$ is Borel $\sigma$-algebra on $\mathcal{S}_1$ and $\lambda_{\mathcal{S}_1}$ is the Lebesgue measure on $\mathcal{S}_1$.

$$\Pr(d\mathbf{Z}_1 | \mathbf{W}) = p(\mathbf{Z}_1 | \mathbf{W}) \lambda_{\mathcal{S}_1}(d\mathbf{Z}_1) = \frac{1}{\mathcal{C}_k(\mathbf{W})} f_k(\mathbf{Z}_1, \mathbf{W}) \lambda_{\mathcal{S}_1}(d\mathbf{Z}_1)$$

$$\mathcal{C}_k(\mathbf{W}) = \int f_k(\mathbf{Z}_1, \mathbf{W}) d\lambda_{\mathcal{S}_1}$$

**The density function of the projection on $\mathcal{S}_0 = (\ker \mathbf{L}) \otimes \mathbb{R}^d$.**

Unlike the previous case, if $k$ is a shift-invariant kernel (i.e., as in InfoNCE), $f_k(\mathbf{Z}_1, \mathbf{W})$ is not integrable on $\mathcal{S}_0$, as demonstrated in Lemma D.8. A new valid distribution density is thus required on the subspace $\mathcal{S}_0$. To address this, we design a Borel function $f^\varepsilon(\cdot, \mathbf{W}) \colon \mathbb{R}^{N \times d} \to \mathbb{R}_+$ ($\varepsilon > 0$) that satisfies the following three conditions for validity and applicability:

- For any $\mathbf{Z} \in \mathbb{R}^{N \times d}$, $f^\varepsilon(\mathbf{Z}, \mathbf{W}) = f^\varepsilon(\mathbf{Z}_0, \mathbf{W})$. This ensures the function depends only on the projection of $\mathbf{Z}$ onto $\mathcal{S}_0$ (i.e., $\mathbf{Z}_0$) rather than the entirety of $\mathbf{Z}$.

- For any nonzero positive real number $\varepsilon \in \mathbb{R}_+$, $f^\varepsilon(\cdot, \mathbf{W})$ is integrable on $\mathcal{S}_0$.

- As $\varepsilon \to 0$, $f^\varepsilon(\cdot, \mathbf{W}) \to 1$. This allows $f^\varepsilon$ to become increasingly flexible, even modeling scenarios where all information is entirely lost (i.e., when the function takes the constant value of 1 everywhere)

With this at hand, we define the distribution $p(\mathbf{Z}_0 | \mathbf{W})$ on measure space $(\mathcal{S}_0, \mathcal{B}(\mathcal{S}_0), \lambda_{\mathcal{S}_0})$.

$$\Pr(d\mathbf{Z}_0 | \mathbf{W}) = p(\mathbf{Z}_0 | \mathbf{W}) \lambda_{\mathcal{S}_0}(d\mathbf{Z}_0) = \frac{1}{\mathcal{C}_k(\mathbf{W})} f^\varepsilon(\mathbf{Z}_0, \mathbf{W}) \lambda_{\mathcal{S}_0}(d\mathbf{Z}_0)$$

$$\mathcal{C}^\varepsilon(\mathbf{W}) = \int f_k(\mathbf{Z}_0, \mathbf{W}) d\lambda_{\mathcal{S}_0}$$

**The density function of the projection on the entire space $\mathcal{S}$.**

Since $\mathcal{S}_0$ is orthogonal to $\mathcal{S}_1$, it holds that the product measure on $\mathcal{S} = \mathcal{S}_0 \oplus \mathcal{S}_1$ is the product of two measures respectively defined on $\mathcal{S}_0$ and $\mathcal{S}_1$

$$
\begin{aligned}
\Pr(d\mathbf{Z} | \mathbf{W}) &= \Pr(d\mathbf{Z}_0 | \mathbf{W}) \Pr(d\mathbf{Z}_1 | \mathbf{W}) \\
&= \frac{1}{\mathcal{C}_k(\mathbf{W})} f^\varepsilon(\mathbf{Z}_0, \mathbf{W}) \lambda_{\mathcal{S}_0}(d\mathbf{Z}_0) \frac{1}{\mathcal{C}_k(\mathbf{W})} f_k(\mathbf{Z}_1, \mathbf{W}) \lambda_{\mathcal{S}_1}(d\mathbf{Z}_1) \\
&= \frac{1}{\mathcal{C}_k^\varepsilon(\mathbf{W})} f^\varepsilon(\mathbf{Z}, \mathbf{W}) f_k(\mathbf{Z}, \mathbf{W}) \lambda_{\mathcal{S}}(d\mathbf{Z}) \qquad (45) \\
&= p(\mathbf{Z} | \mathbf{W}) \lambda_{\mathcal{S}}(d\mathbf{Z}),
\end{aligned}
$$

where $\mathcal{C}_k^\varepsilon(\mathbf{W})$ absorbs the normalization constants of two subspace conditionals, and Eq. (45) is due to $f^\varepsilon(\mathbf{Z}, \mathbf{W}) = f^\varepsilon(\mathbf{Z}_0, \mathbf{W})$ and the shift invariant property of $f_k(\mathbf{Z}_1, \mathbf{W}) = f_k(\mathbf{Z}, \mathbf{W})$.

**The prior distribution of $\mathbf{W}$.**

Similar to Definition 2 in (Assel et al., 2022), we introduce the prior distribution for $\mathbf{W}$ that depends on $\pi \in \mathbb{R}_+^{N \times N}$.

$$\Pr_{D,k}^\varepsilon(\mathbf{W}; \pi) \propto \mathcal{C}_k^\varepsilon(\mathbf{W}) \Omega_D(\mathbf{W}) \prod_{(i,j) \in [N]^2} \pi_{ij}^{W_{ij}},$$

where the kernel $k$ should satisfies the conditions in Theorem D.6 and $\Omega_D(\mathbf{W}) \triangleq \prod_i \mathbb{I}(W_{i+} = 1)$ is to filter out subgraphs not consistent with InfoNCE loss, i.e., those subgraphs with more than one outgoing edges for any node.

**The posterior distribution of W.**

With likelihood and prior, we have the posterior of $\mathbf{W}$

$$
\begin{aligned}
\Pr(\mathbf{W}|\mathbf{Z}) &\propto \Pr(\mathbf{Z}|\mathbf{W}) \overset{\varepsilon}{\underset{D,k}{\Pr}}(\mathbf{W}; \pi) \\
&= \frac{1}{\mathcal{C}_k^\varepsilon(\mathbf{W})} f^\varepsilon(\mathbf{Z}, \mathbf{W}) f_k(\mathbf{Z}, \mathbf{W}) \overset{\varepsilon}{\underset{D,k}{\Pr}}(\mathbf{W}; \pi) \\
&\propto f^\varepsilon(\mathbf{Z}, \mathbf{W}) f_k(\mathbf{Z}_1, \mathbf{W}) \Omega_D(\mathbf{W}) \prod_{(i,j) \in [N]^2} \pi_{ij}^{W_{ij}} \\
&= f^\varepsilon(\mathbf{Z}, \mathbf{W}) \Omega_D(\mathbf{W}) \prod_{(i,j) \in [N]^2} \pi_{ij}^{W_{ij}} \prod_{(i,j) \in [N]^2} k(\mathbf{z}_i - \mathbf{z}_j)^{W_{ij}} \\
&\xrightarrow[\varepsilon \to 0]{} \Omega_D(\mathbf{W}) \prod_{(i,j) \in [N]^2} \left[ \pi_{ij} k(\mathbf{z}_i - \mathbf{z}_j) \right]^{W_{ij}}
\end{aligned}
\tag{46}
$$

Denote the sample space of $\mathbf{W}$ as $\mathcal{S}_\mathbf{W} \triangleq \left\{ \mathbf{W} \in \{0,1\}^{N \times N} \mid \forall (i,j) \in [N]^2, W_{ii} = 0 \right\}$, we can normalize Eq. (46) to

$$
\begin{aligned}
\Pr(\mathbf{W}|\mathbf{Z}) &= \frac{\prod_{(i,j) \in [N]^2} \left[ \pi_{ij} k(\mathbf{z}_i - \mathbf{z}_j) \right]^{W_{ij}} \mathbb{I}(W_{i+} = 1)}{\sum_{\mathbf{W} \in \mathcal{S}_\mathbf{W}} \prod_{(i,j) \in [N]^2} \left[ \pi_{ij} k(\mathbf{z}_i - \mathbf{z}_j) \right]^{W_{ij}} \mathbb{I}(W_{i+} = 1)} \\
&= \frac{\prod_{(i,j) \in [N]^2} \left[ \pi_{ij} k(\mathbf{z}_i - \mathbf{z}_j) \right]^{W_{ij}} \mathbb{I}(W_{i+} = 1)}{\prod_{i \in [N]} \sum_{l \in [N]} \pi_{il} k(\mathbf{z}_i - \mathbf{z}_l)} \\
&= \prod_{i \in [N]} \underbrace{\left( \frac{\prod_{j \in [N]} \left[ \pi_{ij} k(\mathbf{z}_i - \mathbf{z}_j) \right]^{W_{ij}} \mathbb{I}(W_{i+} = 1)}{\sum_{l \in [N]} \pi_{il} k(\mathbf{z}_i - \mathbf{z}_l)} \right)}_{\text{The distribution of each row } \mathbf{W}_i} \\
&= \prod_{(i,j) \in [N]^2} \left( \frac{\pi_{ij} k(\mathbf{z}_i - \mathbf{z}_j)}{\sum_{l \in [N]} \pi_{il} k(\mathbf{z}_i - \mathbf{z}_l)} \right)^{W_{ij}} \mathbb{I}(W_{i+} = 1).
\end{aligned}
\tag{47, 48}
$$

The distribution characterized by Eq. (48) actually has the properties that the rows of $\mathbf{W}$ are all independent and the $i$-th row follows the multinomial distribution

$$
\mathbf{W}_i \overset{\perp}{\sim} p(\mathbf{W}_i|\mathbf{Z}) = \mathcal{M} \left( 1, \left[ \frac{\pi_{ij} k(\mathbf{z}_i - \mathbf{z}_j)}{\sum_{l \in [N]} \pi_{il} k(\mathbf{z}_i - \mathbf{z}_l))} \right]_{j \in [N]} \right),
\tag{49}
$$

where the vector in parentheses is the $i$-th row of $\pi \odot \mathbf{K}$ divided by the corresponding row sum and $\mathbf{K}$ is the pairwise relation matrix induced by kernel $k$ in $\mathcal{Z}$.

**The cross entropy between the posterior of W and $p(\mathbf{W}_X; \mathbf{B})$ that depends on the data augmentation strategy.**

Consider the subgraph distribution in the input space. Methods like SimCLR (Chen et al., 2020) and GRACE (Zhu et al., 2020) perform augmentation as sampling subgraph $\mathbf{W}_X$ from the human prior encoded in $\mathbf{B}$, and the out-degrees of all nodes in $\mathbf{W}_X$ are equal to 1, aligned with the a single positive sample setting in GRACE. When both $B_{ij}$ and $W_{ij}$ are large, nodes $i$ and $j$ are more similar and should form a positive pair. To meet this requirement, the subgraph distribution is designed as follows

$$
p(\mathbf{W}_X; \mathbf{B}) \propto \prod_i \mathbb{I}(W_{X,i+} = 1) \prod_{(i,j) \in [N]^2} B_{ij}^{W_{X,ij}}
$$

Let the sample space of $\mathbf{W}_X$ be $\mathcal{S}_\mathbf{W} \triangleq \left\{ \mathbf{W} \in \{0,1\}^{N \times N} \mid \forall (i,j) \in [N]^2, W_{ii} = 0 \right\}$, we can normalized the above

distribution

$$p(\mathbf{W}_X; \mathbf{B}) = \frac{\prod_{(i,j)\in[N]^2} B_{ij}^{W_{X,ij}} \mathbb{I}(W_{X,i+} = 1)}{\sum_{\mathbf{w}_X \in \mathcal{S}_\mathbf{w}} \prod_{(i,j)\in[N]^2} B_{ij}^{W_{X,ij}} \mathbb{I}(W_{X,i+} = 1)}$$

$$= \frac{\prod_{(i,j)\in[N]^2} B_{ij}^{W_{X,ij}} \mathbb{I}(W_{X,i+} = 1)}{\prod_{i\in[N]} \sum_{l\in[N]} B_{il}}$$

$$= \prod_{i\in[N]} \underbrace{\left( \frac{\prod_{j\in[N]} B_{ij}^{W_{X,ij}} \mathbb{I}(W_{X,i+} = 1)}{\sum_{l\in[N]} B_{il}} \right)}_{\text{The distribution of each row } \mathbf{W}_{X,i}}$$

$$= \prod_{(i,j)\in[N]^2} \left( \frac{B_{ij}}{\sum_{l\in[N]} B_{il}} \right)^{W_{X,ij}} \mathbb{I}(W_{X,i+} = 1),$$

which means

$$\mathbf{W}_{X,i} \overset{\perp}{\sim} p(\mathbf{W}_{X,i}; \mathbf{B}) = \mathcal{M}\left( 1, \left[ \frac{B_{ij}}{\sum_{l\in[N]} B_{il}} \right]_{j\in[N]} \right). \tag{50}$$

Now, let us formalize the cross-entropy. Since each row of $\mathbf{W}$ is independent, we have $p(\mathbf{W}|\mathbf{Z}) = \prod_{i\in[N]} p(\mathbf{W}_i|\mathbf{Z})$. Similarly, during data augmentation, the sampling of positive samples for each instance $i$ is independent, leading to $p(\mathbf{W}_X; \mathbf{B}) = \prod_{i\in[N]} p(\mathbf{W}_{X,i}; \mathbf{B})$. Consequently, the difference between matrix distributions can be decomposed row-wise according to Lemma D.7.

$$D_{CE}(p(\mathbf{W}_X; \mathbf{B}) \| p(\mathbf{W}|\mathbf{Z})) = \sum_{i=1}^{N} D_{CE}(p(\mathbf{W}_{X,i}; \mathbf{B}) \| p(\mathbf{W}_i|\mathbf{Z}))$$

$$= \sum_{i=1}^{N} D_{CE}(P_i \| Q_i)$$

$$= -\sum_{i=1}^{N} \sum_{j\neq i}^{N} P_{ij} \log Q_{ij}$$

$$= -\sum_{i=1}^{N} \sum_{j\neq i}^{N} \frac{B_{ij}}{\sum_{l\in[N]} B_{il}} \log \frac{\pi_{ij} k(\mathbf{z}_i - \mathbf{z}_j)}{\sum_{l\in[N]} \pi_{il} k(\mathbf{z}_i - \mathbf{z}_l)} \tag{51}$$

If the parameter $\pi = \mathbf{1}$ for the prior of $\mathbf{W}$ and $k$ is the Gaussian kernel, then Eq. (51) retrieves the famous InfoNCE loss.

$$D_{CE}(p(\mathbf{W}_X; \mathbf{B}) \| p(\mathbf{W}|\mathbf{Z})) = -\sum_{i=1}^{N} \sum_{j\neq i}^{N} \frac{B_{ij}}{\sum_{l\in[N]} B_{il}} \log \frac{\exp\{\cos(\mathbf{z}_i, \mathbf{z}_j)/\tau\}}{\sum_{l\in[N]} \exp\{\cos(\mathbf{z}_i, \mathbf{z}_l)/\tau\}},$$

where $\frac{B_{ij}}{\sum_{l\in[N]} B_{il}}$ is the probability of sampling $j$ as the positive sample for $i$, and the $-\log \frac{\exp\{\cos(\mathbf{z}_i, \mathbf{z}_j)/\tau\}}{\sum_{l\in[N]} \exp\{\cos(\mathbf{z}_i, \mathbf{z}_l)/\tau\}}$ is the InfoNCE loss on positive pair $(i, j)$. □

### F.2. Proof of Corollary 4.2

**Corollary F.2** (Restate Theorem 4.2). *Minimizing SupCon loss is equivalent to minimizing the cross-entropy stated in Theorem 4.1.*

*Proof.* The overall proof approach is similar to the equivalence proof between InfoNCE and $D_{CE}(p(\mathbf{W}_X; \mathbf{B}) \| p(\mathbf{W}|\mathbf{Z}))$ (Theorem 4.1). The key difference lies in how multiple positive pairs introduced by label information are handled.

It is noted that the out-degree of each node in the sampled subgraph does not exceed $m_i$. If no duplicate nodes are sampled, the out-degree equals $m_i$; otherwise, it is less than $m_i$. For node $i$, the augmentations of $m_i$ same-class samples are independent of each other. Thus, under the SupCon setting, the current distribution of each row of $\mathbf{W}$ can be expressed as

$$
\begin{aligned}
p_{sc}(\mathbf{W}_{X,i}; \mathbf{B}) &= \prod_{r=1}^{m_i} p(\mathbf{W}_{X,i}; \mathbf{B}) \\
p_{sc}(\mathbf{W}_i|\mathbf{Z}) &= \prod_{r=1}^{m_i} p(\mathbf{W}_i|\mathbf{Z}),
\end{aligned}
$$

where $p(\mathbf{W}_{X,i}; \mathbf{B})$ and $p(\mathbf{W}_i; \mathbf{Z})$ correspond to Eq. (50) and Eq. (49), respectively. Consequently, under the SupCon setting, Eq. (51) (with the prior parameter $\pi = \mathbf{1}$) becomes

$$
\begin{aligned}
D_{CE}(p(\mathbf{W}_X; \mathbf{B})\|p(\mathbf{W}|\mathbf{Z})) &= \sum_{i=1}^{N} D_{CE}(p_{sc}(\mathbf{W}_{X,i}; \mathbf{B})\|p_{sc}(\mathbf{W}_i|\mathbf{Z})) \\
&= \sum_{i=1}^{N} \sum_{r=1}^{m_i} D_{CE}(p(\mathbf{W}_{X,i}; \mathbf{B})\|p(\mathbf{W}_i|\mathbf{Z})) \quad (52) \\
&= -\sum_{i=1}^{N} \sum_{r=1}^{m_i} \sum_{j\neq i}^{N} P_{ij} \log Q_{ij} \\
&= -\sum_{i=1}^{N} \underbrace{\left[ \sum_{r=1}^{m_i} \sum_{j\neq i}^{N} \frac{B_{ij}}{\sum_{l\in[N]} B_{il}} \log \frac{k(\mathbf{z}_i - \mathbf{z}_j)}{\sum_{l\in[N]} k(\mathbf{z}_i - \mathbf{z}_l))} \right]}_{A}, \quad (53)
\end{aligned}
$$

where Eq. (52) is due to Lemma D.7 and $A$ can be regarded as the population SupCon loss for anchor node $i$. Denote by $j_r$ the positive node sampled in the $r$-th augmentation step. Then according to the non-negativity of cross entropy and $-\log \frac{k(\mathbf{z}_i - \mathbf{z}_{j_r})}{\sum_{l\in[N]} k(\mathbf{z}_i - \mathbf{z}_l))}$, it follows that

$$
\begin{aligned}
D_{CE}(p(\mathbf{W}_X; \mathbf{B})\|p(\mathbf{W}|\mathbf{Z})) &\geq -\sum_{i=1}^{N} \sum_{r=1}^{m_i} \log \frac{k(\mathbf{z}_i - \mathbf{z}_{j_r})}{\sum_{l\in[N]} k(\mathbf{z}_i - \mathbf{z}_l))} \\
&\geq -\sum_{i=1}^{N} \frac{1}{m_i} \sum_{r=1}^{m_i} \log \frac{k(\mathbf{z}_i - \mathbf{z}_{j_r})}{\sum_{l\in[N]} k(\mathbf{z}_i - \mathbf{z}_l))},
\end{aligned}
$$

which exactly coincides with the empirical SupCon loss (Khosla et al. (2020); Eq. (2) ) for anchor node $i$. □

### F.3. Proof of Theorem 4.3

**Theorem F.3** (Restate Theorem 4.3)**.** *The InfoNCE or SupCon loss optimization is equivalent to minimizing $D_{CE}(p(\mathbf{W}_X; \mathbf{B})\|p(\mathbf{W}|\mathbf{Z}))$ when the conditional distribution of category centers $p^\varepsilon(\mathbf{Z}_0|\mathbf{W})$ diffuse uninformatively.*

*Proof.* In the proof of Theorem F.1, the equivalence condition between InfoNCE loss optimization and the minimization of $D_{CE}(p(\mathbf{W}_X; \mathbf{B})\|p(\mathbf{W}|\mathbf{Z}))$ is that the distribution $p^\varepsilon(\mathbf{Z}_0|\mathbf{W})$ diffuses uninformatively. The proof of Corollary F.2 follows a similar process to that of Theorem F.1, where the equivalence condition between SupCon loss optimization and the minimization of $D_{CE}(p(\mathbf{W}_X; \mathbf{B})\|p(\mathbf{W}|\mathbf{Z}))$ is also that the distribution $p^\varepsilon(\mathbf{Z}_0|\mathbf{W})$ diffuses uninformatively. Therefore, Theorem 4.3 holds. □

## G. Supplementary Details of the Experiments

### G.1. Evaluation Protocol

**The Conventional VGCD metrics** The standard evaluation protocol used in VGCD (Vaze et al., 2022) employs clustering accuracy to measure the overall performance (All ACC), new-class performance (New ACC) and old-class performance (Old ACC).

Let $\mathbf{W} = [w_{ij}]$ be the $C \times C$ confusion matrix, where $w_{ij}$ is the count of samples with true label $j$ and predicted label $i$. Clustering Accuracy is computed by solving a linear sum assignment (LSA) problem (Lovasz, 1986; Crouse, 2016), mapping each ground truth $c$ to a predicted label $\mathbf{a}(c)$ to maximize correct predictions among $n$ (test) nodes.

$$\text{All ACC} = \max_{\mathbf{a}} \frac{1}{n} \sum_{c=1}^{C} \mathbf{W}_{\mathbf{a}(c),c} \tag{54}$$

New ACC and Old ACC are computed as

$$\text{New ACC} = \frac{\sum_{u \in \mathcal{U}} \mathbf{W}_{\mathbf{a}(u),u}}{\sum_{u \in \mathcal{U}} \sum_{c=1}^{C} \mathbf{W}_{c,u}} \tag{55}$$

$$\text{Old ACC} = \frac{\sum_{l \in \mathcal{L}} \mathbf{W}_{\mathbf{a}(l),l}}{\sum_{l \in \mathcal{L}} \sum_{c=1}^{C} \mathbf{W}_{c,l}} \tag{56}$$

**The Rectified GCD metrics** We argue that the traditional GCD metrics introduced above fail to account for the adverse effects of confusion between known and novel categories. Under no circumstances should novel categories be mapped to known categories, as such incorrect mappings violate the core objective of the GCD task: ensuring that known categories are accurately identified as their true labels while distinguishing novel categories. Therefore, we contend that traditional metrics do not fully and faithfully reflect the performance of GCD. Regarding the old-class performance, the LSA within $\mathcal{C}$ may assign true old classes to new classes, which is undesirable. Hence, we adopt the conventional classification accuracy as a rectified version.

$$\textbf{Old RACC} = \frac{\sum_{l \in \mathcal{L}} \mathbf{W}_{l,l}}{\sum_{l \in \mathcal{L}} \sum_{c=1}^{C} \mathbf{W}_{c,l}} \tag{57}$$

Respecting the new-class performance, the computation of New ACC may result in cases where a true old class $l$ is assigned to a new label $u$, violating the intended separation of known and novel categories expected in GCD. To address this, we propose a Rectified Clustering Accuracy (RACC) metric to better evaluate the performance on novel categories. Without loss of generality, assume the first $C_o$ classes are the old categories and the remaining are the new categories. We extract the bottom-right $C_o \times C_o$ submatrix $\mathbf{W}^{(new)}$ of the confusion matrix $\mathbf{W}$, which corresponds to the confusion among new categories. Based on $\mathbf{W}^{(new)}$, we solve the Linear Sum Assignment (LSA) problem within the new categories, mapping each ground truth new label $u$ to a predicted new label $\mathbf{b}(u)$. The Rectified metric is then defined as

$$\textbf{New RACC} = \max_{\mathbf{b}} \frac{1}{\sum_{u=C_o+1}^{C} \sum_{c=1}^{C} \mathbf{W}_{c,u}} \sum_{u=1}^{C-C_o} \left[ \mathbf{W}^{(new)} \right]_{\mathbf{b}(u),u}. \tag{58}$$

It is important to note that when novel-class samples are misclassified as old classes, $\mathbf{W}^{(new)}$ loses many samples and $\sum_{u=1}^{C-C_o} \left[ \mathbf{W}^{(new)} \right]_{\mathbf{b}(u),u}$ decreases significantly, resulting in a low New RACC. This makes New RACC a more faithful reflection of performance on novel categories in the GCD context compared to New ACC. Given that performance on both known and novel categories is critical for GCD, and the harmonic mean penalizes large disparities between values, we replace All ACC with the **H**armonic mean of Old **R**ACC and New RACC **S**cores (**HRScore**), to evaluate the overall GCD performance:

$$\textbf{HRScore} = \begin{cases} 0 & \text{if } \textbf{Old RACC=0 or New RACC=0} \\ \frac{2}{\frac{1}{\text{Old RACC}} + \frac{1}{\text{New RACC}}} & \text{otherwise} \end{cases} \tag{59}$$

Reject ACC is computed by merging all "old classes" into a single class (Known) and all "new classes" into another class (Unknown), constructing a $2 \times 2$ confusion matrix $\mathbf{W}^{(r)}$ and calculating the proportion of the sum of its diagonal elements to the total sum of all elements.

$$\textbf{Reject ACC} = \frac{\sum_{i=1}^{2} \mathbf{W^{(r)}}_{i,i}}{\sum_{i=1}^{2} \sum_{j=1}^{2} \mathbf{W^{(r)}}_{i,j}} \tag{60}$$

### G.2. Hardware and Software Environment

The experiments were carried out across two different systems. The first system runs Ubuntu 22.04 and is equipped with an RTX 4090 GPU (24GB), an Intel i7-12700 CPU, and 64GB of RAM. The second system, which uses Ubuntu 20.04,

features an RTX 4090 GPU (24GB), dual Intel Xeon Gold 6240C processors, and 126GB of RAM. Both systems have the same Conda environment, which includes PyTorch 2.5 (Paszke et al., 2017) and PyG 2.5 (Fey & Lenssen, 2019), all built on CUDA 12.1.

### G.3. Experimental Details for GCD Upper Bound Theory (Sec. 3.4)

**Synthetic Node Embedding Datasets**    In the 2D plane, we generate a square with radius $r_1$, placing its four vertices $\mu_c(c \in [1:4])$ as the centroids of four new classes along the coordinate axes. Then, centered at $S_1$, we generate another square $S_2$ with radius $r_2$ ($> r_1$), and use its vertices $\mu_c(c \in [5:8])$ as the centroids of four old classes. For each class $c$, we sample 200 points from $\mathcal{N}(\mu_c, \sigma_c^2 \mathbf{I})$ as the sample embeddings. By controlling $r_1$, $\sigma_c$, and $r_1/r_2$, we control the confusion between new and old classes. We generate three embedding datasets $\mathcal{D}_1^z \sim \mathcal{D}_3^z$, shown in Fig. 1a-1c, where the corresponding Wasserstein distances satisfy $W_1 > W_2 > W_3$, and the centroids of the new (✖) and old (★) classes are marked by (✖, ✖, ✖, ✖) and (★, ★, ★, ★), respectively.

Specifically, for these three embedding datasets, the raw data is generated with the following settings: for all new classes, $\sigma_c^2 = 15$, and for old classes, $\sigma_c^2 = 10$. The other parameters for each dataset are as follows:

- For $\mathcal{D}_1^z$: $r_1 = 8, r_1/r_2 = 0.5$.

- For $\mathcal{D}_2^z$: $r_1 = 16, r_1/r_2 = 1$ .

- For $\mathcal{D}_3^z$: $r_1 = 16, r_1/r_2 = 2$.

Subsequently, all data undergoes Z-Score Normalization to ensure consistency in numerical ranges across datasets. The entire node set is stratified into training, validation, and test subsets in a 2:2:6 ratio, which are subsequently used for training and evaluation, as described in Sec. 6.1. Note that the old classes are designed to be easily distinguishable due to the large inter-class distances. This enables us to focus on understanding how the old-and-new relationship impacts GCD, with excluding the influence of mixing old classes.

**Model and Training Setup**    In the experiments on synthetic node embedding datasets, the setup is as follows:

- **Classifier**: A 2-layer MLP classifier with an input dimension of 2, a hidden layer dimension of 24, and an output dimension equal to the number of classes (8).

- **Training Loss**: The model is trained using Cross-Entropy (**CE**) loss on labeled samples and Entropy Regularization (**ER**) on all samples. The overall training objective is $\mathcal{L}_{MLP} = (1 - \alpha_2)L_{ER} + \alpha_2 L_{CE}$, where $\alpha_2 = 0.35$.

- **Optimization**: We train the model for 1000 epochs using the Adam optimizer (Kingma & Ba, 2017) with a learning rate of 0.01.

We report the performance with the model weights from the final training epoch. The main results are shown in Fig. 1e. To ensure consistency, the model parameters are initialized with the same weights at the start of training for each dataset.

### G.4. Experimental Details for GCL Theory (Sec. 4.3)

**Synthetic CSBM Graph Datasets**    In the 2D plane, we first generate node features and labels following the process described for $\mathcal{D}_1^z$ and $\mathcal{D}_3^z$ in Sec. G.3. Next, intra-class nodes are connected using a Bernoulli distribution with a probability of $p = 0.05$, while inter-class nodes are connected with a probability of $q = 0.001$, resulting in the adjacency matrix. This procedure produces the CSBM (Deshpande et al., 2018) graph datasets $\mathcal{D}_1$ and $\mathcal{D}_3$, whose details are summarized in Table 3. For each dataset, the entire node set is stratified into training, validation, and test subsets in a 2:2:6 ratio, which are then used for training and evaluation, as described in Sec. 6.1.

**Model and Training Setup for Verifying Theorem 4.3**    Theorem 4.3 discusses the uncontrollability of the global structure between categories in the node embedding space learned through InfoNCE and SupCon losses. To validate this theorem, we train an encoder using GRACE and GRACE-SC. The setup is as follows:

- **Encoder:** The 2-layer GCN encoder has input, hidden, and output dimensions of 2, 36, and 2, respectively.

Table 3: The statistics of CSBM graphs. $W^x$ is the Wasserstein distance between old and new classes in the input node feature space.

| Dataset | #Nodes | #Edges | #Features | #Classes | #Old Classes | Homophily Ratio | $W^x$ |
|---------|--------|--------|-----------|----------|--------------|-----------------|-------|
| $\mathcal{D}_1$ | 1600 | 18014 | 2 | 8 | 4 | 0.88 | 0.33 |
| $\mathcal{D}_3$ | 1600 | 18356 | 2 | 8 | 4 | 0.89 | 0.19 |

- **Augmentation**: The graph data augmentation module randomly drops 20% of the edges and injects Gaussian noise with a standard deviation of 0.2 and a mean of zero into the node features.

- **Training Loss**: The training loss is a weighted combination of InfoNCE and SupCon losses, i.e., $\mathcal{L}_{GCL} = (1 - \alpha_2)L_{NCE} + \alpha_2 L_{SC}$, and we set $\alpha_2 = 0.35$.

- **Optimization**: The model is trained over 1000 epochs using the Adam optimizer, set at a learning rate of 0.01.

After training, we utilize the GCN encoder to generate embeddings for all nodes and visualize them. Figs 2b and 2c show the embedding spaces obtained on the dataset $\mathcal{D}_1$ using random seeds 2050 and 800, respectively, under the same model weight initialization. Figs. 2e and 2f present the results on $\mathcal{D}_3$. The analysis in Sec. 4.3 confirms the randomness of global structure.

**Model and Training Setup for Verifying the impact of GCN encoder on GCD**  Sec. 4.2 examines the impact of the GCN encoder's smoothing effect on GCD, focusing on the undesired mixing of categories. While replacing the GCN encoder with an MLP in a control experiment eliminates the effects of GCN's local smoothing, it completely discards graph structural information, which would distort the contrastive results. Therefore, we opt for the high-pass GNN, GPR (Chien et al., 2021), which preserves structural information while escaping from the local smoothing effect[2]. The setup is:

- **Encoder:** The 2-layer GCN encoder has input, hidden, and output dimensions of 2, 36, and 2, respectively. The GPR encoder is chained by two parts: the first is a 2-layer MLP feature extractor with dimensions (2, 36, 2) that output $\mathbf{X}'$; the second is a GPR convolution layer formulated as $\mathbf{Z} = \sum_{k=0}^{10} \theta_k \left( (\mathbf{D} + \mathbf{I})^{-1/2} (\mathbf{A} + \mathbf{I})(\mathbf{D} + \mathbf{I})^{-1/2} \right)^k \mathbf{X}'$, where $\mathbf{D}$ is the degree matrix of $\mathbf{A}$ and $\theta_k$ is initialized with Personal Page Rank (Jeh & Widom, 2003).

- **Classifier:** The MLP classifier has input, hidden, and output dimensions of 2, 24, and 2, respectively.

- **Augmentation**: The graph augmentation process involves randomly removing 20% of edges and adding zero-mean Gaussian noise with a standard deviation of 0.2 to node features.

- **Training Loss**: The training loss is $\mathcal{L}_{SimGCD} = (1 - \alpha_2)(L_{NCE} + L_{SD} + \alpha_1 L_{ER}) + \alpha_2(L_{SC} + L_{CE})$. We set $\alpha_2 = 0.35$ and $\alpha_1 = 2$, as is commonly done by default in many baselines (Vaze et al., 2022; Wen et al., 2023).

- **Optimization**: Training involves 1000 epochs with the Adam optimizer, employing a learning rate of 0.01.

**G.5. Experimental Details for SWIRL on Synthetic CSBM Graphs (Sec. 5.1)**

SWIRL is a novel GCL method, inspired by the GCD theory for understanding parametric GCD methods (Theorem 3.5) and designed to enhance the GCL module in the baseline parametric GCD method, SimGCD. To visually demonstrate that SWIRL learns an embedding space more beneficial for GCD—one that better aligns with the embedding conditions **Lessons 1** and **2** derived from Theorem 3.5—we conduct comparative experiments on the CSBM graphs $\mathcal{D}_1$ and $\mathcal{D}_3$, evaluating SimGCD, SimGCD-GPR, and SWIRL. The learned embedding spaces for the three methods are shown in Fig. 3, with the metrics presented in Table 1. For SimGCD and SimGCD-GPR, the experimental setup is as described in Sec. G.4. The setup for SWIRL is as follows:

- **Encoder:** The 2-layer GCN encoder has input, hidden, and output dimensions of 2, 36, and 2, respectively.

---

[2]By analyzing the learned GPR coefficients from our experiments, we found that the two GPR encoders on $\mathcal{D}_1$ and $\mathcal{D}_3$ effectively act as high-pass graph filters, distinguishing them from the low-pass and locally smoothing nature of GCN.

- **Classifier**: The MLP classifier has 2 dimensions for the input layer, 24 dimensions for the hidden layer, and 2 dimensions for the output layer.

- **Augmentation**: The graph data augmentation component randomly removes 20% of the edges and introduces Gaussian noise with a mean of zero and a standard deviation of 0.2 into the node features.

- **Training Loss**: The training loss is $\mathcal{L}_{SW} = (1 - \alpha_2)(L_{NCE} + \beta_1 L_{SW} + \alpha_1 L_{ER}) + \alpha_2 L_{CE}$. We set $\alpha_2 = 0.35$ and $\alpha_1 = 2$, which are commonly used values in many baseline models (Vaze et al., 2022; Wen et al., 2023), and choose $\beta_1 = 20$.

- **Optimization**: We use the Adam optimizer with a learning rate of 0.01 to train the model for 1000 epochs.

### G.6. Experimental Details for Real-world Experiments (Sec. 6)

**Real-world Graph Datasets**   We consider public graph datasets: Cora, Citeseer (Sen et al., 2008), Wiki (Cao et al., 2016), Amazon-Photo (A-Photo), and Amazon-Computers (A-Computers) (Shchur et al., 2019). The statistics of these graphs are summarized in Table 4. For each real-world graph dataset, we use the same hyperparameter settings and conduct five experiments with different random seeds (20, 21, 22, 23, 24), reporting the mean and standard deviation of the results.

Table 4: The data statistics of the used real-world graph datasets.

| Dataset | #Nodes | #Edges | #Features | #Classes | #Old Classes | #New Classes | Homophily Ratio |
|---|---|---|---|---|---|---|---|
| Cora | 2485 | 5069 | 1433 | 7 | 3 | 4 | 0.81 |
| Citeseer | 2110 | 3668 | 3703 | 6 | 3 | 3 | 0.74 |
| Wiki | 2405 | 23192 | 4973 | 17 | 8 | 9 | 0.61 |
| Amazon-Photo | 7650 | 238162 | 745 | 8 | 4 | 4 | 0.83 |
| Amazon-Computers | 13752 | 491722 | 767 | 10 | 5 | 5 | 0.78 |

**Implementation Details**   We begin by provide additional GGCD details uncovered earlier, and then provide the hyper-parameter values or the search space in Table 5. For each method, we use Optuna (Akiba et al., 2019) to perform 40 optimization runs, with the Old RACC on the validation node set as the objective for hyperparameter search. Note that since only the old classes are known, we cannot use the New RACC or HRScore on validation set to guide the hyperparameter search. Additionally, we employ early stopping for each experimental run. If the validation Old RACC does not improve for $\frac{2}{3}$Max Epochs consecutive epochs, training is terminated immediately. This approach not only reduces experimental costs but also accounts for findings by Wen et al. (2023), which suggest that prolonged training can degrade Old Capability.

- SS-KM (Vaze et al., 2022): The baseline directly applies SS-KM to the node features, utilizing the K-means++ initialization strategy for cluster centroids (Arthur & Vassilvitskii, 2007). The number of clusters is set to the number of categories $C$, the distance metric is Euclidean distance. The maximum number of iterations *max_iter=200*, the number of centroid reinitializations *n_init=10 , and* the convergence check condition is *tolerance=1e-6*.

- GCN (Kipf & Welling, 2017): The output layer dimension of the GCN baseline is set to the total number of classes $C$. Since the Cross-Entropy (CE) loss can only leverage the supervision from the $C_o$ old classes, it is unable to handle novel categories, making it an ineffective baseline. Therefore, we incorporate an Entropy Regularization (ER) loss and optimize the objective

$$\mathcal{L}_{GCN} = (1 - 0.35) \cdot 4L_{ER} + 0.35L_{CE}.$$

- Vanilla GCD (Vaze et al., 2022): We use the Graph adaptation variant from Sec. B. Specifically, we first perform semi-supervised contrastive training based on GRACE and GRACE-SC to obtain the GCN encoder, and then apply SS-KM to make predictions based on the encoder's output. The number of clusters for SS-KM is set to the number of categories $C$, with K-means++ initialization and the following parameters: *tolerance = 1e-5*, *max_iter = 100*, and *n_init = 1*. The training objective is

$$\mathcal{L}_{VanillaGCD} = (1 - 0.35)L_{NCE} + 0.35L_{SC}, \qquad \alpha_2 > 0.$$

- UNO+ (Fini et al., 2021; Vaze et al., 2022): The optimal transport problem is solved using Sinkhorn (Cuturi, 2013) to compute the optimal node-to-prototype probability pseudo-label matrices $\mathbf{Z}^{(1)}$ and $\mathbf{Z}^{(2)}$ for the view embeddings $\mathbf{Z}^{(1)}$ and $\mathbf{Z}^{(2)}$. For old-class nodes with known labels, their pseudo-labels are forcibly replaced with the corresponding one-hot vectors of the true labels. Then, a low-temperature cross-entropy loss is applied to supervise the predictions from the other view. The loss function is

$$\mathcal{L}_{UNO+} = \frac{\alpha_2}{2} \sum_{u=1}^{n} D_{CE}\left(\left[\mathbf{Q}^{(2)}\right]_u, \left[\mathbf{Z}^{(1)}\right]_u\right) + D_{CE}\left(\left[\mathbf{Q}^{(1)}\right]_u, \left[\mathbf{Z}^{(2)}\right]_u\right).$$

- SimGCD (Wen et al., 2023): The original version of SimGCD, designed for VGCD, uses a prototype classifier. However, we found that a two-layer MLP generally performs better on graph data, particularly on the CSBM graph datasets $\mathcal{D}_1$ and $\mathcal{D}_3$ that we generated. The training loss for SimGCD is

$$\mathcal{L}_{SimGCD} = (1 - 0.35)(L_{NCE} + L_{SD} + 2L_{ER}) + 035(L_{SC} + L_{CE}).$$

- SWIRL (Ours): SWIRL is a novel Graph Contrastive Learning (GCL) method designed for Parametric GGCD. Unlike the classic Parametric method SimGCD, SWIRL eliminates the use of Self Distillation (SD) and SupCon losses, while retaining the InfoNCE loss to optimize the local structure of the embedding space. The SWIRL loss is introduced to regulate the global structure. The number of clusters used in the SS-KM for computing the SWIRL loss is set to $K(> C)$, with *tolerance = 1e-4, max_iter = 20,* and *n_init = 1*. The training loss for SWIRL is

$$\mathcal{L}_{SW} = (1 - 0.35)(L_{NCE} + \beta_1 L_{SW} + 2L_{ER}) + 0.35 L_{CE}.$$

Table 5: Hyperparameters of GGCD methods and the corresponding values or search spaces

| Common Hyperparameters | | |
|---|---|---|
| **Group** | **Hyperparameter** | **Value or Search Space** |
| Optimization | Learning Rate | [0.001, 0.005, 0.01, 0.05, 0.1] |
| | Dropout | [0.1, 0.2, 0.3, 0.4, 0.5, 0.6, 0.7, 0.8] |
| | Weight Decay | [0.0, 5e-5, 1e-4, 5e-4, 1e-3, 5e-3, 1e-2] |
| | Max Epochs | [200, 400, 800] |
| Neural Network Arch. | GCN Encoder Layer | [2, 3, 4] |
| | GCN Encoder Activation | ["relu", "prelu"] |
| | Hidden Dim. | [64, 128, 256, 512, 1024] |
| | MLP Projector Layer | 2 |
| | MLP Classifier Layer | 2 |
| Graph Augmentation | Edge Removal Rate | [0, 0.1, 0.2, 0.3, 0.4, 0.5, 0.6] |
| | Feature Masking Rate | [0, 0.1, 0.2, 0.3, 0.4, 0.5, 0.6] |
| Losses | InfoNCE Loss Temp. $\tau$ | [0.1:1.5], step=0.1 |
| | SupCon Loss Temp. | 0.07 |
| | ER Loss Temp. | 0.1 |
| | ER Weight $\alpha_1$ | 2 |
| | Supervised Loss Weight $\alpha_2$ | 0.35 |
| Some Specific Hyperparameters | | |
| **Group** | **Hyperparameter** | **Value or Search Space** |
| SimGCD | SD Loss Teacher Temp. | 0.04 |
| | SD Loss Student Temp. | 0.1 |
| | SD Loss Warmup Teacher Temp. | 0.07 |
| | SD Loss Warmup Epochs | 30 |
| GCN | ER Loss Weight $\alpha_1$ | 4 |
| UNO+ | Temp. | 0.1 |
| | Sinkhorn Num Iterations | 3 |
| | Sinkhorn Epsilon | 0.05 |
| SWIRL | SWIRL Loss Weight $\beta_1$ | [1, 50] |
| | Repulsion Force Scale $s$ | [0.00001, 0.0001, 0.001, 0.01] |
| | Repulsion Force Degree $t_1 : t_6$ | $t_i = 1 + 0.1i$ |
| | Prototype Number $K$ | [20, 200] |

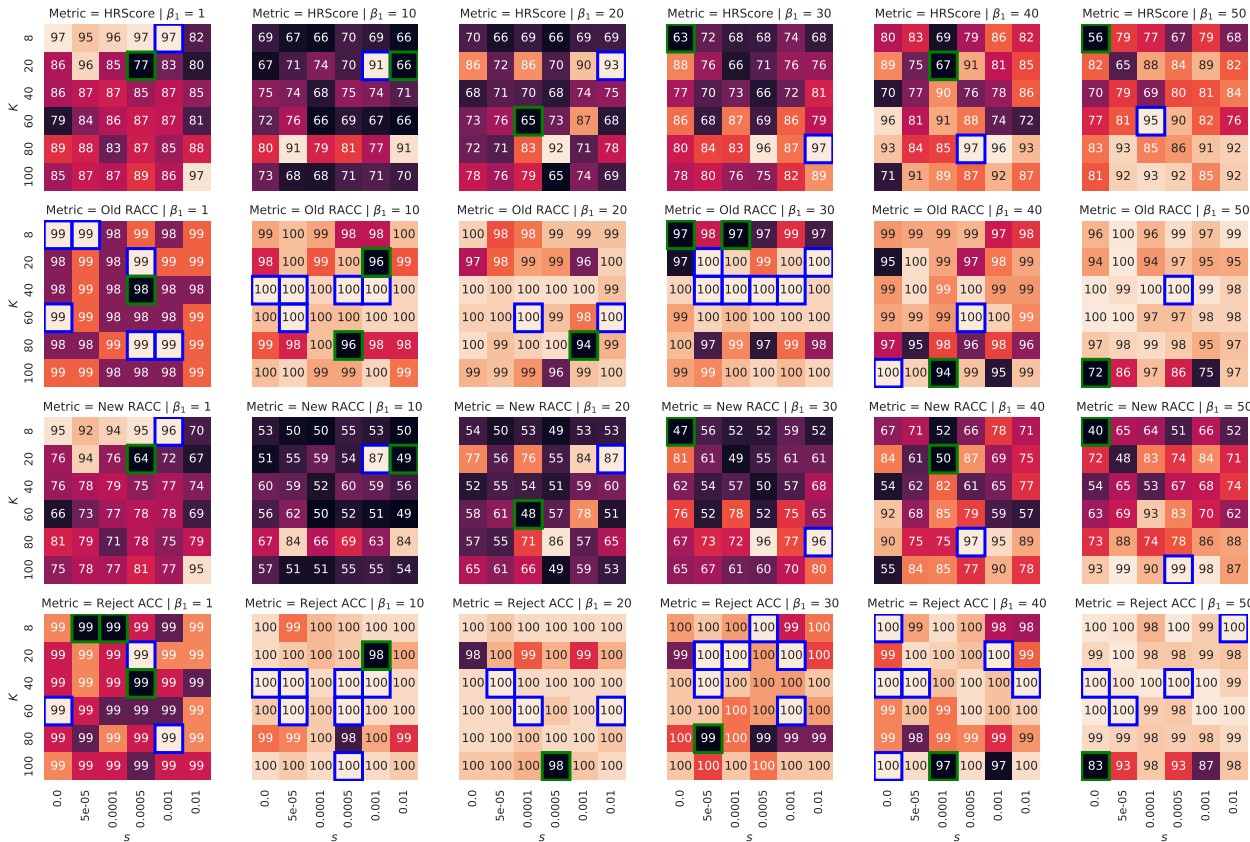

Figure 6: The hyperparameter sensitivity of our SWIRL on the CSBM graph $\mathcal{D}_1$, where the challenge is distinguishing new classes. Each row corresponds to a GCD metric, while each column represents a different value of the SWIRL loss weight $\beta_1$. At each position, the heatmap visualizes how the number of prototypes $K$ and the repulsion force scale $s$ jointly impact the GCD metric values. In each heatmap, the maximum value is highlighted with a blue box, and the minimum value is highlighted with a green box.

### G.7. Hyperparameter Analysis for SWIRL on Synthetic CSBM Graphs

SWIRL has three key hyperparameters to tune: $\beta_1$, $s$, and $K$. To analyze their impact on SWIRL, we conduct experiments on the synthetic CSBM graphs $\mathcal{D}_1$ and $\mathcal{D}_3$ described in Sec. G.4. The results of the hyperparameter grid search on these datasets are presented in Fig. 6 and Fig. 7, respectively. For each hyperparameter, we select six discrete values from the search space defined in Table 5, resulting in a total of 216 experiments ($6 \times 6 \times 6$ grid search). To comprehensively illustrate the relationship between hyperparameters and various GCD metrics, we organize the visualization using faceted plots based on four GCD metrics and six values of $\beta_1$. Within each heatmap, we show the effects of $s$ and $K$ on GCD performance.

For $\mathcal{D}_1$, the primary challenge is distinguishing different novel categories. As shown by Fig. 6, the second and fourth rows of the heatmaps indicate that the three hyperparameters have little impact on Old RACC and Reject ACC. When the number of prototypes is low (i.e., $K = 8$), SWIRL likely fails to adequately approximate the global structure, making it difficult to regulate the true global structure simply by increasing the SWIRL loss weight $\beta_1$. Since global structure is crucial for leveraging old-class knowledge in novel category discovery, both HRScore (first row) and New RACC (third row) exhibit a clear downward trend for $K = 8$ as $\beta_1$ increases. Conversely, when the number of prototypes is large (i.e., $K = 100$), the model approximates the global structure more effectively. As the regulation strength increases from left to right, New RACC (third row) and HRScore (first row) improve significantly. This suggests that when distinguishing novel categories is the primary challenge, a larger number of prototypes should be used alongside a higher SWIRL loss weight $\beta_1$.

For $\mathcal{D}_3$, the key challenge is distinguishing closely related known and novel categories. As displayed in Fig. 7, the first and fourth rows reveal that as $\beta_1$ increases, placing more emphasis on global structure, Reject ACC and HRScore decline

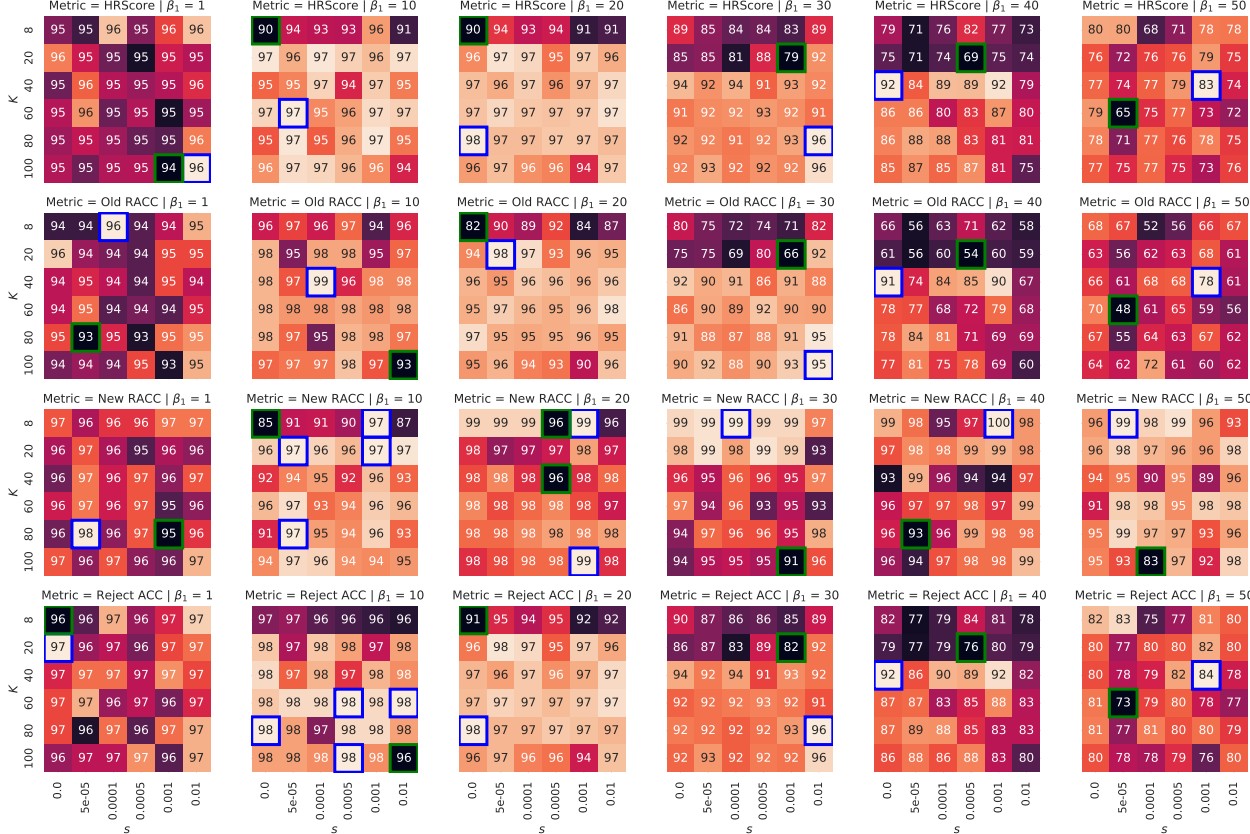

Figure 7: The hyperparameter sensitivity of our SWIRL on the CSBM graph $\mathcal{D}_3$, where the challenge is distinguishing between adjacent new and old classes. Each row corresponds to a GCD metric, while each column represents a different value of the SWIRL loss weight $\beta_1$. At each position, the heatmap visualizes how the number of prototypes $K$ and the repulsion force scale $s$ jointly impact the GCD metric values. In each heatmap, the maximum value is highlighted with a blue box, and the minimum value is highlighted with a green box.

significantly. This is likely due to overemphasizing global structure, which constrains instance-level discrimination between similar known and novel category nodes, making them harder to distinguish. To validate this hypothesis, we visualize the embedding space and decision boundaries from the experiment with $\beta_1 = 50$, $s = 0.01$ and $K = 100$ in Fig. 8. The visualization clearly shows that many old-class nodes are misclassified as novel categories, confirming that when the primary challenge is distinguishing adjacent known and novel categories, a smaller SWIRL loss weight $\beta_1$ should be used.

Across both $\mathcal{D}_1$ and $\mathcal{D}_3$, we observe that the optimal results are rarely achieved when $s = 0$. This demonstrates the effectiveness of SWIRL's distinct design, six levels of repulsion force, compared to other instance-to-prototype contrastive losses. Finally, identifying the primary challenge in a target dataset is non-trivial. Since this work primarily focuses on parametric GGCD theory and its validation, addressing this more practical issue is left for future research.

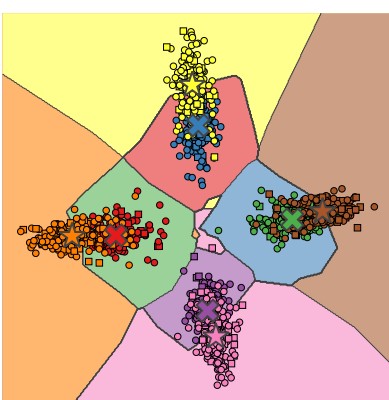

Figure 8: The learned embedding space of SWIRL on the CSBM graph $\mathcal{D}_3$, with $\beta_1 = 50$, $s = 0.01$ and $K = 100$.

