# OpenReview forum: "Towards Understanding Parametric Generalized Category Discovery on Graphs"
_ICML.cc/2025/Conference — ICML 2025 poster_

### Official Review · Reviewer_Mgiw · 2025-03-10

**Overall Recommendation:** 3

**Summary:**

This paper first proposes a theoretical analysis about parametric GCD, and then design a new graph contrastive learning method, SWIRL, with the insights from the theoretical analysis. They propose the first GCD loss upper bound theory and identifying some necessary conditions about category relationships for good GCD performance. Besides, they reveal that the current GCL methods can not satisfy these conditions from the lens of a pairwise Markov random field. The proposed GCL method alleviates the randomness of category relation in the embedding space.

**Claims And Evidence:**

Yes, all the claims made in the submission supported by clear and convincing evidence.

**Essential References Not Discussed:**

The references are comprehensive.

**Experimental Designs Or Analyses:**

I have reviewed all the experimental designs. I think these experiments are essentially complete and can support the authors' claims.

**Methods And Evaluation Criteria:**

Yes, the methods and evaluation criteria make sense for the problem at hand.

**Other Comments Or Suggestions:**

It appears that Lemma D.3 in Appendix D has not been utilized. Please verify this point and make the necessary revisions.

**Other Strengths And Weaknesses:**

**Strengths**:

1. Current (parametric) GCD methodologies lack a systematic theoretical analysis, and this paper fills that void.
2. The theoretical analysis in this paper inspired the authors' proposed SWIRL method, which has demonstrated high experimental performance. This indicates that the authors' theories can, to some extent, guide the design of GCD methods, a trait that is highly commendable in theoretical work.
3. The visualization of representations and decision boundaries (as depicted in Figures 1-3) intuitively elucidates the implications of the principal findings in Sections 3 and 4, thereby facilitating an understanding of the main theoritical results.

**Weaknesses**:

1. If I have not misunderstood, the term $D_{KL}(P_{\mathcal{C}}\|\bar{h})$ included in the upper bound, according to Proposition 3.9, should originate from a prerequisite condition: the data distribution is class-balanced. However, the authors did not declare this condition in Theorem 3.5.
2. This paper employs the Wasserstein distance (Definition 3.1) to quantify the relationship between old and new categories, which incorporates a hyperparameter $\lambda$ that balances the influence of features and labels. However, the subsequent calculations do not elucidate the determination of this value or the methodology for its selection (e.g., in Figure 1).
3. The new evaluation metrics proposed in this paper, such as HRScore, appear to actually come from existing work [1].
4. The meaning of $N$ in line 263, p5 is not given. What’s the difference between $N$ here and $n$ in Section 2.1?
5. In Sections 3 and 4, the authors demonstrate the embedding space and the Wasserstein distance between old and new categories in this space. Calculating this Wasserstein distance requires considering the semantic distances between categories in the label space. However, if the authors consider one-hot labels, the semantic distance between any pair of categories \( u \) and \( l \) would be the same as that between \( u \) and \( l' \). This implies that the distance would effectively reduce to a Wasserstein distance based solely on the representations \( z \). As a result, the authors' statement in line 290—"the embedding relations (i.e., distances) between categories are random and thus not easy to be aligned with the (either known or unknown) category semantic relations in \(\mathcal{Y}_C\)"—would become less meaningful.


[1] W. An et al., “Transfer and alignment network for generalized category discovery,” in Proceedings of the AAAI Conference on Artificial Intelligence, 2024.

**Questions For Authors:**

- Q1: In line 285, p6, the authors state that “The above presents a more refined treatment of the embedding space than [1]”. Can you provide more details about this comparison?

- Q2: In the propsoed SWIRL, the authors derive prototypes by clustering the average representations across two views. The rationale for not clustering the complete set of representations from both views remains unaddressed. Is there a discernible difference in performance between these two approaches?

- Q3: Prototype-based contrastive learning methods have been extensively explored in the literature. The advantage of SWIRL over these existing approaches appears to be primarily rooted in its theoretical motivation. Are there any other distinctive features that significantly differentiate SWIRL from current prototype-based contrastive learning methods?

I may change my score based on the authors' rebuttal regarding weaknesses and questions.

[1] Z. Tan, Y. Zhang, J. Yang, and Y. Yuan, “Contrastive Learning is Spectral Clustering on Similarity Graph,” presented at the The Twelfth International Conference on Learning Representations, 2023.

**Relation To Broader Scientific Literature:**

This article presents the first systematic theoretical analysis of GCD methods that employ a parametric classifier, addressing a gap in the current theoretical understanding of GCD methods. The authors' analysis of contrastive learning approaches utilizing InfoNCE and SupCon loss functions can, in fact, be regarded as an independent contribution to the theory of contrastive learning.

**Theoretical Claims:**

Yes. I have checked the proofs of the main results, i.e., Theorem 3.5 and Theorem 4.3. I didn't see any obvious mistakes.

---

> ### Author Rebuttal · Authors · 2025-03-30
>
> Thank you for the helpful comments and guidance!
>
> ### **Weaknesses**
>
> + **W1**: You are absolutely right! Thank you for your meticulous attention. We will revise this.
> + **W2**: In subsequent calculations involving **W**, we set $\lambda = 1$, with assuming that the embeddings and labels are important equally.
> + **W3**: The **H-score** in [1] is essentially the harmonic mean of **New ACC** and **Old ACC** (defined in **Appendix G.1**), both computed after remapping all classes via *linear sum assignment*. This may maps true new classes as old classes and vice versa, intruding the GCD purpose. Our proposed **HRScore** improves upon this by harmonizing **New RACC** and **Old RACC**—*without* remapping new clasess to old ones —better fitting GCD’s objectives. For formulas and distinctions, see **Appendix G.1**.
> + **W4**: Here, $n$ is the number of *observed* nodes in the graph dataset, while $N$ represents the total *potential* nodes (e.g., in citation networks, $N$ includes unpublished/unobserved papers). During contrastive learning, augmentations on $n$ nodes effectively select positive samples from $N \gg n$ nodes based on human priors.
> + **W5**: You’ve identified a key point we didn't elaborate on for a not that length manuscript. In the computation of Wasserstein distances in **Secs. 3 and 4**, class labels $\mathbf{y}^c$ are *not* one-hot. For example, in **Fig. 1**, adjacent old (yellow, $c=1$) and new (blue, $c=5$) classes are assigned:
>   $$
>   \mathbf{y}^1 = [0.95, 0, 0, 0, 0.05, 0, 0, 0, 0], \quad \mathbf{y}^5 = [0.05, 0, 0, 0, 0.95, 0, 0, 0, 0].
>   $$
>   Similar setups apply to other adjacent (old, new) class pairs. All **Wasserstein distances** in **Secs. 3–4** are computed under this scheme.
> Some experiments conducted by us but not included in this manuscript show such *non-one-hot* labels generally improve GCD performance since such semantic-aware labels facilitate transfering the knowledge of distinguishing old classes into distinguishing new ones, aligning with our statement that "*embedding-space category relations (distances) are random and hard to align with semantic relations in $\mathcal{Y}_C$*".
>
> ---
>
> ### **Questions**
>
> + **Q1**: Using our notation, the differences between [2] and our work are summarized below, where $\pi$ is our *relation prior* in embedding space and $\mathbf{B}$ is the prior in the input space. Our key point lies in constructing a **PMRF** for embeddings and introducing posterior probabilities for the node relation graph $\mathbf{W}$ in the embedding space:
>
>   |                | [2]                          | Ours                                                                 |
>   |----------------|------------------------------|----------------------------------------------------------------------|
>   | **Input Space**  | $P(\mathbf{W}_X; \mathbf{B})$ | $P(\mathbf{W}_X; \mathbf{B})$                                    |
>   | **Embedding Space** | $P(\mathbf{W}; \mathbf{Z})$ | $P(\mathbf{W} \mid \mathbf{Z}) \propto P(\mathbf{Z} \mid \mathbf{W}) P(\mathbf{W}; \pi)$ |
>
> + **Q2**: Our design is motivated by:
>   1. **Reduced computation**: Running **SS-KM** on $n$ samples is far cheaper than on $2n$.
>   2. **Enhanced consistency**: Prototypes derived from *averaged node embeddings* resemble clustering "mini-cluster" centroids from paired views, improving positive-pair alignment. Our empirical tests confirm that this strategy is better.
>
> + **Q3**: **SWIRL**’s distinction from other prototype contrastive methods is its *multi-level repulsion force*, whose efficacy is highlighted in **Line 2015 (p.38)** of the manuscript.
>
> ---
>
> ### **References**
> [1] *Transfer and Alignment Network for Generalized Category Discovery* , AAAI2024
>
> [2] *Contrastive Learning is Spectral Clustering on Similarity Graph*, ICLR2024

---

### Official Review · Reviewer_Af98 · 2025-03-11

**Overall Recommendation:** 3

**Summary:**

This paper introduces Generalized Category Discovery on Graphs (GGCD), a novel task addressing open-world learning on graph-structured data where unlabeled nodes may belong to both known and novel classes. The authors defined a surrogate GCD loss to reflect the GCD performance and established a theoretical bound. To minimize this bound, they argue that it is necessary to reduce the Wasserstein distance between the distributions of known and novel classes while maintaining their independence. Subsequently, they analyze the limitations of existing baseline methods in satisfying these conditions and propose a heuristic approach, SWIRL, to address these challenges. The experiments demonstrate SWIRL’s effectiveness and support the theoretical claims.

**Claims And Evidence:**

Overall, the claims here are well-supported by clear and convincing evidence.

**Essential References Not Discussed:**

No.

**Experimental Designs Or Analyses:**

Yes, I reviewed the experimental design and analyses, and overall, they are well-structured and appropriate for validating the paper’s claims. However, one issue is the lack of scalability analysis for SWIRL.

**Methods And Evaluation Criteria:**

The datasets are pervasive, and the criteria make sense.

**Other Comments Or Suggestions:**

Typos:

-	Section 1. Paragraph 4. Transferrs -> Transfers

-	P20. Postive-pair-> Positive-pair

Suggestions:

Subsection 4.1 is hard to read for those not familiar with MRF and the underlying probabilistic graph model is not very clear under the context of contrastive learning. It would be helpful to provide some intuitive examples or diagrams to illustrate how MRF concepts integrate with contrastive learning, thereby enhancing the readability and clarity of this section.

**Other Strengths And Weaknesses:**

Strengths:

-	The theory assumptions are well discussed and not very strong.

-	Many theory claims are verified with controllable experiments excluding irrelevant factors not mentioned in theories.

Weaknesses:

-	This paper has so much math stuff that completely grasping it is not easy.

-	The three GCD baseline models from the computer vision community are not up to date.

**Questions For Authors:**

1: It is interesting to know if it is possible to extend the theoretical analysis in this article to non-parametric GCD methods, e.g., GCD [1].

2: What’s the relationship between $\mathbf{W}_X$ in Sec. 4 and the data graph $\mathbf{A}$?

[1] Generalized Category Discovery

**Relation To Broader Scientific Literature:**

The paper advances generalized category discovery (GCD) and (open-world) graph machine learning.

-	It extends GCD [1] to graph data

-	It introduces a theoretical framework based on Wasserstein distance to quantify the relationships between old and new classes, diverging from prior GCD theories based on mutual information [2] or pairwise consistency distribution [3]. With this framework, a GCD loss upper bound is achieved.

-	This loss bound leads to two category relation conditions that the representation learning should meet. And the authors critiques popular graph contrastive learning (GCL) paradigm through a PMRF perspective, revealing its inability to control global category relations. This flaw is not discoverable with prior contrastive learning theory [4].

-	Additionally, it contextualizes known GCN smoothing effects within GCD, demonstrating how oversmoothing disrupts category separation.


[1] Generalized Category Discovery

[2] Parametric Information Maximization for Generalized Category Discovery

[3] SelEx: Self-expertise in fine-grained generalized category discovery

[4] Contrastive Learning is Spectral Clustering on Similarity Graph

**Theoretical Claims:**

I have reviewed the proof of Theorem 3.5, and it is substantially correct under the assumptions employed by the authors.

---

> ### Author Rebuttal · Authors · 2025-03-30
>
> We are grateful for your constructive suggestions!
>
> ### **Experimental Designs or Analyses**
> We will include a **complexity analysis** in the revised version. Let:
> - $n$: Number of samples
> - $K$: Number of prototypes
> - $I$: SS-KM iterations
> - $d$: Embedding dimension
>
> **SS-KM** (executed every $t$ epochs):
> - **Space**: $O(nd + Kd)$
> - **Time**: $O(nKId)$ (amortized: $O(nKId/t)$ per epoch)
>
> **SWIRL loss** (per epoch):
> - Similarity computation: $O(nKd)$
> - Denominator summation + division: $O(nK)$
> - **Total time**: $O(nKd)$
> - **Space**: $O(nK)$ (similarity matrix)
>
> **Overall**:
> - **Time**: $O(TnKd + TnKdI/t)$ → With $I=10$, $t=20$: $O(TnKd)$
> - **Space**: $O(nK + nd)$
> - Compared to **InfoNCE** ($O(n^2d)$), this overhead is negligible.
>
> ---
>
> ### **Weaknesses**
> + **W1**: We will add schematic diagrams to **Sections 3 and 4** for clarity.
> + **W2**: This paper’s primary goal is *not* to propose the highest-performing method, but to provide a **theoretical analysis** of parametric GCD’s core motivation—*leveraging old-class knowledge to distinguish new classes*—and validate it via controlled experiments. **SWIRL** serves only to demonstrate that our theory can inspire new designs. Thus, comparisons with state-of-the-art GCD baselines are deferred to future work.
>
> ---
>
> ### **Suggestions**
> **Typos**: Thank you for your meticulous review!
>
> + **S1**: We treat node embeddings $z_i$ ($i \in [N]$) as random variables with pairwise dependencies defined by $\mathbf{W}$. Here, $W_{ij}=1$ indicates a relationship between nodes $i$ and $j$. $\mathbf{W}$ can be seen as the underlying graph of node embeddings, charterizing the relationships between node embedding random variables and the corresponding pairwise Markov random field (PMRF). The joint distribution on this PMRF is:
>   \[
>   f_k(\mathbf{Z}, \mathbf{W}) \propto \prod_{(i,j) \in [N]^2} k(\mathbf{z}_i - \mathbf{z}_j)^{W_{ij}}.
>   \]
>   Each class corresponds to a connected component/cluster of the graph $\mathbf{W}$, with its center representing global class position in the embedding space. **Theorems 4.1–4.3** show these centers have *infinite variance*, making their positions uncontrollable.
>   We will add diagrams to illustrate this intuitively in the revised manuscript.
>
> ---
>
> ### **Questions**
> + **Q1**: The conclusions in **Sec. 4** apply to *all* representation learning methods using **InfoNCE** or **SupCon loss** (thus including GCD [1]). For **Sec. 3**, the assumptions (e.g., Lipschitzness of $h$) do *not* hold for GCD’s SS-KM classifier.
> + **Q2**: $\mathbf{A}$ is the observed graph of $n$ nodes—a tiny subset of $N$ possible (augmented) nodes. Each augmentation step connects these $n$ nodes to another $n$ positives in $\mathbf{W}_X$ (an $N \times N$ graph with $N-2n$ isolated nodes). For theoretical purposes, large $N-2n$ does *not* matter, and the methods inspired by theoretical results actually plays with the small $n\times n$ graph.
>
> [1] *Generalized Category Discovery* , CVPR2022

---

> > ### Comment · Reviewer_Af98 · 2025-04-03
> >
> > Thanks for the response. Most of my concerns have been addressed. The remaining concern is that SS-KM employed in SWIRL appears to require application to the entire dataset at the beginning or end of each epoch. However, in many cases, the sample size n is excessively large, making the computational complexity potentially prohibitive. Could we alternatively implement it on mini-batches of b samples to reduce computational complexity?

---

> > > ### Author Response · Authors · 2025-04-04
> > >
> > > We sincerely appreciate your thorough review and insightful questions. In SWIRL, the SS-KM algorithm can indeed be applied at the batch level: within each mini-batch, SS-KM is used to obtain cluster centroids and assignments, which are then linearly combined with the old centroids from before processing the mini-batch via momentum-based updating. In fact, this approach was adopted in our experiments on **CIFAR100** (please see **W1 in our response to Reviewer Hzoz**). However, in the graph experiments reported in the manuscript, since most datasets did not require training in mini-batches, we did not implement this design. In future work, we will update the code to support mini-batch training on larger datasets.

---

### Official Review · Reviewer_ZGRQ · 2025-03-12

**Overall Recommendation:** 4

**Summary:**

The paper focuses Graph Generalized Category Discovery (GGCD), an node-level task aimed at identifying both known and novel categories in unlabeled nodes by leveraging knowledge from labeled old classes. The authors provide the first theoretical analysis for parametric GCD on graphs, quantifying the relationship between old and new classes using the Wasserstein distance in the embedding space. This analysis establishes a provable upper bound for the GCD loss, revealing two critical conditions: (1) small Wasserstein distance between old and new classes to transfer discriminative knowledge, and (2) sufficient separation to avoid overlap between old and new classes. Through a Pairwise Markov Random Field (PMRF) perspective, they show that mainstream graph contrastive learning (GCL) methods employing InfoNCE or SupCon losses inherently violate these conditions due to uncontrolled category relations and the smoothing effect of GCN encoders. To address this, the paper proposes SWIRL, a new GCL method designed to better control global category relations. Experiments on synthetic and real-world graphs validate the theoretical findings and show SWIRL's effectiveness

**Claims And Evidence:**

The efficacy of the SWIRL method has been amply demonstrated. The principal theoretical outcomes of the authors were validated under a specific category distribution layout (as illustrated in Figure 2). Although the results align closely with the theory, it is incumbent upon the authors to further elucidate whether this layout sufficiently attests to the universality of the theoretical findings.

**Essential References Not Discussed:**

No

**Experimental Designs Or Analyses:**

I have checked all claims, experimental designs, and analyses. Overall, the designs and analyses are valid to demonstrate authors’ claims. However, in the experiments of Section 4, the authors should present and discuss the performance of SWIRL-GPR, which is trained using the SWIRL method with a GPR encoder.

**Methods And Evaluation Criteria:**

The authors have employed reasonable evaluation metrics, and the proposed method effectively addresses the target problem at hand.

**Other Comments Or Suggestions:**

- The last sentence of Assumption 3.3 mentions that the author's experiments also support the equivalence between \( L = D_{CE} \) and \( L = MSE \), but the author does not analyze this in the results discussions.

- In Appendix G.1, the author has improved the existing evaluation protocol, but the explanation of its advantages is not sufficiently intuitive. I think some specific examples would help readers quickly grasp the key points of the improvements.

**Other Strengths And Weaknesses:**

**Strengths**

-  The authors represent the first attempt to quantify the relationship between the distributions of old and new classes, which I think is crucial for GCD analysis, particularly for the analysis of generalization error.
- The theoretical findings provide valuable insights that can inform the development of novel GCD methodologies.
- The proposed method is designed with the inspiration from authors’ theoretical analyses.

**Weaknesses**

- The final hyperparameters for all methods are not reported.
- Despite the extensive theoretical content presented earlier, the method proposed by the authors, SWIRL, appears to be essentially heuristic in nature.
- The author's title suggests an exploration of generalized category discovery on graphs, but the content of the article primarily focuses on GCD methods that utilize parametric classifiers. I believe this is not appropriate.
- The paper is too mathematical to read in some parts (e.g., Sec. 4.1).

**Questions For Authors:**

1. Do the theoretical results in Sections 3 and 4 directly apply to parametric GCD methods for image data?
2. The computational efficiency of the InfoNCE loss is high when the batch size is large. Some works in other fields [1] have found that the pairwise sigmoid loss without global normalization can enable more efficient contrastive training. Is it possible for your theory in Section 4 to extend to contrastive learning algorithms that utilize this type of loss?
3. The loss \( L_{te}(h) \) (Eq. 5), which reflects the GCD performance of the classifier \( h \), is a weighted sum of three GCD capabilities. However, how are the weights for the three components determined?
4. In the illustrative experiments (Sections 3.4 and 4.3), the authors employed a similar spatial arrangement for the category space: new classes are situated inside, while old classes are placed outside. Why was this design necessary? Have the authors experimented with other types of layouts?

[1] Sigmoid Loss for Language Image Pre-Training, ICCV2023

**Relation To Broader Scientific Literature:**

This paper extends GCD from image domain to graph domain, with a novel theoretical analysis on GCD methods (e.g., SimGCD) that relies on parametric classification heads. Though the application cases are about graphs, most analyses in this work seem to fit into the cases of non-graph data. Hence, it can be taken as the first comprehensive analysis of parametric GCD performance from both classification head and representation learning sides.

**Theoretical Claims:**

I haven't looked at the proof process, but these theorems do align with my intuition.

---

> ### Author Rebuttal · Authors · 2025-03-30
>
> We sincerely thank you for the valuable comments that helped improve our paper!
>
> ---
>
> ### **Claims and Evidence**
> **Theorem 3.5** summarizes three key aspects of **GCD**: **Old**, **New**, and **Reject Capability**. This work primarily focuses on the latter two (involving new-class data), so we intentionally made old classes easily distinguishable to eliminate interference and study how the ability to separate old classes transfers to new ones. From this perspective, our experimental setup validates the **New/Reject Capability** aspects of **Theorem 3.5**.
>
> For the **Old Capability** part of **Theorem 3.5**, we previously conducted experiments under an alternative setting: placing old classes *inside* and new classes *outside* (current old-class positions). The results confirmed that insufficient Old Capability negatively impacts new-class separation, aligning with theoretical predictions.
>
>
>
> ### **Experimental Designs or Analyses**
> The results for **SWIRL-GPR** are shown below. On dataset **D1**, SWIRL outperforms **SimGCD-GPR**, while the opposite holds for **D3**. Upon visualizing the representation space of **SWIRL-GPR on D3**, we observed significantly weaker intra-cluster cohesion compared to **SWIRL-GCN**, leading to poorer GCD performance. We attribute this to both **GPR encoders** and **SWIRL** emphasizing information from distant nodes/spaces *beyond local neighborhoods*, which neglects local structural cues and hinders tight cluster formation.
>
> Additionally, since **GPR encoders** are inherently challenging to train in unsupervised graph contrastive learning (e.g., via GRACE-style methods) [1], we recommend pairing **SWIRL** with a **GCN encoder** for better local-space cohesion.
>
> | **D1**       | **HRScore** | **Old RACC** | **New RACC** | **Reject ACC** |
> |--------------|------------|--------------|--------------|----------------|
> | SimGCD-GPR   | 77.18      | 99.79        | 62.92        | 99.48          |
> |SWIRL-GPR| **87.72**  | **100.0**    | **78.13**    | 98.54          |
>
> | **D3**       | **HRScore** | **Old RACC** | **New RACC** | **Reject ACC** |
> |--------------|------------|--------------|--------------|----------------|
> | SimGCD-GPR   | **88.77**  | **98.12**    | **81.04**    | 90.52          |
> | SWIRL-GPR    | 82.39      | 96.88        | 71.67        | 89.58          |
>
> ### **Weaknesses**
> + **W1**: We will promptly release the code and document all hyperparameter settings in configuration files.
> + **W2**: This work focuses on *theoretically* explaining how old-class knowledge transfers in GCD. We achieved this goal, and our theoretical results guide GCD method design. The heuristic **SWIRL** merely validates this guidance; rigorous provable methods are left for future work.
> + **W3**: Thank you—we will explicitly clarify the paper’s scope in the revision.
> + **W4**: We will add schematic diagrams for **Sections 3 and 4** to improve readability.
> ### **Suggestions**
> + **S1**: Although not re-emphasized, all verification experiments used $L = D_{CE}$ (cross-entropy loss), and results aligned well with theoretical predictions, demonstrating their partial equivalence.
> + **S2**: Practical examples will be added in the revised version.
> ### **Questions**
> + **Q1**: Yes, except the content about GCN encoder.
> + **Q2**: Excluding global normalization violates the PMRF’s fundamental dependency assumption, making our framework incompatible with such losses.
> + **Q3**: This ultimately depends on which capability is prioritized in a specific application. Regarding the theoritical side, it does not influence the bound formulation.
> + **Q4**: Please refer to **Claims and Evidence**.
>
>
> [1] PolyGCL: GRAPH CONTRASTIVE LEARNING via Learnable Spectral Polynomial Filters, ICLR2024

---

> > ### Comment · Reviewer_ZGRQ · 2025-04-05
> >
> > I appreciate the authors' response and are pleased to see the additional experimental results and discussion. We kindly request the authors to share the schematic diagrams designed for Sections 3 and 4 at their earliest convenience.

---

### Official Review · Reviewer_Hzoz · 2025-03-13

**Overall Recommendation:** 3

**Summary:**

This paper studies the generalized category discovery problem in graph node classification context. The authors aim to answer the question “When and how do old classes help (parametric) generalized category discovery on graphs?” in a theoretical way. The answer from the authors is based on the relationships between old and new categories in the embedding space, the elements of which are later fed into a parametric classifier to make final predictions. Specifically, the Wasserstein distance between the distributions of old-class and new-class embeddings should be minimized, provided that the embeddings of the old and new categories do not overlap. Later, the authors find that the representation learning methods in current GCD methods tend not to meet this. And to address this issue, they propose a new GCL method, SWIRL.

**Claims And Evidence:**

Yes.

**Essential References Not Discussed:**

No.

**Experimental Designs Or Analyses:**

Yes. This paper has sound and valid experimental designs to support the claims of authors.

**Methods And Evaluation Criteria:**

Yes.

**Other Comments Or Suggestions:**

See the weakness part.

**Other Strengths And Weaknesses:**

Strengths:
1.	This paper is well-organized to answer the question “When and how do old classes help (parametric) generalized category discovery on graphs?”
2.	The theorems answer the question clearly and the proposed SWIRL is motivated by the theoritical results.
3.	The hyperparameter analysis shows that the proposed method is not sensitive to hyperparameters, which is good for practice.

Weaknesses:

1.	I suggest the authors test SWIRL on image datasets such as CIFAR100 to make this work more comprehensive.

2.	There are a lot of theorems, and the paper is very dense in math. More intuitive discussion can help readers to understand.
3.	In line 1938, p36, 035 should be 0.35.

4.	For all datasets, the weights of supervision and self-supervision losses are fixed to 0.35 and 0.65, respectively. Is this an appropriate way?

5.	In line 2076, p38, the authors state that “compared to other instance-to-prototype contrastive loss”. This statement is not accurate since $s=0$ can only derive one form of instance-to-prototype contrastive loss, whereas in reality, there are many other forms of instance-to-prototype contrastive losses.

6.	In Sec. 4.1, the authors say that $\mathbf{W}_{X}$ is the subgraph sampled from $\mathbf{B}$. But no clear sampling definition and meaning are provided, leading to some confusion. What’s the shape of the sampled subgraph?

Furthermore, how can $\Omega_{D}(\mathbf{W})$ align with the single-positive-sample manner in GCL methods such as GRACE?

**Questions For Authors:**

See the weakness part.

**Relation To Broader Scientific Literature:**

This paper is, as far as I know, the first theoretical work to answer “When and how do old classes help (parametric) generalized category discovery?” in the field of GCD. But the theories seem to hold only for those methods using parametric classification head (e.g., MLP). In spite of this, the quantization of the relationship between old and new categories and the GCD loss upper bound make this work quite novel. The analysis of contrastive learning methods in Sec. 4 is insightful to design new representation learning methods for GCD problem. Though the authors put a graph context, the main results in this work seem to hold for image data as well.

**Theoretical Claims:**

No.

---

> ### Author Rebuttal · Authors · 2025-03-30
>
> Thanks for your constructive suggestions! We hope the response can address your concerns.
>
> + **W1**: When keeping other hyperparameters consistent, adding **SWIRL** loss alone to **SimGCD** not only achieves a better **HRScore** but also reaches a high performance level as fast as at **epoch=5**.  The implementation is based on the code https://github.com/CVMI-Lab/SimGCD.
>
> |   CIFAR100       | HRScore (epoch = 5) | HRScore (epoch = 200)| Old RACC | New RACC | Reject ACC |
> |----------|----------|------------|----------|----------|------------|
> | **SimGCD** | 33.9     | 77.9       | 81.8     | 74.3     | 91.9       |
> | **SWIRL**  | **65.7**     | **78.4**      | 78.4     | 78.1     | 90.8       |
>
> + **W2**: In the revised version, we will include a schematic diagram to illustrate the theoretical result in **Sec. 3**: semantically similar categories should also be close in the representation space. Additionally, we will add another diagram to visualize the **PMRF perspective of GCL** mentioned in **Sec. 4**.
>
> + **W3**: We have carefully checked the entire manuscript to ensure no similar errors remain.
>
> + **W4**: For **parametric GCD methods** (the main focus of this work), preventing new-class samples from being misclassified as old classes is crucial. Thus, the weight of the supervised signal typically needs to be lower than that of the self-supervised signal. Here, we follow the hyperparameter selection of **SimGCD**. Moreover, since all methods use the same weight settings, adopting different weights on other datasets does not affect the conclusions of this paper.
>
> + **W5**: Thank you for pointing this out. To be precise, when $s=0$, the **SWIRL loss** closely resembles **PCL [1]**. The key distinction of our method lies in how we **push away different samples and clusters with varying repulsion froce**.
>
> + **W6**: $\mathbf{B}$ represents the relationship among all **N samples** (including all possible augmented views) in our framework. The dataset of **n nodes** is just a subset. $B_{ij}$ denotes the similarity between nodes **i** and **j**, as well as the probability of sampling them as a positive pair. In each augmentation step, we sample one positive node **j** for each node **i**. Once sampled, the edge **(i,j)** becomes part of the subgraph $\mathbf{W}_X$. Since **n << N**, the **N × N** matrix $\mathbf{W}_X$ is actually a small graph with **2n non-isolated nodes**. The remaining **N-2n** samples are isolated and retained in $\mathbf{W}_X$  only for formulation convenience.
>
> + **W7**:   $\Omega_D(\mathbf{W})$ equals **1** only when each row of $\mathbf{W}_X$ has exactly one non-zero element (i.e., each node has only one neighbor). Thus, multiplying by it ensures that in the sampled subgraph each node has a single positive node.
>
> [1] *Prototypical Contrastive Learning of Unsupervised Representations*, ICLR 2021.

---

> > ### Comment · Reviewer_Hzoz · 2025-04-03
> >
> > The authors have addressed my questions. I maintain my score and support its acceptance.

---

### Decision · Program_Chairs · 2025-05-01

**Decision:**

Accept (poster)

**Comment:**

This paper presents the first theoretical analysis of parametric generalized category discovery (GCD) on graphs, introducing a GCD loss upper bound and identifying conditions under which old-class knowledge benefits the discovery of new categories. The proposed SWIRL framework is derived from these insights and empirically validated. Key concerns were raised initially, including the scalability and efficiency of SWIRL and technical details. The authors provided comprehensive and thoughtful responses to all raised issues, and the consensus is that it meets the bar for acceptance.